# Graph Attention is Not Always Beneficial: A Theoretical Analysis of Graph Attention Mechanisms via Contextual Stochastic Block Models

**Zhongtian Ma** [1 2]    **Qiaosheng Zhang** [2 3]    **Bocheng Zhou** [2 4]    **Yexin Zhang** [1 2]    **Shuyue Hu** [2]    **Zhen Wang** [1]

## Abstract

Despite the growing popularity of graph attention mechanisms, their theoretical understanding remains limited. This paper aims to explore the conditions under which these mechanisms are effective in node classification tasks through the lens of Contextual Stochastic Block Models (CSBMs). Our theoretical analysis reveals that incorporating graph attention mechanisms is *not universally beneficial*. Specifically, by appropriately defining *structure noise* and *feature noise* in graphs, we show that graph attention mechanisms can enhance classification performance when structure noise exceeds feature noise. Conversely, when feature noise predominates, simpler graph convolution operations are more effective. Furthermore, we examine the over-smoothing phenomenon and show that, in the high signal-to-noise ratio (SNR) regime, graph convolutional networks suffer from over-smoothing, whereas graph attention mechanisms can effectively resolve this issue. Building on these insights, we propose a novel multi-layer Graph Attention Network (GAT) architecture that significantly outperforms single-layer GATs in achieving *perfect node classification* in CSBMs, relaxing the SNR requirement from $\omega(\sqrt{\log n})$ to $\omega(\sqrt{\log n}/\sqrt[3]{n})$. To our knowledge, this is the first study to delineate the conditions for perfect node classification using multi-layer GATs. Our theoretical contributions are corroborated by extensive experiments on both synthetic and real-world datasets, highlighting the practical implications of our findings. [1]

[1]Northwestern Polytechnical University [2]Shanghai Artificial Intelligence Laboratory [3]Shanghai Innovation Institute [4]Shanghai Jiao Tong University. Correspondence to: Zhen Wang <w-zhen@nwpu.edu.cn>, Qiaosheng Zhang <zhangqiaosheng@pjlab.org.cn>.

*Proceedings of the $42^{nd}$ International Conference on Machine Learning*, Vancouver, Canada. PMLR 267, 2025. Copyright 2025 by the author(s).

[1]The source code for this article is available on https://github.com/mztmzt/GAT_CSBM.

## 1. Introduction

Graph Neural Networks (GNNs) have become essential tools for analyzing graph-structured data, with applications in social networks (Fan et al., 2019), biology (Gligorijević et al., 2021), computer vision (Ma et al., 2022) and recommendation systems (Wu et al., 2020; 2022a). A foundational approach within GNNs is the Graph Convolutional Network (GCN) (Kipf & Welling, 2022), which aggregates information from a node's neighbors to generate feature representations. Building on GCNs, Graph Attention Networks (GATs) (Veličković et al., 2018) introduce the *graph attention mechanism* that dynamically assigns weights to neighboring nodes based on the similarity of their features, thereby enhancing performance by prioritizing the most relevant information.

Despite the growing interest in graph attention mechanisms (Wang et al., 2019c; Lee et al., 2019; Wang et al., 2019a;b; Hu et al., 2020), the understanding of when and why they are effective remains limited. While these mechanisms are designed to prioritize relevant nodes in a graph, their effectiveness appears to be highly influenced by the graph's properties, particularly in the presence of noise. The graph data commonly used in contemporary tasks is *featured graph*, containing both topological and node feature information. Consequently, two types of noise emerge: *structure noise* and *feature noise*. Structure noise disrupts graph connections, complicating the accurate identification of community structures. Feature noise refers to inaccuracies in node feature information, such as imprecise values or excessive similarity among features of different nodes, which can lead to incorrect classifications (Yang et al., 2024). Given that both types of noise have the potential to affect the performance of attention mechanisms, a critical question arises: **What factors influence the effectiveness of graph attention mechanisms, and how do structure noise and feature noise impact their performance in different scenarios?**

This paper addresses this question by providing an in-depth theoretical analysis of the graph attention mechanism. We employ the Contextual Stochastic Block Model (CSBM) (Deshpande et al., 2018), a commonly used tool for simulating graph structures and node features to model real

graph data. In the CSBM, the graph structure is generated using the well-known Stochastic Block Model (SBM) (Holland et al., 1983)—a random graph model that consists of community structures, while the node features are generated through a Gaussian Mixture Model (GMM) (Reynolds et al., 2009). A key focus in the CSBM is the signal-to-noise ratio (SNR) of the node features, linked to the mean and variance parameters of the GMM. A higher SNR indicates greater distinguishability of the node features. By utilizing the CSBM, we can precisely control levels of structure noise and feature noise by tuning model parameters—structure noise relates to connection probabilities between different communities in the SBM, while feature noise is defined as the inverse of the SNR[2]. Moreover, we use *node classification*, a fundamental task in graph learning that is widely employed to explore GNN properties (Baranwal et al., 2023; Wei et al., 2022), as a benchmark to assess the effectiveness of graph attention mechanisms across different levels of structure and feature noise.

Through our investigation, we provide a clear understanding of how graph attention mechanisms can be leveraged more effectively, and identify scenarios where simpler GCNs may provide better performance. By rigorously analyzing the impact of graph attention in the context of CSBM, this paper not only advances theoretical understandings but also provides valuable insights for practical applications in various domains. Our main contributions are as follows:

**Main Contributions**

- Inspired by (Fountoulakis et al., 2023), we design a nonlinear graph attention mechanism and show that its effectiveness is comparable to the mechanism in (Fountoulakis et al., 2023), while being simpler and easier to analyze (Theorem 1). Then, by analyzing the changes in SNR after applying graph attention layers (Theorem 2), we show that the graph attention mechanism is *not always* effective. Specifically, when the structure noise of the graph exceeds the feature noise, incorporating graph attention is beneficial, with higher attention intensity yielding better results. Conversely, when the feature noise of the nodes is greater than the structure noise, using graph attention can degrade node classification performance. In such cases, a simple graph convolution is more effective (see Section 3.2.1 for details).

- We investigate the impact of the graph attention mechanism on the *over-smoothing* problem. First, we introduce a refined definition of over-smoothing in an asymptotic setting where the number of nodes $n$ approaches infinity, highlighting its occurrence when the network depth is $O(n)$. We then show that for featured graphs generated

by the CSBM, the graph attention mechanism is able to resolve the over-smoothing issue in the high SNR regime (see Theorem 3).

- Building on our analysis of the graph attention mechanism, we design an effective multi-layer GAT and demonstrate that it significantly outperforms the single-layer GAT in achieving *perfect node classification* (see Definition 1). Specifically, the requirement is relaxed from SNR $= \omega(\sqrt{\log n})$ as stated in (Fountoulakis et al., 2023), to SNR $= \omega(\sqrt{\log n}/\sqrt[3]{n})$ (see Theorem 4). To our knowledge, this is the first study to examine the conditions for perfect node classification with multi-layer GATs.

- We conduct extensive experiments on synthetic datasets, as well as on three widely used real-world datasets, to validate our theoretical findings.

### 1.1. Related Works

In recent years, there has been growing interest in the theoretical analysis of GNNs, particularly using the CS-BMs (Baranwal et al., 2021; 2023; Luan et al., 2023; Adam-Day et al., 2024; Wang et al., 2024; Javaloy et al., 2023). Among these works, the two most relevant to our study are (Fountoulakis et al., 2023; Javaloy et al., 2023), whose settings are partially adopted in our work. Fountoulakis et al. (2023) primarily investigate the role of graph attention mechanisms in the presence of structural noise, where the graph itself provides limited information. They are the first to establish the feasible region for achieving perfect node classification using a single-layer GAT. Motivated by similar challenges, Javaloy et al. (2023) propose a learnable GAT, termed L-CAT, which combines the strengths of GCNs and GATs to address cases where GATs may not always outperform GCNs. Our work broadens this perspective by analyzing the effects of both structural and feature noise on graph attention mechanisms. We identify the precise regimes where GCNs or GATs perform better, extend the feasible region for perfect node classification to multi-layer GATs, and achieve improved results on sparse graphs compared to Javaloy et al. (2023).

The issue of over-smoothing in GNNs has also garnered extensive attention (Xu et al., 2018; Keriven, 2022; Liu et al., 2020; Yang et al., 2020; Zhao & Akoglu, 2020). Two closely related works are (Wu et al., 2022b; 2024), both of which theoretically explore the over-smoothing problem in GNNs. Wu et al. (2022b) analyzes how the SNR evolves through GCN layers within the CSBM framework, showing that GCNs experience over-smoothing after $O(\log n/\log(\log n))$ layers. In (Wu et al., 2024), the authors examine the impact of the graph attention mechanism on over-smoothing and concludes that it does not resolve the issue. In contrast, this paper investigates the effect of

---

[2]We refer readers to Eqn. 2 for detailed definitions of structure noise, feature noise, and SNR.

the graph attention mechanism on over-smoothing within the CSBM framework, demonstrating that under suitable conditions, a well-designed GAT can avoid over-smoothing for up to $\Theta(n)$ layers.

Finally, research on community detection within SBMs is also pertinent to our study (Abbe, 2018; Abbe & Sandon, 2015; Zhang & Zhou, 2016; Zhang & Tan, 2022; Chen et al., 2020). Specifically, the problem of community detection in CSBMs has recently attracted considerable attention from statisticians, including investigations into thresholds for exact and almost exact recovery and algorithm design (Lu & Sen, 2023; Deshpande et al., 2018; Braun et al., 2022; Duranthon & Zdeborova, 2024; Dreveton et al., 2024). The node perfect classification problem examined in this paper is analogous to performing exact node recovery in the community detection problem.

## 2. Preliminaries and Problem Setup

**Notations:** For any positive integer $a$, let $[a] \triangleq \{1, 2, \ldots, a\}$. For an undirected graph $\mathcal{G}$ with $n$ nodes, we use the adjacency matrix $\mathbf{A} \in \{0, 1\}^{n \times n}$ to represent the graph, such that for any $(i, j) \in [n] \times [n]$, $A_{ij} = 1$ if $i$ and $j$ are connected, and $A_{ij} = 0$ otherwise. Besides, we consider a featured graph where we use $\mathbf{X} \in \mathbb{R}^{n \times d}$ to represent the features for all $n$ nodes, with $\mathbf{X}_i \in \mathbb{R}^{1 \times d}$ denoting the feature of node $i$. When the dimension $d = 1$ (as considered in Section 2.1 and from Section 3 onwards), we use un-bold letters $X$ or $X_i$ instead. We use standard *asymptotic notations*, including $O(.), o(.), \Omega(.), \omega(.)$, and $\Theta(.)$, to describe the limiting behaviour of functions/sequences (Leiserson et al., 2001).

Let $\|\cdot\|_F$ be the Frobenius norm. Let $\text{sgn}(\cdot)$ denote to the *sign function* that maps a number to $-1, 0$, or 1 based on its sign. Let $\Phi(\cdot)$ be the cumulative distribution function of the standard Gaussian distribution. For an event $\Delta$, we denote by $\mathbb{1}\{\Delta\}$ the *indicator function*, which equals 1 if $\Delta$ is true and 0 otherwise.

### 2.1. Contextual Stochastic Block Model (CSBM)

We consider a CSBM with a balanced setting where the $n$ nodes are divided into two classes of approximately equal size. Let $\epsilon_1, \epsilon_2, \ldots, \epsilon_n \sim \text{Bern}(1/2)$ be $n$ independent Bernoulli random variables, and the class assignment is given by $C_k = \{j \in [n] \mid \epsilon_j = k\}$, where $k \in \{0, 1\}$. For a pair of nodes $i, j$ in the same class, they are connected with probability $p$, i.e., $\mathbf{A}_{ij} \sim \text{Bern}(p)$; for a pair of nodes $i, j$ in different classes, they are connected with probability $q$, i.e., $\mathbf{A}_{ij} \sim \text{Bern}(q)$. For simplicity, we assume node features are one-dimensional (i.e., $d = 1$), with $X \in \mathbb{R}^n$ representing the node feature vector of all $n$ nodes and $X_i$ denoting the feature of node $i$. We employ a one-dimensional GMM with parameters $(\mu, \sigma)$ to generate the feature of each node as $X_i \sim N((2\epsilon_i - 1)\mu, \sigma^2)$, and we assume $\mu > 0$. Let

$(\mathbf{A}, X) \sim \text{CSBM}(p, q, \mu, \sigma)$ denote the featured graph sampled from the above CSBM.

### 2.2. Graph Convolution and Graph Attention Mechanism

The following provides an overview of graph convolution operations and graph attention mechanisms in their general form. We then detail the multi-layer GAT for CSBMs, where each layer consists of a simplified graph convolution layer combined with an attention mechanism.

**Graph convolution operation:** For a node $i \in [n]$ with feature $\mathbf{X}_i \in \mathbb{R}^d$, the output feature $\mathbf{X}'_i$ after one layer of graph convolution is given by: $\mathbf{X}'_i = \alpha\left(\sum_{j \in [n]} \mathbf{A}_{ij} d_{ij} \mathbf{\Theta} \mathbf{X}_j\right)$, where $d_{ij} \triangleq (\sum_{l \in [n]} \mathbf{A}_{il})^{-1}$. Here, $\mathbf{\Theta} \in \mathbb{R}^{d' \times d}$ is a learnable matrix, and $\alpha(\cdot)$ represents a non-linear activation function.

**Graph attention mechanism:** Graph attention mechanism enables nodes in a graph to focus on relevant edges when aggregating information, based on the similarity between node features. Assuming an edge connects two nodes $i$ and $j$, and $\mathbf{X}_i$ and $\mathbf{X}_j$ are the features of these two nodes, the attention mechanism is defined as: $\Psi(\mathbf{X}_i, \mathbf{X}_j) \triangleq f(\mathbf{W}\mathbf{X}_i, \mathbf{W}\mathbf{X}_j)$, where $f : \mathbb{R}^{d'} \times \mathbb{R}^{d'} \to \mathbb{R}$ and $\mathbf{W} \in \mathbb{R}^{d' \times d}$ is another learnable matrix.

For any node $i$, let $\mathcal{N}_i$ be the set of neighbors of node $i$. Then, the attention coefficient $c_{ij}$ for a node $i$ and its neighbor $j \in \mathcal{N}_i$ is calculated using a softmax function $c_{ij} \triangleq \frac{\exp(\Psi(\mathbf{X}_i, \mathbf{X}_j))}{\sum_{k \in \mathcal{N}_i} \exp(\Psi(\mathbf{X}_i, \mathbf{X}_k))}$. By substituting $c_{ij}$ for $d_{ij}$, the output after one layer of attention-based graph convolution is given by $\mathbf{X}'_i = \alpha\left(\sum_{j \in [n]} \mathbf{A}_{ij} c_{ij} \mathbf{W} \mathbf{X}_j\right).$[3]

**Generalization to multi-layer GAT in the CSBM:** The previous discussion explained the standard operation of each GAT layer. However, we make some adjustments for the CSBM-generated data. First, recall from Section 2.1 that we assume each node feature $X_i \in \mathbb{R}$ is one-dimensional, thus the learnable matrices $\mathbf{\Theta}$ and $\mathbf{W}$ are unnecessary. Additionally, to simplify our analysis, the non-linear activation function $\alpha(\cdot)$ is applied only to the last layer of the multi-layer GAT. Consequently, the output of each GAT layer is $X'_i = \sum_{j \in [n]} \mathbf{A}_{ij} c_{ij} X_j$.

For a multi-layer GAT with $L \geq 1$ layers, the output feature

---

[3]Note that a graph attention mechanism may consist of multiple layers of neural networks. This paper adopts a standardized definition of a GAT layer, as presented here, regardless of the specific attention mechanism employed, to ensure clarity. This definition indicates that each layer in the GAT involves a graph convolution operation that integrates the graph attention mechanism.

of node $i$ at the $l$-th layer is given by

$$X_i^l = \sum_{j \in [n]} \mathbf{A}_{ij} c_{ij}^{l-1} X_j^{l-1}, \text{ and } X_i^{\text{out}} = \text{sgn}\left(X_i^L\right), \quad (1)$$

where $X_i^{l-1}$ is the output feature of node $i$ in the $(l-1)$-th layer, and $\{c_{ij}^{l-1}\}_{j \in \mathcal{N}_i}$ are the attention coefficients of its neighbors derived from the features of the $(l-1)$-th layers. Here, $X_i^{\text{out}}$ is the final output of this GAT, i.e., the classification result for node $i$.

**Remark 1.** *In a multi-layer GAT, neighbor coefficients vary across layers and depend on the node features of each specific layer, unlike GCNs that merely average neighbor information. Note that Eqn. 1 illustrates the single-head attention setting, which is the main focus of this paper.*

### 2.3. Perfect Node Classification

This paper considers the node classification problem for CSBMs using multi-layer GATs, with *perfect node classification* serving as the evaluation metric. This metric is equivalent to *exact recovery* (Abbe et al., 2015) in the community detection literature.

**Definition 1** (Perfect node classification)**.** *Suppose we have a GAT with $L$ layers. For a given node $i$, we say that the GAT **correctly** classifies this node if its output $X_i^L$ satisfies $X_i^L = 1$ when $i \in C_1$, and $X_i^L = -1$ when $i \in C_0$. We say this GAT achieves **perfect node classification** if it correctly classifies all nodes simultaneously with probability at least $1 - o(1)$.*

## 3. Main Results

This section presents a number of results derived in this paper. We begin by introducing the graph attention mechanism used and analyzed in our work (Section 3.1). In Section 3.2, we investigate the conditions under which the graph attention mechanism proves effective on node classification task. Next, we delve into the influence of the graph attention mechanism on the over-smoothing issue in Section 3.3. Following our analysis, we assess the enhancements that a well-designed multi-layer GAT can bring to the node classification task compared to a single-layer GAT (see Section 3.4).

Before diving into the main text, we first define signal-to-noise ratio (SNR), structure noise $\mathcal{S}_{\text{noise}}$, and feature noise $\mathcal{F}_{\text{noise}}$, as these concepts are essential for the subsequent analysis:

$$\text{SNR} \triangleq \frac{\mu}{\sigma}, \ \mathcal{S}_{\text{noise}} \triangleq \frac{p+q}{p-q}, \ \mathcal{F}_{\text{noise}} \triangleq \text{SNR}^{-1}. \quad (2)$$

Following Fountoulakis et al. (2023), we introduce the following assumption to focus on homophilic, reasonably

dense graphs that cover many practical graph data. The assumption is primarily motivated by the requirements of the proof technique. For sparser graphs, alternative proof techniques would be required.

**Assumption 1.** $p, q = \Omega(\log^2 n/n)$ and $p > q$.

### 3.1. A Simple Non-linear Graph Attention Mechanism and Its Performance

In this section, we first present a graph attention mechanism inspired by Fountoulakis et al. (2023) and then demonstrate that its performance in node classification is comparable to that of the mechanism described in (Fountoulakis et al., 2023), within a single-layer GAT setting.

In the homophilic CSBMs, edges between nodes in the same class, referred to as *intra-class* edges, should receive higher weights, while edges between nodes in different classes, referred to as *inter-class* edges, should receive lower weights. Therefore, the goal of incorporating graph attention mechanisms in CSBMs is to more effectively distinguish between intra-class and inter-class edges. Fountoulakis et al. (2023) framed this as an "XOR" problem and addressed it using a two-layer neural network. A detailed description of their attention mechanism is provided in Appendix B. However, their approach is computationally complex and challenging to analyze, particularly for multi-layer GATs. Therefore, we propose a simpler non-linear function to approximate the attention mechanism from (Fountoulakis et al., 2023), as detailed below.

**Proposed graph attention mechanism:** For a node $i$ and its neighbor $j$, with $X_i$ and $X_j$ representing their respective features, the graph attention mechanism used in this paper is defined as

$$\Psi(X_i, X_j) \triangleq \begin{cases} t & \text{if } X_i \cdot X_j \geq 0, \\ -t & \text{if } X_i \cdot X_j < 0, \end{cases} \quad (3)$$

where $t > 0$ is referred to as the *attention intensity*.

Next, we compare the performance of the two attention mechanisms described above, using perfect node classification as the evaluation metric and focusing on the single-layer GAT scenario.

**Perfect Node Classification for Single-Layer GAT:** Section 3 of (Fountoulakis et al., 2023) demonstrates that the graph attention mechanism proposed in their work can achieve perfect node classification when SNR $= \omega(\sqrt{\log n})$, which is referred to as the "easy regime". In this study, we are also interested in the influence of SNR on node classification when employing our designed attention mechanism in Eqn. 3. Pleasingly, we prove that in the aforementioned "easy regime", a single-layer GAT equipped with the attention mechanism in Eqn. 3 is equally capable for perfect node classification. This implies that our

designed attention mechanism is as efficient as those introduced in (Fountoulakis et al., 2023). The aforementioned result is summarized in Theorem 1 below.

**Theorem 1.** *For a featured graph* $(\mathbf{A}, X) \sim$ CSBM$(p, q, \mu, \sigma)$, *suppose that* SNR $= \omega(\sqrt{\log n})$ *and that Assumption 1 is satisfied. Then, employing the graph attention mechanism in Eqn. 3, a single-layer GAT, as specified in Eqn. 1 with $L = 1$, is capable of achieving perfect node classification (i.e., perfectly classifying all nodes with probability at least $1 - o(1)$).*

### 3.2. When Does Graph Attention Mechanism Help Node Classification?

The previous subsection shows that node classification performance is inherently linked to the SNR, while in this subsection we investigate the conditions under which GAT layers can enhance the SNR and when they fail to do so. Two type of noises, $\mathcal{S}_{\text{noise}}$ and $\mathcal{F}_{\text{noise}}$ (as defined in Eqn. 2), are considered. Note that $\mathcal{S}_{\text{noise}}$ increases as $p$ and $q$ get closer, making the graph less informative. As $\mathcal{F}_{\text{noise}}$ increases, the SNR decreases, resulting in less informative node features. The key implications from our findings is that when $\mathcal{S}_{\text{noise}}$ exceeds $\mathcal{F}_{\text{noise}}$, the graph attention mechanism is effective, with higher attention intensity $t$ yielding better performance. Conversely, when $\mathcal{F}_{\text{noise}}$ predominates and $\mathcal{S}_{\text{noise}}$ is relatively low, the graph attention mechanism is less effective, and a high attention intensity may even be detrimental.

Since the SNR is correlated with the expectations and variances of the node features, below we first present the *changes* in the expectations and variances of the node features after a GAT layer (Theorem 2). Before introducing the theorem, we first define $\mathcal{N}_i^p$ as the set of neighbors of node $i$ that are in the same class as node $i$, and $\mathcal{N}_i^q$ as the set of neighbors from the different class.

**Theorem 2.** *For any node* $i \in C_{\epsilon_i}$ *where* $X_i \sim N((2\epsilon_i - 1)\mu, \sigma^2)$, *let* $X_i'$ *represent the node feature after a single GAT layer, with* $\mathbb{E}[X_i']$ *denoting the* **expectation** *of* $X_i'$ *and* $\text{Var}(X_i')$ *denoting the* **variance**. *Then, there exist two computable functions* $F(\cdot)$ *and* $\widehat{F}(\cdot)$ *such that as $n$ tends to infinity, with probability at least $1 - o(1)$, we have*

- $\lim_{n \to +\infty} \frac{\mathbb{E}[X_i']}{(2\epsilon_i - 1)\mu'} = 1$, *where* $\mu' \triangleq F(\mu, \sigma, t, |\mathcal{N}_i^p|, |\mathcal{N}_i^q|)$,
- $\lim_{n \to +\infty} \frac{\text{Var}(X_i')}{(\sigma')^2} = 1$, *where* $(\sigma')^2 \triangleq \widehat{F}(\mu, \sigma, t, |\mathcal{N}_i^p|, |\mathcal{N}_i^q|)$.

*The detailed expressions of $F(\cdot)$ and $\widehat{F}(\cdot)$ are provided in Appendix C.*

It is important to highlight that, unlike simple graph convolutions, graph attention mechanisms perform non-linear operations on node features. As a result, the output node features no longer follows a simple Gaussian distribution,

making the analysis non-trivial. To tackle this challenge, we conduct a case-by-case examination of the non-linear attention mechanism, calculating expectations and variances for each scenario and aggregating the results (see Appendices E and F). The key to these calculations lies in the higher-order moments of the truncated Gaussian distribution (see Lemma 4). Additionally, during the simplification process, we were pleasantly surprised to find two seemingly different pairs of sequences whose sums converge to the same limit. We provide a proof for this observation, which led to the final expression (see Lemmas 5 and 6).

The following corollary specializes Theorem 1 to several specific parameter regimes.

**Corollary 1.** *For the expectation and variance of $X_i'$ in Theorem 2, the following statements hold,*
- *If $t = 0$, then $\mu' = \frac{p-q}{p+q}\mu$ and $(\sigma')^2 = \frac{1}{n(p+q)}\sigma^2$.*
- *If* SNR$= \omega(\sqrt{\log n})$, *then $\mu' = \frac{pe^t - qe^{-t}}{pe^t + qe^{-t}}\mu$ and $(\sigma')^2 = \frac{1}{n(p+q)}\sigma^2$.*
- *If* SNR$= o(1)$ *and $t = O(1)$, then $\mu' = \Theta\left(\frac{p-q}{p+q}\mu\right)$ and $(\sigma')^2 = \Theta\left(\left((e^t - e^{-t})^2 + \frac{1}{n(p+q)}\right)\sigma^2\right)$.*

**Remark 2.** *In Corollary 1, when $t = 0$, the GAT layer reduces to a simple graph convolution layer. In this case, our conclusions on expectation and variance align with the results in (Wu et al., 2022b).*

#### 3.2.1. DISCUSSIONS

Having obtained the expectation and variance (i.e., $\mu'$ and $\sigma'$) after a GAT layer, we will now discuss the effectiveness of the graph attention mechanism in two distinct cases. Notably, our goal is to increase the SNR (i.e., increase $\mu'/\sigma'$ compared to $\mu/\sigma$) after applying the GAT layer, as this enhances node classification performance, which serves as the criterion for evaluating the efficacy of the graph attention mechanism.

**Graph attention mechanism helps when:** $\mathcal{S}_{\text{noise}} = \omega(1)$ and $\mathcal{F}_{\text{noise}} = o(\frac{1}{\sqrt{\log n}})$.

In this case, where structure noise is high and feature noise is low, based on Corollary 1, we obtain

$$\frac{\mu'}{\sigma'} = \sqrt{n} \cdot \delta(t) \cdot \frac{\mu}{\sigma}, \quad \text{where } \delta(t) \triangleq \sqrt{\frac{(pe^t - qe^{-t})^2}{pe^{2t} + qe^{-2t}}}. \quad (4)$$

Note that $\delta(t)$ has a unique inflection point at $t = \frac{1}{2}\log\frac{q}{p} < 0$ and is monotonically increasing in the interval $t > 0$. Thus, the graph attention mechanism proves effective, with the improvement in the SNR becoming more pronounced as the attention strength $t$ increases. When the attention strength is sufficiently large, the SNR can be enhanced by up to $\mu'/\sigma' = \sqrt{np} \cdot \mu/\sigma$.

**Graph attention mechanism does not help when:** $\mathcal{S}_{\text{noise}} = O(1)$ and $\mathcal{F}_{\text{noise}} = \omega(1)$.

Now we consider the case where feature noise is high and structure noise is low. It follows from Corollary 1 that

$$\frac{\mu'}{\sigma'} = \Theta\left(\frac{p-q}{p+q} \cdot \left(c_1 \cdot (e^t - e^{-t})^2 + c_2 \cdot \frac{1}{n(p+q)}\right)^{-\frac{1}{2}}\right) \cdot \frac{\mu}{\sigma}. \tag{5}$$

For the above expression, it is clear that as $t$ increases, $\mu'/\sigma'$ decreases. Furthermore, we observe that if $t$ is not infinitesimal, meaning $(e^t - e^{-t})^2$ is constant, then passing through such a GAT layer does not necessarily guarantee an increase in the SNR. This implies that the GAT layer may not serve a useful purpose. Therefore, when feature noise predominates, using the attention mechanism can be counterproductive. In this case, simple graph convolution (with $t = 0$) performs better, yielding an improvement in SNR of $\mu'/\sigma' = \Theta(\sqrt{n(p+q)}) \cdot \mu/\sigma$.

**Remark 3.** *Note that the previous discussion does not cover all possible parameter regimes of $\mathcal{F}_{\text{noise}}$ and $\mathcal{S}_{\text{noise}}$, and below we present our comments or conjectures for the remaining regimes. When $\mathcal{S}_{\text{noise}} = \omega(1)$ and $\mathcal{F}_{\text{noise}} = \Omega(\frac{1}{\sqrt{\log n}})$, both structure and feature noise are strong, meaning the feature graph contains very little information. In such a scenario, no method is likely to perform well in node classification, making the discussion of the attention mechanism meaningless. When $\mathcal{S}_{\text{noise}} = O(1)$ and $\frac{1}{\sqrt{\log n}} \ll \mathcal{F}_{\text{noise}} \ll 1$, we conjecture that the graph attention mechanism may have some effect, but a smaller value of $t$ would be required. When $\mathcal{S}_{\text{noise}} = O(1)$ and $\mathcal{F}_{\text{noise}} = o(\frac{1}{\sqrt{\log n}})$, both structure and feature noise are minimal, leading to strong performance from both GCN and GAT, with little additional benefit from the graph attention mechanism.*

To summarize, our theoretical analysis indicates that the graph attention mechanism is not always effective for node classification tasks. When $\mathcal{F}_{\text{noise}}$ is high and $\mathcal{S}_{\text{noise}}$ is low, it performs worse than simple graph convolutions. This occurs because graph convolution leverages structure information for message passing, whereas the graph attention mechanism assigns edge weights based on feature similarity. Under these conditions, GAT's weights become unreliable and may introduce additional noise. This finding complements the results in (Fountoulakis et al., 2023), which highlighted the benefits of graph attention in reducing structure noise. Furthermore, carefully timing the application of graph attention can enhance SNR in both scenarios.

### 3.3. How Does Graph Attention Mechanism Affect Over-smoothing?

We begin by introducing a formal definition of over-smoothing, based on the definition in (Rusch et al., 2023) with some improvements. Our improvement stems from a

consensus regarding the issue of over-smoothing, namely, that over-smoothing tends to occur in shallow layers relative to the number of nodes in the graph (Yang et al., 2020; Wu et al., 2022b). To facilitate our analysis, we consider the scenario where the number of nodes $n$ approaches infinity, and assume that the number of layers $L$ in the GNN is $O(n)$. The refined definition of oversmoothing is as follows

**Definition 2** (Over-smoothing). *For an undirected featured graph $\mathcal{G}$ with $\mathbf{A}$ being the adjacency matrix and $X$ being the the features of all nodes, we say $\gamma : \mathbb{R}^n \to \mathbb{R}_{\geq 0}$ is a **node-similarity measure** if it satisfies the following axioms:*
*• $\exists\, c \in \mathbb{R}$ such that $X_i = c$ for all $i \in [n]$ if and only if $\gamma(X) = 0$, for $X \in \mathbb{R}^n$;*
*• $\gamma(X + Y) \leq \gamma(X) + \gamma(Y)$, for all $X, Y \in \mathbb{R}^n$.*

*We denote the output node features after $l$ layers as $X^{(l)}$. For a GNN with $L$ layers (where $L = O(n)$), we define over-smoothing to occur if there exist constants $C_1, C_2 > 0$ such that for all $l \in [L]$: $\gamma(X^{(l)}) \leq C_1 e^{-C_2 l} \gamma(X^{(0)})$.*

In this paper, we employ a node-similarity measure function similar to (Wu et al., 2024), which has been proved to satisfy the above axioms and takes the form

$$\gamma(X) \triangleq \frac{1}{\sqrt{n}} \|X - \frac{\mathbf{1} \cdot \mathbf{1}^T}{n} X\|_F. \tag{6}$$

The difference from (Wu et al., 2024) is that the function $\gamma$ we use incorporates normalization; however, this does not prevent it from serving as a node-similarity measure.

The above definition of over-smoothing is a general one applicable to any featured graph. Within the CSBM, it becomes apparent that over-smoothing is related to the model's parameters, particularly the model's expectation and variance. The following lemma describes their relationship.

**Lemma 1.** *For a featured graph $(\mathbf{A}, X) \sim CSBM(p, q, \mu, \sigma)$, as $n$ approaches infinity, with probability at least $1 - o(1)$, the node-similarity measure in Eqn. 6 satisfies:* $\lim_{n \to +\infty} \frac{\gamma(X)}{\sqrt{\mu^2 + \sigma^2}} = 1$.

We focus on cases with low feature noise, i.e., SNR $= \omega(\sqrt{\log n})$, as the previous section concluded that when feature noise is high, the attention mechanism offers no improvement for node classification. Therefore, discussing over-smoothing in such cases is irrelevant.

The following theorem demonstrates that when SNR is sufficiently high, GCN suffers from over-smoothing, while the graph attention mechanism can resolve the over-smoothing problem.

**Theorem 3.** *Assume that* SNR$= \omega(\sqrt{\log n})$. *Based on Definition 2, the graph convolutional networks suffer from over-smoothing. However, when $t = \omega(\sqrt{\log n})$, networks with graph attention mechanisms can prevent this over-smoothing phenomenon.*

To prove Theorem 3, we begin by analyzing how the expectations of node features evolve through multiple layers of GCN or GAT. Subsequently, we use Lemma 1 to assess how the node-similarity measure function changes in these two network architectures, allowing us to determine whether over-smoothing occurs. Specifically, for an $L$-layer GCN, we show that $\gamma(X^{(l)}) = (1 - \frac{2q}{p+q})^l \gamma(X^{(0)})$ holds for every $l \in [L]$, indicating that over-smoothing occurs. In contrast, for an $L$-layer GAT with $L = O(n)$ and $t = \omega(\sqrt{\log n})$, we demonstrate that $\gamma(X^{(l)}) = (1 - \frac{2q}{pe^{2t}+q})^l \gamma(X^{(0)}) = \Theta(\gamma(X^{(0)}))$ holds for every $l \in [L]$, thereby resolving the over-smoothing problem according to Definition 2. The detailed proof is provided in Appendix H. A synthetic experiment is presented in Section 4.1, and the results (see Figure 1c) support this theoretical result.

### 3.4. Perfect Node Classification in Multi-layer GATs

Based on the preceding discussion, we have identified scenarios where the graph attention mechanism enhances node classification and mitigates the over-smoothing issue. Leveraging these insights, we can strategically design more effective multi-layer GATs for node classification tasks, i.e., using our proposed graph attention mechanism with different values of $t$ for different layers. Furthermore, we show that the well-designed multi-layer GATs can significantly relax the "easy regime" conditions required by single-layer GATs to achieve perfect node classification (Theorem 1).

**Theorem 4.** *For a featured graph* $(\mathbf{A}, X) \sim CSBM(p, q, \mu, \sigma)$, *suppose* $p = \frac{a \log^2 n}{n}$ *and* $q = \frac{b \log^2 n}{n}$ *where* $a > b > 0$ *are positive constants[4]. When* $SNR = \omega\left(\frac{\sqrt{\log n}}{\sqrt[3]{n}}\right)$, *there exists a multi-layer GAT capable of achieving perfect node classification.*

By comparing Theorem 4 with Theorem 1, we find that the multi-layer GATs can significantly expand the conditions for achieving perfect node classification from $SNR = \omega(\sqrt{\log n})$ to $SNR = \omega(\sqrt{\log n}/\sqrt[3]{n})$ when the structure noise is not excessively high. This represents a considerable advancement, indicating that while previously an infinitely large SNR used to be required for perfect classification, now even an infinitely small SNR suffices. This underscores the superior noise tolerance of multi-layer GATs compared to single-layer GATs.

In our proof, we employ a hybrid network combining GCN and GAT layers (introduced in Appendix J). Specifically, for layers where the input SNR is less than $\sqrt{\log n}$, we utilize graph convolution layers without the attention mechanism (i.e., setting $t = 0$). As the SNR increases beyond $\sqrt{\log n}$ after multiple layers of graph convolution, we switch to graph attention layers with higher values of $t$. This design

ensures that each layer effectively enhances the SNR while preventing the over-smoothing problem.

Importantly, although this approach is tailored for the CSBM for theoretical convenience, it also offers practical insights for GAT design in real-world applications. In scenarios with substantial feature noise, one can initially set a low intensity for the graph attention mechanism to fully leverage structure information. As the network depth increases, the intensity of the attention mechanism can be gradually increased to prevent premature over-smoothing.

## 4. Experiments

In this section, we perform extensive experiments on both synthetic and real-world datasets to validate the theorems and findings of this paper. The synthetic datasets are created using CSBMs, while the real-world datasets include the widely used `Citeseer`, `Cora`, and `Pubmed`, utilizing the default train-test splits provided by PyTorch Geometric (Fey & Lenssen, 2019). The characteristics of the real-world datasets are provided in Table 2 in Appendix K. All experiments are conducted on a machine equipped with an Intel(R) Xeon(R) Silver 4215R CPU @ 3.20GHz, 64GB RAM, and an NVIDIA GeForce RTX 3090.

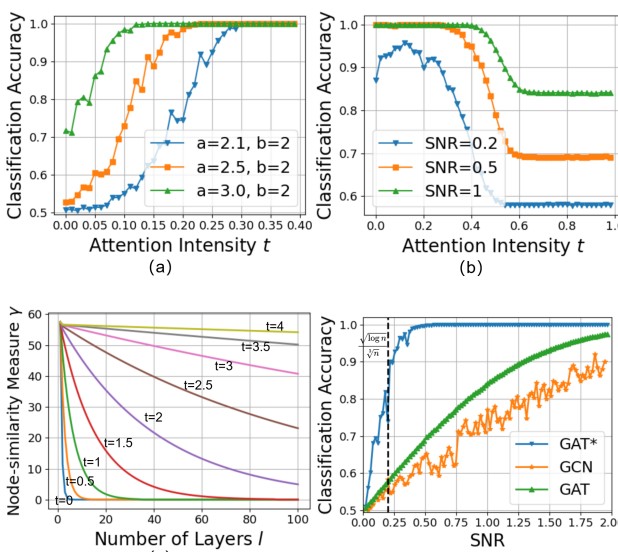

*Figure 1.* Results of the four experiments conducted on synthetic datasets. Here, Figure 1a shows the results of node classification with high $\mathcal{S}_{noise}$ and low $\mathcal{F}_{noise}$; Figure 1b presents the results for node classification with high $\mathcal{F}_{noise}$ and low $\mathcal{S}_{noise}$; Figure 1c shows the results of the over-smoothing experiment; and Figure 1d illustrates node classification results across three different networks.

### 4.1. Synthetic datasets

We conduct four experiments on synthetic datasets. Experiments 1 and 2 are designed to validate the conclusions from Section 3.2.1 on the conditions under which the graph attention mechanism is effective. Experiments 3 and 4 are aimed

---

[4]Here, we adopt a slightly stricter assumption than Assumption 1 to ensure that the structure noise is not excessively large.

at confirming Theorems 3 and 4, respectively. In all experiments, the CSBMs used to generate the data share some identical settings: $n = 3000$, $\sigma = 10$, $p = \frac{a \log^2 n}{n}$, and $q = \frac{b \log^2 n}{n}$, where $a$ and $b$ are positive constants. For Experiments 1, 2, and 4, classification accuracy is used as the evaluation metric, defined as $\sum_{i \in [n]} \mathbb{1}\{X_i^L = 2\epsilon_i - 1\}/n$. All results are averaged over 100 trials.

For Experiment 1, we investigate the effectiveness of the graph attention mechanism in a high $\mathcal{S}_{\text{noise}}$ and low $\mathcal{F}_{\text{noise}}$ scenario. We use a four-layer GAT as specified in Eqn. 1 with the attention mechanism defined in Eqn. 3, setting the attention intensity to $t$. We fix $\mu = 2\sigma\sqrt{\log n}$ and $b = 2$, and explore cases with $a = 2.1$, $a = 2.5$, and $a = 3$. Classification accuracy as a function of $t$ is recorded, with each data point representing the average of 100 independent trials, as shown in Figure 1a. The trends in Figure 1a indicate that the graph attention mechanism enhances classification performance under these conditions, supporting the conclusions in Section 3.2.1. Performance improvements become more pronounced with higher values of $t$ and $\mathcal{S}_{\text{noise}}$.

Experiment 2 examines a scenario with low $\mathcal{S}_{\text{noise}}$ and high $\mathcal{F}_{\text{noise}}$. We fix $a = 6$ and $b = 2$, and test three values for $\mu$: 2, 5, and 10, while recording the relationship between classification accuracy and $t$. Using a three-layer GAT with a uniform attention intensity $t$ across all layers, we find that classification accuracy decreases with increasing $t$, indicating that the graph attention mechanism becomes counterproductive. This observation corroborates the conclusions drawn in Section 3.2.1 regarding the conditions under which the graph attention mechanism fails.

Experiment 3 aims to validate Theorem 3, which suggests that the graph attention mechanism can prevent oversmoothing under certain conditions. We set $a = 2$, $b = 3$, and $u = 10$, using the similarity metric $\gamma$ from Eqn. 6 to measure node similarity. We construct a 100-layer GAT, varied $t$, and record changes in $\gamma$ after each attention layer, as shown in Figure 1c. The results show that, for small values of $t$, $\gamma$ decreases exponentially, indicating over-smoothing. As $t$ increases, the rate of decrease in $\gamma$ slows, and for sufficiently large $t$, the node similarity metric $\gamma$ approximates a linear decline rather than an exponential one. This indicates that, under the current settings, over-smoothing can be eliminated when $t$ is sufficiently large.

In Experiment 4, we compare three graph neural network models for node classification across different SNRs, setting to $a = 2$ and $b = 4$. The first model is a four-layer GCN. The second is a four-layer GAT with fixed attention intensity $t = 5$. The third model, referred to as GAT*, uses a gradually increasing attention intensity, with values of $[0, 0.5, 0.5, 5]$ across the four layers. Figure 1d shows that GAT* consistently delivers the highest classification

accuracy, especially at low SNRs, where it significantly outperforms the other models. As SNR increases, GAT's performance approaches that of GAT*, with both models surpassing GCN. The figure also highlights the line $\text{SNR} = \frac{\sqrt{\log n}}{\sqrt[3]{n}}$. When SNR exceeds approximately $\frac{2\sqrt{\log n}}{\sqrt[3]{n}}$, GAT* achieves perfect classification accuracy, thus validating Theorem 4.

## 4.2. Real-world datasets

We select three commonly used real-world datasets, `Citeseer`, `Cora` and `Pubmed`, and constructed three different models to compare their classification accuracy under varying levels of feature noise. Specifically, we build a two-layer GCN, a two-layer GAT, and a hybrid model where the first layer is a graph convolution layer and the second layer is a graph attention layer, referred to as GAT*. To control the feature noise, we added Gaussian noise with zero mean to the features of the three datasets, where the noise intensity is determined by the variance of the Gaussian distribution. The experiment tracked the classification accuracy of the three models as a function of the Gaussian noise intensity, with the results shown in Figure 2. From Figure 2, we observe that when the feature noise is small, GAT outperforms GCN. However, as the feature noise increases, GAT's performance begins to fall behind that of GCN, which is consistent with our theoretical analysis in Section 3.2.1. Furthermore, GAT* exhibits greater robustness to feature noise, maintaining high accuracy regardless of the noise strength, which also validates our theoretical results in Section 3.4.

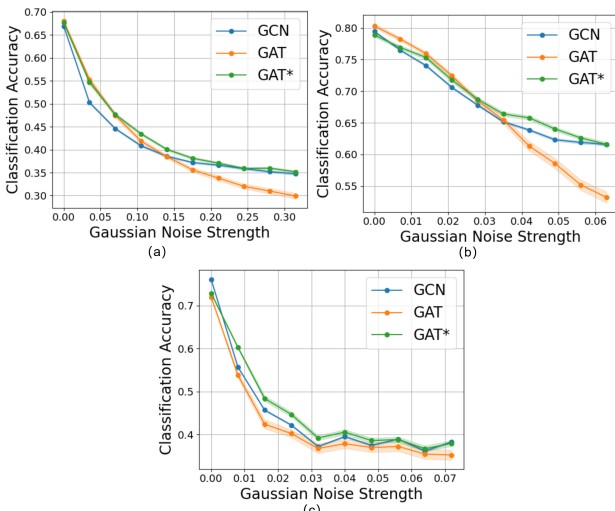

*Figure 2.* Experimental results on real-world datasets. Figures 2a, 2b and 2c illustrate the results for the `Citeseer`, `Cora` and `Pubmed` datasets, respectively.

We also conduct more extensive experiments on a larger dataset `ogbn-arxiv` (Hu et al., 2020) to evaluate the effects of the two types of noise. The results further validate

our theoretical findings. Detailed experimental results and analysis are provided in Appendix L.2.

## 5. Conclusion

This paper analyzes the graph attention mechanism using CSBM, revealing its potential failures under certain conditions. We rigorously define its effective and ineffective ranges based on structure and feature noise and explore its role in mitigating the over-smoothing problem, particularly in high SNR regime. We also propose a multi-layer GAT, establishing conditions for perfect node classification and demonstrating its superiority over single-layer GATs. Our findings provide insights for practical applications, such as selecting graph attention based on graph data characteristics and designing noise-robust networks, which we validate through experiments on real datasets.

While our analysis provides valuable theoretical insights, it has several limitations that suggest directions for future work. First, the attention mechanism we study is a simplified version that omits learnable parameters and does not incorporate multi-head attention. While this simplification enables clearer analysis, it may not fully reflect the complexity of modern attention-based GNNs. Second, we focus on multi-layer GATs but apply attention only at the final layer; incorporating attention at every layer would introduce intricate dependencies that require more advanced theoretical tools. Extending the analysis to these more general and expressive settings is an important avenue for future research.

## Impact Statement

This work provides a theoretical analysis of the graph attention mechanism, demonstrating that it is not universally effective and offering a precise mathematical characterization of its applicable range. Additionally, it presents the first theoretical sufficient conditions for exact recovery using multi-layer GATs on the CSBM model, highlighting the performance gains enabled by deep architectures. Moreover, extensive experimental validation further supports the conclusions, offering valuable insights for the design of future algorithms in the GNN field.

## Acknowledgements

This work was supported by the National Natural Science Foundation of China (Nos.U22B2036, 62261136549), the National Science Fund for Distinguished Young Scholars (Grant No.62025602), the Fundamental Research Funds for the Central Universities (Nos.G2024WD0151, D5000240309), the Tencent Foundation and XPLORER PRIZE, and Shanghai Artificial Intelligence Laboratory.

We thank our friend Yizi Wang for the helpful discussions on the proofs of certain lemmas.

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

## A. Outline of Appendices

**Outline:** In Appendix B, we provide additional details on the graph attention mechanism designed in (Fountoulakis et al., 2023) and explain how the mechanism used in this paper approximates it. Appendix C supplements definitions and a vital lemma that will be referenced throughout the proofs. Appendix D presents the proof of Theorem 1. Appendices E and F provide the proof of Theorem 2 in two parts: the expectation and variance components. Appendix G details the proof of Corollary 1, while Appendix H covers the proof of Lemma 1. Appendix I provides the proof of Theorem 3, and Appendix J presents the proof of Theorem 4. Appendix K includes additional proofs of lemmas, and Appendix L gives the results of additional experiments.

## B. Graph Attention Mechanism in Fountoulakis et al. (2023)

In the referenced work (Fountoulakis et al., 2023), the authors indicate that the edge classification problem is essentially an *"XOR problem"* and have designed a two-layer neural network architecture $\Psi$ to address this XOR issue, as detailed below,

$$\Psi(X_i, X_j) \triangleq \mathbf{r}^T \text{LeakyRelu}\left(\mathbf{S}\begin{bmatrix} X_i \\ X_j \end{bmatrix}\right), \tag{7}$$

where

$$\mathbf{S} \triangleq \begin{bmatrix} 1 & 1 \\ -1 & -1 \\ 1 & -1 \\ -1 & 1 \end{bmatrix}, \quad r \triangleq R \cdot \begin{bmatrix} 1 \\ 1 \\ -1 \\ -1 \end{bmatrix} \tag{8}$$

where $R > 0$ is the scaling parameter. Furthermore, LeakyRelu$(\cdot)$ is a non-linear activation function characterized as

$$\text{LeakyRelu}(x) = \begin{cases} x & \text{if } x \geq 0, \\ \beta x & \text{if } x < 0, \end{cases}$$

where $\beta > 0$ typically refers to a very small constant.

Substituting Eqn. 8 into Eqn. 7, we have

$$\Psi(X_i, X_j) = \begin{cases} -2R(1-\beta)X_i, & \text{if } X_j \leq -|X_i|, \\ 2R(1-\beta)\text{sgn}(X_i)X_j, & \text{if } -|X_i| < X_j < |X_i|, \\ 2R(1-\beta)X_i, & \text{if } X_j > |X_i|. \end{cases} \tag{9}$$

Then we find that when the features of the two input nodes, $X_i$ and $X_j$, have the same sign, the value of $\Psi$ is greater than $0$. Conversely, when $X_i$ and $X_j$ have opposite signs, the value of $\Psi$ is less than $0$. After applying the softmax function, edges with positive $\Psi$ values are considered intra-class edges and are assigned higher weights, while edges with negative $\Psi$ values are treated as inter-class edges and are given lower weights. Additionally, the disparity in the weights can be regulated by the scaling parameter $R$.

Motivated by the preceding insights, in this paper we abandon the neural network framework and adopt a simpler graph attention mechanism for CSBM, that is,

$$\Psi(X_i, X_j) \triangleq \begin{cases} t & \text{if } X_i \cdot X_j \geq 0, \\ -t & \text{if } X_i \cdot X_j < 0, \end{cases} \tag{10}$$

where $t > 0$ serves a similar role to $R$, which we refer to as the *attention intensity*.

Additionally, it is worth noting that the attention mechanism proposed in (Fountoulakis et al., 2023) can handle cases where the dimensionality of node features $d$ is greater than 1. In (Fountoulakis et al., 2023), when the CSBM generates node features, the following change occurs: for a node $i$, its feature $X_i$ is generated by $N((2\epsilon_i - 1)\boldsymbol{\mu}, \sigma^2 \boldsymbol{I})$, where $\boldsymbol{\mu} \in \mathbb{R}^d$, $\sigma \in \mathbb{R}$ and $\boldsymbol{I} \in \{0,1\}^{d \times d}$ is the identity matrix. Thus, for a pair of nodes $(i, j)$ and their features $\mathbf{X}_i$ and $\mathbf{X}_j$, the attention mechanism in (Fountoulakis et al., 2023) becomes

$$\Psi(\mathbf{X}_i, \mathbf{X}_j) \triangleq \mathbf{r}^T \text{LeakyRelu}\left(\mathbf{S}\begin{bmatrix} \frac{\boldsymbol{\mu}^T}{\|\boldsymbol{\mu}\|}\mathbf{X}_i \\ \frac{\boldsymbol{\mu}^T}{\|\boldsymbol{\mu}\|}\mathbf{X}_j \end{bmatrix}\right), \tag{11}$$

where $\mathbf{S}$ and $\mathbf{r}$ follow from Eqn. 8.

In this case, our proposed attention mechanism can also approximate the above-mentioned one with minor modifications, leading to the following expression:

$$\Psi(\mathbf{X}_i, \mathbf{X}_j) \triangleq \begin{cases} t & \text{if } \boldsymbol{\mu}^T\mathbf{X}_i \cdot \boldsymbol{\mu}^T\mathbf{X}_j \geq 0, \\ -t & \text{if } \boldsymbol{\mu}^T\mathbf{X}_i \cdot \boldsymbol{\mu}^T\mathbf{X}_j < 0. \end{cases} \tag{12}$$

By comparing Eqns. 11 and 12, we observe that our proposed attention mechanism eliminates two matrix multiplication operations, resulting in greater efficiency.

Note that since the node features in real datasets have $d > 1$, the attention mechanisms in Eqns. 11 and 12 are employed in experiments with real datasets in Appendix K.

## C. Preliminaries for Proofs

We begin by providing the complete expressions for functions $F(\mu, \sigma, t, |\mathcal{N}_i^p|, |\mathcal{N}_i^q|)$ and $\widehat{F}(\mu, \sigma, t, |\mathcal{N}_i^p|, |\mathcal{N}_i^q|)$, which were omitted in Theorem 2 of the main text. For simplicity, we define

$$y \triangleq \frac{\sigma}{\sqrt{2\pi}} e^{-\frac{\mu^2}{2\sigma^2}}, \ z \triangleq \Phi\left(\frac{\mu}{\sigma}\right), \ A(z,t) \triangleq e^t\left(y + \mu(1-z)\right) + e^{-t}\left(-y + \mu z\right),$$

$$B(z,t) \triangleq e^{2t}\left(\mu y + \mu^2(1-z) + \sigma^2(1-z)\right) + e^{-2t}\left(-\mu y + \mu^2 z + \sigma^2 z\right) - A^2(z,t). \tag{13}$$

Then we present that

$$F(\mu, \sigma, t, |\mathcal{N}_i^p|, |\mathcal{N}_i^q|) = S(z, t, |\mathcal{N}_i^p|, |\mathcal{N}_i^q|) \cdot T\left(z, y, t, |\mathcal{N}_i^p|, |\mathcal{N}_i^q|\right), \ \text{where} \tag{14}$$

$$S(z, t, |\mathcal{N}_i^p|, |\mathcal{N}_i^q|) \triangleq \sum_{r=0}^{|\mathcal{N}_i^p|}\sum_{s=0}^{|\mathcal{N}_i^q|} \frac{\binom{|\mathcal{N}_i^p|}{r}\binom{|\mathcal{N}_i^q|}{s}(1 - \Phi(\frac{\mu}{\sigma}))^{|\mathcal{N}_i^q|-s+r} \cdot \Phi^{|\mathcal{N}_i^p|+s-r}(\frac{\mu}{\sigma})}{(r+s)e^t + (|\mathcal{N}_i| - r - s)e^{-t}},$$

$$T\left(z, y, t, |\mathcal{N}_i^p|, |\mathcal{N}_i^q|\right)$$
$$\triangleq |\mathcal{N}_i^p| \cdot \left((1-z)A(z,t) + zA(z,-t)\right) - |\mathcal{N}_i^q| \cdot \left((1-z)A(z,-t) + zA(z,t)\right); \tag{15}$$

and

$$\widehat{F}(\mu, \sigma, t, |\mathcal{N}_i^p|, |\mathcal{N}_i^q|) = \widehat{S}(z, t, |\mathcal{N}_i^p|, |\mathcal{N}_i^q|) \cdot \widehat{T}\left(z, y, t, |\mathcal{N}_i^p|, |\mathcal{N}_i^q|\right), \ \text{where} \tag{16}$$

$$\widehat{S}(z, t, |\mathcal{N}_i^p|, |\mathcal{N}_i^q|) \triangleq \sum_{r=0}^{|\mathcal{N}_i^p|}\sum_{s=0}^{|\mathcal{N}_i^q|} \frac{\binom{|\mathcal{N}_i^p|}{r}\binom{|\mathcal{N}_i^q|}{s}(1 - \Phi(\frac{\mu}{\sigma}))^{|\mathcal{N}_i^q|-s+r} \cdot \Phi^{|\mathcal{N}_i^p|+s-r}(\frac{\mu}{\sigma})}{((r+s)e^t + (|\mathcal{N}_i| - r - s)e^{-t})^2},$$

$$\widehat{T}\left(z, y, t, |\mathcal{N}_i^p|, |\mathcal{N}_i^q|\right) \triangleq (|\mathcal{N}_i^p|^2 + |\mathcal{N}_i^q|^2) \cdot (e^t - e^{-t})^2 \cdot z(1-z) \cdot (2y + \mu(1-2z))^2 +$$
$$2|\mathcal{N}_i^p||\mathcal{N}_i^q| \cdot (e^t - e^{-t}) \cdot \left(-2(1-z)y + \mu z(1-2z)\right) \cdot \left((1-z)A(z,t) + zA(z,-t)\right) + \tag{17}$$
$$|\mathcal{N}_i^p| \cdot \left((1-z)B(z,t) + zB(z,-t)\right) + |\mathcal{N}_i^q| \cdot \left((1-z)B(z,-t) + zB(z,t)\right).$$

Then we introduce an important lemma from the referenced paper (Fountoulakis et al., 2023), which plays a key role in the proofs of several theorems. This lemma concerns a series of high-probability events, which can be proven by directly use of the Chernoff bound and the union bound. See (Fountoulakis et al., 2023) for the detailed proof.

**Lemma 2.** *Consider the following events,*

1. $\Delta_1$: $|C_0| = \frac{n}{2} \pm O(\sqrt{n \log n})$ *and* $|C_1| = \frac{n}{2} \pm O(\sqrt{n \log n})$.

2. $\Delta_2$: *for each node* $i \in [n]$, $|\mathcal{N}_i| = \frac{n(p+q)}{2}\left(1 \pm \frac{\sqrt{\log n}}{10}\right)$.

3. $\Delta_3$: *for each node $i \in [n]$, $|\mathcal{N}_i^p| = |\mathcal{N}_i| \cdot \frac{p}{p+q} \left(1 \pm \frac{\sqrt{\log n}}{10}\right)$ and $|\mathcal{N}_i^q| = |\mathcal{N}_i| \cdot \frac{q}{p+q} \left(1 \pm \frac{\sqrt{\log n}}{10}\right)$.*

4. $\Delta_4$: *for each node $i \in [n]$, $|X_i - \mathbf{E}[X_i]| \le 10\sigma\sqrt{\log n}$.*

*Suppose that Assumption 1 holds. For a featured graph $(\mathbf{A}, X)$ sampled from $CSBM(p, q, \mu, \sigma)$, the event $\Delta \triangleq \Delta_1 \cap \Delta_2 \cap \Delta_3 \cap \Delta_4$ happens with probability at least $1 - o(1)$.*

## D. Proof of Theorem 1

Without loss of generality, we first discuss a node $i$ that belongs to $C_1$. For any neighbor $j \in \mathcal{N}_i^p$, using the graph attention $\Psi$ defined in Eqn. 3, we have

$$
\begin{aligned}
P\{\Psi(X_i, X_j) = t\} &= P\{X_i \cdot X_j \ge 0\} = \left(1 - \Phi\left(\frac{\mu}{\sigma}\right)\right)^2 + \Phi^2\left(\frac{\mu}{\sigma}\right), \\
P\{\Psi(X_i, X_j) = -t\} &= P\{X_i \cdot X_j < 0\} = 2\left(1 - \Phi\left(\frac{\mu}{\sigma}\right)\right)\Phi\left(\frac{\mu}{\sigma}\right).
\end{aligned}
\tag{18}
$$

The following lemma gives a tail bound of $\Phi$.

**Lemma 3.** *Assume a random variable $y \sim N(0, 1)$, then for any constant $s > 0$, the following tail bound holds,*

$$
P\{y \ge s\} = \Phi(s) \le \min\left\{\frac{1}{2}e^{-\frac{s^2}{2}}, \frac{1}{s\sqrt{2\pi}}e^{-\frac{s^2}{2}}\right\}.
\tag{19}
$$

*Proof.* See Appendix K for the detailed proof. $\square$

Next, we illustrate the concentration of the attention coefficients in the easy regime. Consider the probability of the following event of node $i$,

$$
\begin{aligned}
P\{\forall j \in \mathcal{N}_i^p : X_i \cdot X_j \ge 0\} &= 1 - P\{\exists j \in \mathcal{N}_i^p : X_i \cdot X_j < 0\} \\
&\overset{(i)}{\ge} 1 - 2 \cdot |\mathcal{N}_i^p| \cdot \left(1 - \Phi\left(\frac{\mu}{\sigma}\right)\right)\Phi\left(\frac{\mu}{\sigma}\right) \\
&\overset{(ii)}{\ge} 1 - 2 \cdot |\mathcal{N}_i^p| \cdot \frac{1}{\omega(\sqrt{\log n})\sqrt{2\pi}} \cdot e^{-\frac{\omega(\log n)}{2}} \\
&\overset{(iii)}{\ge} 1 - 2 \cdot |\mathcal{N}_i^p| \cdot o\left(\frac{1}{n\sqrt{\log n}}\right) = 1 - o(1),
\end{aligned}
\tag{20}
$$

where $(i)$ is derived using the union bound, $(ii)$ follows from SNR$= \frac{\mu}{\sigma} = \omega(\sqrt{\log n})$ and Lemma 3, $(iii)$ is due to the fact that $|\mathcal{N}_i^p| = O(n)$.

Similarly, for the inter-class neighbors of node $i$, we have

$$
P\{\forall j \in \mathcal{N}_i^q : X_i \cdot X_j < 0\} = 1 - o(1).
\tag{21}
$$

Then, for any $j \in \mathcal{N}_i^p$, the attention coefficient $c_{ij}$, with high probability, is determined as

$$
\begin{aligned}
c_{ij} &= \frac{\exp(\Psi(X_i, X_j))}{\sum_{k \in \mathcal{N}_i} \exp(\Psi(X_i, X_k))} \\
&= \frac{\exp(\Psi(X_i, X_j))}{\sum_{k \in \mathcal{N}_i^p} \exp(\Psi(X_i, X_k)) + \sum_{k' \in \mathcal{N}_i^q} \exp(\Psi(X_i, X_{k'}))} \\
&\overset{(i)}{=} \frac{e^t}{|\mathcal{N}_i^p|e^t + |\mathcal{N}_i^q|e^{-t}}
\end{aligned}
\tag{22}
$$

where $(i)$ is due to Eqn. 20 and Eqn. 21.

Accordingly, for any $j' \in \mathcal{N}_i^q$,

$$
c_{ij'} = \frac{e^{-t}}{|\mathcal{N}_i^p|e^t + |\mathcal{N}_i^q|e^{-t}}, \text{ w.h.p..}
\tag{23}
$$

Then, after a single-layer GAT as outlined in Eqn. 1 with $L = 1$, the output of node $i$ is determined as

$$
\begin{aligned}
X_i^{\text{out}} &= \text{sgn}\Big( \sum_{j \in [n]} \mathbf{A}_{ij} c_{ij} X_j \Big) \\
&= \text{sgn}\Big( \sum_{j \in \mathcal{N}_i^p} c_{ij} X_j + \sum_{j' \in \mathcal{N}_i^q} c_{ij'} X_{j'} \Big) \\
&\overset{(i)}{\underset{\text{w.h.p.}}{=}} \text{sgn}\Big( \frac{|\mathcal{N}_i^p| e^t}{|\mathcal{N}_i^p| e^t + |\mathcal{N}_i^q| e^{-t}} \cdot (\mu \pm 10\sigma\sqrt{\log n}) + \frac{|\mathcal{N}_i^q| e^{-t}}{|\mathcal{N}_i^p| e^t + |\mathcal{N}_i^q| e^{-t}} \cdot (-\mu \pm 10\sigma\sqrt{\log n}) \Big) \\
&\overset{(ii)}{\underset{\text{w.h.p.}}{=}} \text{sgn}\Big( \frac{p e^t - q e^{-t}}{p e^t + q e^{-t}} \cdot \mu \cdot (1 \pm o(1)) \Big),
\end{aligned}
\tag{24}
$$

where $(i)$ directly follows from the high probability events $\Delta_4$ in Lemma 2 and Eqn. 22- 23, $(ii)$ is due to the high probability event $\Delta_3$ in Lemma 2 and the fact that $\mu = \omega(\sigma\sqrt{\log n})$. Notably, for a sufficienst large $t$, we have

$$
\frac{p e^t - q e^{-t}}{p e^t + q e^{-t}} = 1 - \frac{2q}{p e^{2t} + q} = 1 - o(1).
\tag{25}
$$

Thus, Eqn. 24 can be further calculated as

$$
X_i^{\text{out}} \overset{\text{w.h.p.}}{=} \text{sgn}\Big( \mu \cdot (1 \pm o(1)) \Big) = 1.
\tag{26}
$$

Likewise, for any node $i' \in C_0$, it can be proven that, with high probability, the output $X_{i'}^{\text{out}}$ equals $-1$.

# E. Proof of Theorem 2 (Expectation Part)

We first present two lemmas that play a significant role in the proofs of the expectation part of Theorem 2.

**Lemma 4.** *Assume a random variable $x \sim N(\mu, \sigma^2)$ with $f(x)$ being the probability density function of $x$, then*

$$
\begin{cases}
\int_0^{+\infty} x f(x)\, dx = \frac{\sigma}{\sqrt{2\pi}} e^{-\frac{\mu^2}{2\sigma^2}} + \mu\Big(1 - \Phi\big(\frac{\mu}{\sigma}\big)\Big), \\
\int_{-\infty}^0 x f(x)\, dx = -\frac{\sigma}{\sqrt{2\pi}} e^{-\frac{\mu^2}{2\sigma^2}} + \mu\Phi\big(\frac{\mu}{\sigma}\big),
\end{cases}
\tag{27}
$$

*and*

$$
\begin{cases}
\int_0^{+\infty} x^2 f(x)\, dx = \mu \frac{\sigma}{\sqrt{2\pi}} e^{-\frac{\mu^2}{2\sigma^2}} + \mu^2\Big(1 - \Phi\big(\frac{\mu}{\sigma}\big)\Big) + \sigma^2\Big(1 - \Phi\big(\frac{\mu}{\sigma}\big)\Big), \\
\int_{-\infty}^0 x^2 f(x)\, dx = -\mu \frac{\sigma}{\sqrt{2\pi}} e^{-\frac{\mu^2}{2\sigma^2}} + \mu^2\Phi\big(\frac{\mu}{\sigma}\big) + \sigma^2\Phi\big(\frac{\mu}{\sigma}\big).
\end{cases}
\tag{28}
$$

*Accordingly, if $x \sim N(-\mu, \sigma^2)$, then*

$$
\begin{cases}
\int_0^{+\infty} x f(x)\, dx = \frac{\sigma}{\sqrt{2\pi}} e^{-\frac{\mu^2}{2\sigma^2}} - \mu\Phi\big(\frac{\mu}{\sigma}\big), \\
\int_{-\infty}^0 x f(x)\, dx = -\frac{\sigma}{\sqrt{2\pi}} e^{-\frac{\mu^2}{2\sigma^2}} - \mu\Big(1 - \Phi\big(\frac{\mu}{\sigma}\big)\Big),
\end{cases}
\tag{29}
$$

*and*

$$
\begin{cases}
\int_0^{+\infty} x^2 f(x)\, dx = -\mu \frac{\sigma}{\sqrt{2\pi}} e^{-\frac{\mu^2}{2\sigma^2}} + \mu^2\Phi\big(\frac{\mu}{\sigma}\big) + \sigma^2\Phi\big(\frac{\mu}{\sigma}\big), \\
\int_{-\infty}^0 x^2 f(x)\, dx = \mu \frac{\sigma}{\sqrt{2\pi}} e^{-\frac{\mu^2}{2\sigma^2}} + \mu\Big(1 - \Phi\big(\frac{\mu}{\sigma}\big)\Big) + \sigma^2\Big(1 - \Phi\big(\frac{\mu}{\sigma}\big)\Big).
\end{cases}
\tag{30}
$$

*Proof.* Refer to Appendix K for the complete proof. □

**Lemma 5.** *Assume $0 < x < 1/2$, for any constants $t > 0$ and $k > 0$, let*

$$
\Gamma(n, m) \triangleq \sum_{i=0}^n \sum_{j=0}^m \frac{\binom{n}{i}\binom{m}{j}(1-x)^{m+i-j} x^{n-i+j}}{((i+j)e^t + (n+m-i-j)e^{-t})^k}.
$$

*Then the following equation holds*

$$\lim_{n,m\to+\infty} \frac{\Gamma(n,m)}{\Gamma(n+c_1, m+c_2)} = 1, \tag{31}$$

*where $c_1$ and $c_2$ are positive integer constants.*

*Proof.* See Appendix K for the full proof. □

For the expectation part of Theorem 2, without loss of generality, assume that node $i \in C_1$, then we have

$$X_i' = \sum_{j\in\mathcal{N}_i^p} \frac{X_j \cdot e^{\Psi(X_i, X_j)}}{\sum_{l\in\mathcal{N}_i} e^{\Psi(X_i, X_l)}} + \sum_{j'\in\mathcal{N}_i^q} \frac{X_{j'} \cdot e^{\Psi(X_i, X_{j'})}}{\sum_{l\in\mathcal{N}_i} e^{\Psi(X_i, X_l)}}. \tag{32}$$

And the expectation of $X_i'$ is then given by

$$\mathbb{E}[X_i'] = \mathbb{E}\Big[ \sum_{j\in\mathcal{N}_i^p} \frac{X_j \cdot e^{\Psi(X_i, X_j)}}{\sum_{l\in\mathcal{N}_i} e^{\Psi(X_i, X_l)}} + \sum_{j'\in\mathcal{N}_i^q} \frac{X_{j'} \cdot e^{\Psi(X_i, X_{j'})}}{\sum_{l\in\mathcal{N}_i} e^{\Psi(X_i, X_l)}} \Big]$$

$$\overset{(i)}{=} |\mathcal{N}_i^p| \cdot \mathbb{E}\Big[ \underbrace{\frac{X_j \cdot e^{\Psi(X_i, X_j)}}{\sum_{l\in\mathcal{N}_i} e^{\Psi(X_i, X_l)}}}_{\mathcal{A}} \Big] + |\mathcal{N}_i^q| \cdot \mathbb{E}\Big[ \underbrace{\frac{X_{j'} \cdot e^{\Psi(X_i, X_{j'})}}{\sum_{l\in\mathcal{N}_i} e^{\Psi(X_i, X_l)}}}_{\mathcal{B}} \Big], \tag{33}$$

where $(i)$ follows from the fact that each node's feature is generated independently.

Next, we calculate $\mathbb{E}[\mathcal{A}]$ and $\mathbb{E}[\mathcal{B}]$ in Eqn.33 separately.

### E.1. Calculation of $\mathbb{E}[\mathcal{A}]$

Calculating $\mathbb{E}[\mathcal{A}]$ essentially entails determining the expectation of a joint probability distribution, with the random variables of this distribution being the features of node $i$ and the features of all the neighboring nodes of $i$. Here, we denote them as $\{X_1, X_2, \ldots, X_{|\mathcal{N}_i|}\}$. Then, for every $j \in \mathcal{N}_i^p$, it follows that

$$\mathbb{E}\Big[ \frac{X_j \cdot e^{\Psi(X_i, X_j)}}{\sum_{l\in\mathcal{N}_i^p} e^{\Psi(X_i, X_l)} + \sum_{l'\in\mathcal{N}_i^q} e^{\Psi(X_i, X_{l'})}} \Big]$$

$$= \int_{X_i} \int_{X_1} \int_{X_2} \cdots \int_{X_{|\mathcal{N}_i|}} \frac{X_j \cdot e^{\Psi(X_i, X_j)}}{\sum_{l\in\mathcal{N}_i^p} e^{\Psi(X_i, X_l)} + \sum_{l'\in\mathcal{N}_i^q} e^{\Psi(X_i, X_{l'})}}$$

$$\qquad\qquad\qquad\qquad\qquad\qquad\qquad\qquad\qquad\qquad \cdot f(X_i, X_1, .., X_{|\mathcal{N}_i|})\, dX_i dX_1 dX_{|\mathcal{N}_i|} \tag{34}$$

$$\overset{(i)}{=} \int_{X_i} \int_{X_1} \cdots \int_{X_{|\mathcal{N}_i|}} \frac{X_j \cdot e^{\Psi(X_i, X_j)}}{\sum_{l\in\mathcal{N}_i^p} e^{\Psi(X_i, X_l)} + \sum_{l'\in\mathcal{N}_i^q} e^{\Psi(X_i, X_{l'})}}$$

$$\qquad\qquad\qquad\qquad\qquad\qquad\qquad\qquad\qquad \cdot f(X_i) f(X_1) .. f(X_{|\mathcal{N}_i|})\, dX_i dX_1 dX_{|\mathcal{N}_i|},$$

where $(i)$ is due to the fact that each node's feature is generated independently.

Noting that $i \in C_1$ and considering the graph attention mechanism outlined in Eqn.3, we categorize the discussions into four cases depending on the values of $X_i$ and $X_j$ being above or below zero. Thus we have

$$\mathbb{E}[\mathcal{A}]$$
$$= \mathbb{E}[\mathcal{A}|X_i > 0, X_j > 0] \cdot P\{X_i > 0, X_j > 0\} + \mathbb{E}[\mathcal{A}|X_i > 0, X_j < 0] \cdot P\{X_i > 0, X_j < 0\} \tag{35}$$
$$+ \mathbb{E}[\mathcal{A}|X_i < 0, X_j > 0] \cdot P\{X_i < 0, X_j > 0\} + \mathbb{E}[\mathcal{A}|X_i < 0, X_j < 0] \cdot P\{X_i < 0, X_j < 0\}.$$

**Case 1:** $X_i > 0,\ X_j > 0,\ \Psi(X_i, X_j) = t.$

Excluding node $j$, node $i$ has $(|\mathcal{N}_i^p| - 1)$ intra-class neighbors and $|\mathcal{N}_i^q|$ inter-class neighbors. Let $\mathcal{N}_R \triangleq \{l \in \mathcal{N}_i^p | X_l \geq 0\}$ and $\mathcal{N}_S \triangleq \{l' \in \mathcal{N}_i^q | X_{l'} \geq 0\}$. For some integers $r, s \geq 0$, we define the event $\Delta_{rs}$ as

$$\Delta_{rs} : |\mathcal{N}_R| = r \text{ and } |\mathcal{N}_S| = s. \tag{36}$$

For every $j \in \mathcal{N}_i^p$, given that $i$ is in $C_0$, it follows that $X_j \sim N(\mu, \sigma^2)$. Conversely, for every $j' \in \mathcal{N}_i^q$, $X_{j'} \sim N(-\mu, \sigma^2)$. Then we have

$$\int_0^{+\infty} f(X_j) \, dX_j = \int_{-\infty}^{0} f(X_{j'}) \, dX_{j'} = 1 - \Phi\left(\frac{\mu}{\sigma}\right),$$

$$\int_{-\infty}^{0} f(X_j) \, dX_j = \int_0^{+\infty} f(X_{j'}) \, dX_{j'} = \Phi\left(\frac{\mu}{\sigma}\right). \tag{37}$$

Hence,

$$\mathbb{E}[\mathcal{A}|X_i > 0, X_j > 0] \cdot P\{X_i > 0, X_j > 0\}$$

$$= \sum_{r=0}^{|\mathcal{N}_i^p|-1} \sum_{s=0}^{|\mathcal{N}_i^q|} \mathbb{E}[\mathcal{A}|X_i > 0, X_j > 0, \Delta_{rs}] P\{X_i > 0, X_j > 0, \Delta_{rs}\} \tag{38}$$

$$= \sum_{r=0}^{|\mathcal{N}_i^p|-1} \sum_{s=0}^{|\mathcal{N}_i^q|} \frac{\binom{|\mathcal{N}_i^p|-1}{r}\binom{|\mathcal{N}_i^q|}{s} \cdot e^t}{(r+s+1)e^t + (|\mathcal{N}_i| - r - s - 1)e^{-t}}$$

$$\cdot \underbrace{\int_0^{+\infty} \cdots \int_0^{+\infty}}_{r+1} \underbrace{\int_{-\infty}^{0} \cdots \int_{-\infty}^{0}}_{|\mathcal{N}_i^p|-r-1} f(X_i) f(X_1) \ldots f(X_{|\mathcal{N}_i^p|-1}) \, dX_i dX_1 \ldots dX_{|\mathcal{N}_i^p|-1}$$

$$\cdot \underbrace{\int_0^{+\infty} \cdots \int_0^{+\infty}}_{s} \underbrace{\int_{-\infty}^{0} \cdots \int_{-\infty}^{0}}_{|\mathcal{N}_i^q|-s} f(X_{|\mathcal{N}_i^p|+1}) \ldots f(X_{|\mathcal{N}_i|}) \, dX_{|\mathcal{N}_i^p|+1} \ldots dX_{|\mathcal{N}_i|} \cdot \int_0^{+\infty} X_j f(X_j) \, dX_j$$

$$\overset{(i)}{=} \sum_{r=0}^{|\mathcal{N}_i^p|-1} \sum_{s=0}^{|\mathcal{N}_i^q|} \frac{\binom{|\mathcal{N}_i^p|-1}{r}\binom{|\mathcal{N}_i^q|}{s} \cdot e^t}{(r+s+1)e^t + (|\mathcal{N}_i| - r - s - 1)e^{-t}}$$

$$\cdot \left(1 - \Phi\left(\frac{\mu}{\sigma}\right)\right)^{|\mathcal{N}_i^q|+r-s+1} \cdot \left(\Phi\left(\frac{\mu}{\sigma}\right)\right)^{|\mathcal{N}_i^p|-r+s-1} \cdot \int_0^{+\infty} X_j f(X_j) \, dX_j,$$

where $(i)$ is due to Eqn. 37.

Note that $X_j \sim N(\mu, \sigma^2)$, according to Lemma 4, we get that

$$\int_0^{+\infty} X_j f(X_j) \, dX_j = \frac{\sigma}{\sqrt{2\pi}} e^{-\frac{\mu^2}{2\sigma^2}} + \mu\left(1 - \Phi\left(\frac{\mu}{\sigma}\right)\right). \tag{39}$$

Hence,

$$\mathbb{E}[\mathcal{A}|X_i > 0, X_j > 0] \cdot P\{X_i > 0, X_j > 0\} = \sum_{r=0}^{|\mathcal{N}_i^p|-1} \sum_{s=0}^{|\mathcal{N}_i^q|} \frac{\binom{|\mathcal{N}_i^p|-1}{r}\binom{|\mathcal{N}_i^q|}{s} \cdot e^t}{(r+s+1)e^t + (|\mathcal{N}_i| - r - s - 1)e^{-t}}$$

$$\cdot \left(1 - \Phi\left(\frac{\mu}{\sigma}\right)\right)^{|\mathcal{N}_i^q|+r-s+1} \cdot \left(\Phi\left(\frac{\mu}{\sigma}\right)\right)^{|\mathcal{N}_i^p|-r+s-1} \cdot \left(\frac{\sigma}{\sqrt{2\pi}} e^{-\frac{\mu^2}{2\sigma^2}} + \mu\left(1 - \Phi\left(\frac{\mu}{\sigma}\right)\right)\right). \tag{40}$$

**Case 2:** $X_i > 0$, $X_j < 0$, $\Psi(X_i, X_j) = -t$.

Similar to the analysis of Case 1, we have that

$$
\mathbb{E}[\mathcal{A}|X_i > 0, X_j > 0] \cdot P\{X_i > 0, X_j > 0\}
$$

$$
= \sum_{r=0}^{|\mathcal{N}_i^p|-1} \sum_{s=0}^{|\mathcal{N}_i^q|} \mathbb{E}[\mathcal{A}|X_i > 0, X_j > 0, \Delta_{rs}] \cdot P\{X_i > 0, X_j > 0, \Delta_{rs}\}
$$

$$
= \sum_{r=0}^{|\mathcal{N}_i^p|-1} \sum_{s=0}^{|\mathcal{N}_i^q|} \frac{\binom{|\mathcal{N}_i^p|-1}{r}\binom{|\mathcal{N}_i^q|}{s} \cdot e^{-t}}{(r+s+1)e^t + (|\mathcal{N}_i|-r-s-1)e^{-t}}
$$

$$
\cdot \underbrace{\int_0^{+\infty} \int_0^{+\infty}}_{r+1} \underbrace{\int_{-\infty}^0 \int_{-\infty}^0}_{|\mathcal{N}_i^p|-r-1} f(X_i)f(X_1)\dots f(X_{|\mathcal{N}_i^p|-1}) \, dX_i dX_1 \dots dX_{|\mathcal{N}_i^p|-1}
$$

$$
\cdot \underbrace{\int_0^{+\infty} \int_0^{+\infty}}_{s} \underbrace{\int_{-\infty}^0 \int_{-\infty}^0}_{|\mathcal{N}_i^q|-s} f(X_{|\mathcal{N}_i^p|+1})\dots f(X_{|\mathcal{N}_i|}) \, dX_{|\mathcal{N}_i^p|+1} \dots dX_{|\mathcal{N}_i|} \cdot \int_{-\infty}^0 X_j f(X_j) \, dX_j
$$

$$
\overset{(i)}{=} \sum_{r=0}^{|\mathcal{N}_i^p|-1} \sum_{s=0}^{|\mathcal{N}_i^q|} \frac{\binom{|\mathcal{N}_i^p|-1}{r}\binom{|\mathcal{N}_i^q|}{s} \cdot e^{-t}}{(r+s+1)e^t + (|\mathcal{N}_i|-r-s-1)e^{-t}}
$$

$$
\cdot \left(1 - \Phi\left(\frac{\mu}{\sigma}\right)\right)^{|\mathcal{N}_i^q|+r-s+1} \cdot \left(\Phi\left(\frac{\mu}{\sigma}\right)\right)^{|\mathcal{N}_i^p|-r+s-1} \cdot \left(-\frac{\sigma}{\sqrt{2\pi}}e^{-\frac{\mu^2}{2\sigma^2}} + \mu\Phi\left(\frac{\mu}{\sigma}\right)\right),
$$

(41)

where $(i)$ is due to Lemma 4.

**Case 3:** $X_i < 0$, $X_j > 0$, $\Psi(X_i, X_j) = -t$.

In this case, we have that

$$
\mathbb{E}[\mathcal{A}|X_i < 0, X_j > 0] \cdot P\{X_i < 0, X_j > 0\}
$$

$$
= \sum_{r=0}^{|\mathcal{N}_i^p|-1} \sum_{s=0}^{|\mathcal{N}_i^q|} \mathbb{E}[\mathcal{A}|X_i < 0, X_j > 0, \Delta_{rs}] \cdot P\{X_i < 0, X_j > 0, \Delta_{rs}\}
$$

(42)

$$
= \sum_{r=0}^{|\mathcal{N}_i^p|-1} \sum_{s=0}^{|\mathcal{N}_i^q|} \frac{\binom{|\mathcal{N}_i^p|-1}{r}\binom{|\mathcal{N}_i^q|}{s} \cdot e^{-t}}{(r+s+1)e^t + (|\mathcal{N}_i|-r-s-1)e^{-t}}
$$

$$
\cdot \underbrace{\int_0^{+\infty} \int_0^{+\infty}}_{r} \underbrace{\int_{-\infty}^0 \int_{-\infty}^0}_{|\mathcal{N}_i^p|-r} f(X_i)f(X_1)\dots f(X_{|\mathcal{N}_i^p|-1}) \, dX_i dX_1 \dots dX_{|\mathcal{N}_i^p|-1}
$$

$$
\cdot \underbrace{\int_0^{+\infty} \int_0^{+\infty}}_{s} \underbrace{\int_{-\infty}^0 \int_{-\infty}^0}_{|\mathcal{N}_i^q|-s} f(X_{|\mathcal{N}_i^p|+1})\dots f(X_{|\mathcal{N}_i|}) \, dX_{|\mathcal{N}_i^p|+1} \dots dX_{|\mathcal{N}_i|} \cdot \int_0^{+\infty} X_j f(X_j) \, dX_j
$$

$$
= \sum_{r=0}^{|\mathcal{N}_i^p|-1} \sum_{s=0}^{|\mathcal{N}_i^q|} \frac{\binom{|\mathcal{N}_i^p|-1}{r}\binom{|\mathcal{N}_i^q|}{s} \cdot e^{-t}}{(r+s+1)e^t + (|\mathcal{N}_i|-r-s-1)e^{-t}}
$$

$$
\cdot \left(1 - \Phi\left(\frac{\mu}{\sigma}\right)\right)^{|\mathcal{N}_i^q|+r-s} \cdot \left(\Phi\left(\frac{\mu}{\sigma}\right)\right)^{|\mathcal{N}_i^p|-r+s} \cdot \left(\frac{\sigma}{\sqrt{2\pi}}e^{-\frac{\mu^2}{2\sigma^2}} + \mu\left(1 - \Phi\left(\frac{\mu}{\sigma}\right)\right)\right).
$$

**Case 4:** $X_i < 0$, $X_j < 0$, $\Psi(X_i, X_j) = t$.

Similarly, in this case, we get that

$$
\mathbb{E}[\mathcal{A}|X_i < 0, X_j < 0] \cdot P\{X_i < 0, X_j < 0\}
$$

$$
= \sum_{r=0}^{|\mathcal{N}_i^p|-1} \sum_{s=0}^{|\mathcal{N}_i^q|} \mathbb{E}[\mathcal{A}|X_i < 0, X_j < 0, \Delta_{rs}] \cdot P\{X_i < 0, X_j < 0, \Delta_{rs}\}
$$

$$
= \sum_{r=0}^{|\mathcal{N}_i^p|-1} \sum_{s=0}^{|\mathcal{N}_i^q|} \frac{\binom{|\mathcal{N}_i^p|-1}{r}\binom{|\mathcal{N}_i^q|}{s} \cdot e^t}{(r+s)e^t + (|\mathcal{N}_i| - r - s)e^{-t}}
$$

$$
\cdot \underbrace{\int_0^{+\infty} \int_0^{+\infty}}_{r+1} \underbrace{\int_{-\infty}^0 \int_{-\infty}^0}_{|\mathcal{N}_i^p|-r-1} f(X_i)f(X_1)\ldots f(X_{|\mathcal{N}_i^p|-1})\, dX_i dX_1 \ldots dX_{|\mathcal{N}_i^p|-1}
$$

(43)

$$
\cdot \underbrace{\int_0^{+\infty} \int_0^{+\infty}}_{s} \underbrace{\int_{-\infty}^0 \int_{-\infty}^0}_{|\mathcal{N}_i^q|-s} f(X_{|\mathcal{N}_i^p|+1})\ldots f(X_{|\mathcal{N}_i|})\, dX_{|\mathcal{N}_i^p|+1} \ldots dX_{|\mathcal{N}_i|} \cdot \int_{-\infty}^0 X_j f(X_j)\, dX_j
$$

$$
= \sum_{r=0}^{|\mathcal{N}_i^p|-1} \sum_{s=0}^{|\mathcal{N}_i^q|} \frac{\binom{|\mathcal{N}_i^p|-1}{r}\binom{|\mathcal{N}_i^q|}{s} \cdot e^t}{(r+s)e^t + (|\mathcal{N}_i| - r - s)e^{-t}}
$$

$$
\cdot \left(1 - \Phi\left(\frac{\mu}{\sigma}\right)\right)^{|\mathcal{N}_i^q|+r-s+1} \cdot \left(\Phi\left(\frac{\mu}{\sigma}\right)\right)^{|\mathcal{N}_i^p|-r+s-1} \cdot \left(-\frac{\sigma}{\sqrt{2\pi}} e^{-\frac{\mu^2}{2\sigma^2}} + \mu\Phi\left(\frac{\mu}{\sigma}\right)\right).
$$

Recall that, for the sake of brevity, the following definations are given in Eqn. 13,

$$
y \triangleq \frac{\sigma}{\sqrt{2\pi}} e^{-\frac{\mu^2}{2\sigma^2}}, \quad z \triangleq \Phi\left(\frac{\mu}{\sigma}\right), \quad A(z,t) \triangleq e^t\left(y + \mu(1-z)\right) + e^{-t}\left(-y + \mu z\right).
$$

(44)

By substituting Eqns. 40-43 into Eqn. 35, we obtain

$$
\mathbb{E}[\mathcal{A}]
$$

$$
= \mathbb{E}[\mathcal{A}|X_i > 0, X_j > 0] \cdot P\{X_i > 0, X_j > 0\} + \mathbb{E}[\mathcal{A}|X_i > 0, X_j < 0] \cdot P\{X_i > 0, X_j < 0\}
$$

$$
+ \mathbb{E}[\mathcal{A}|X_i < 0, X_j > 0] \cdot P\{X_i < 0, X_j > 0\} + \mathbb{E}[\mathcal{A}|X_i < 0, X_j < 0] \cdot P\{X_i < 0, X_j < 0\}
$$

$$
= \sum_{r=0}^{|\mathcal{N}_i^p|-1} \sum_{s=0}^{|\mathcal{N}_i^q|} \frac{\binom{|\mathcal{N}_i^p|-1}{r}\binom{|\mathcal{N}_i^q|}{s} \cdot e^t}{(r+s+1)e^t + (|\mathcal{N}_i| - r - s - 1)e^{-t}}
$$

$$
\cdot \left(1 - \Phi\left(\frac{\mu}{\sigma}\right)\right)^{|\mathcal{N}_i^q|+r-s+1} \cdot \left(\Phi\left(\frac{\mu}{\sigma}\right)\right)^{|\mathcal{N}_i^p|-r+s-1} \cdot \left(\frac{\sigma}{\sqrt{2\pi}} e^{-\frac{\mu^2}{2\sigma^2}} + \mu\left(1 - \Phi\left(\frac{\mu}{\sigma}\right)\right)\right)
$$

$$
+ \frac{\binom{|\mathcal{N}_i^p|-1}{r}\binom{|\mathcal{N}_i^q|}{s} \cdot e^{-t}}{(r+s+1)e^t + (|\mathcal{N}_i| - r - s - 1)e^{-t}}
$$

$$
\cdot \left(1 - \Phi\left(\frac{\mu}{\sigma}\right)\right)^{|\mathcal{N}_i^q|+r-s+1} \cdot \left(\Phi\left(\frac{\mu}{\sigma}\right)\right)^{|\mathcal{N}_i^p|-r+s-1} \cdot \left(-\frac{\sigma}{\sqrt{2\pi}} e^{-\frac{\mu^2}{2\sigma^2}} + \mu\Phi\left(\frac{\mu}{\sigma}\right)\right)
$$

$$
+ \frac{\binom{|\mathcal{N}_i^p|-1}{r}\binom{|\mathcal{N}_i^q|}{s} \cdot e^{-t}}{(r+s+1)e^t + (|\mathcal{N}_i| - r - s - 1)e^{-t}}
$$

$$
\cdot \left(1 - \Phi\left(\frac{\mu}{\sigma}\right)\right)^{|\mathcal{N}_i^q|+r-s} \cdot \left(\Phi\left(\frac{\mu}{\sigma}\right)\right)^{|\mathcal{N}_i^p|-r+s} \cdot \left(\frac{\sigma}{\sqrt{2\pi}} e^{-\frac{\mu^2}{2\sigma^2}} + \mu\left(1 - \Phi\left(\frac{\mu}{\sigma}\right)\right)\right)
$$

$$
+ \frac{\binom{|\mathcal{N}_i^p|-1}{r}\binom{|\mathcal{N}_i^q|}{s} \cdot e^t}{(r+s)e^t + (|\mathcal{N}_i| - r - s)e^{-t}}
$$

$$
\cdot \left(1 - \Phi\left(\frac{\mu}{\sigma}\right)\right)^{|\mathcal{N}_i^q|+r-s+1} \cdot \left(\Phi\left(\frac{\mu}{\sigma}\right)\right)^{|\mathcal{N}_i^p|-r+s-1} \cdot \left(-\frac{\sigma}{\sqrt{2\pi}} e^{-\frac{\mu^2}{2\sigma^2}} + \mu\Phi\left(\frac{\mu}{\sigma}\right)\right)
$$

$$\overset{(i)}{\underset{\text{w.h.p.}}{=}} \sum_{r=0}^{|\mathcal{N}_i^p|-1} \sum_{s=0}^{|\mathcal{N}_i^q|} \frac{\binom{|\mathcal{N}_i^p|-1}{r}\binom{|\mathcal{N}_i^q|}{s} \cdot (1-z)^{|\mathcal{N}_i^q|+r-s} \cdot z^{|\mathcal{N}_i^p|-r+s-1}}{(r+s)e^t + (|\mathcal{N}_i|-r-s)e^{-t}}$$

$$\cdot \left( (1-z) \cdot \left( e^t(y+\mu(1-z)) + e^{-t}(-y+\mu z) \right) + z \cdot \left( e^{-t}(y+\mu(1-z)) + e^t(-y+\mu z) \right) \right) \tag{45}$$

$$\overset{(ii)}{\underset{\text{w.h.p.}}{=}} \sum_{r=0}^{|\mathcal{N}_i^p|-1} \sum_{s=0}^{|\mathcal{N}_i^q|} \frac{\binom{|\mathcal{N}_i^p|-1}{r}\binom{|\mathcal{N}_i^q|}{s}(1-z)^{|\mathcal{N}_i^q|+r-s}z^{|\mathcal{N}_i^p|-r+s-1}}{(r+s)e^t + (|\mathcal{N}_i|-r-s)e^{-t}} \left( (1-z)A(z,t) + zA(z,-t) \right),$$

where $(i)$ holds since Lemma 2 ensures that $|\mathcal{N}_i| = \frac{n(p+q)}{2}\left(1 \pm \frac{\sqrt{\log n}}{10}\right) = \omega(1)$, and $(ii)$ follows from Eqn. 44.

## E.2. Calculation of $\mathbb{E}[\mathcal{B}]$

The process for calculating $\mathbb{E}[\mathcal{B}]$ is the same as for $\mathbb{E}[\mathcal{A}]$, focusing on finding the expectation of a joint probability distribution for all the features of node $i$'s neighbors. Moreover, because of the graph attention mechanism, both calculations require a discussion for when the product of $X_i$ and $X_{j'}$ is positive, involving four different cases. The main difference between calculating $\mathbb{E}[\mathcal{B}]$ and $\mathbb{E}[\mathcal{A}]$ is that $X_{j'}$ is considered an inter-class neighbor, implying it follows a different normal distribution, $X_{j'} \sim N(-\mu, \sigma^2)$. Similarly, we have that

$$\mathbb{E}[\mathcal{B}] = \mathbb{E}[\mathcal{B}|X_i > 0, X_j > 0] \cdot P\{X_i > 0, X_j > 0\} + \mathbb{E}[\mathcal{B}|X_i > 0, X_j < 0] \cdot P\{X_i > 0, X_j < 0\}$$
$$+ \mathbb{E}[\mathcal{B}|X_i < 0, X_j > 0] \cdot P\{X_i < 0, X_j > 0\} + \mathbb{E}[\mathcal{B}|X_i < 0, X_j < 0] \cdot P\{X_i < 0, X_j < 0\}. \tag{46}$$

Additionally, we continue to use the event $\Delta_{rs}$ as defined in Eqn. 36. Notably, with $j'$ being an inter-class neighbor, $r$ is constrained to a maximum of $|\mathcal{N}_i^p|$, and correspondingly, $s$ reaches its upper limit at $(|\mathcal{N}_i^q|-1)$.

Then for the case that $X_i > 0$ and $X_{j'} > 0$, we have that

$$\mathbb{E}[\mathcal{B}|X_i > 0, X_{j'} > 0] \cdot P\{X_i > 0, X_{j'} > 0\}$$

$$= \sum_{r=0}^{|\mathcal{N}_i^p|} \sum_{s=0}^{|\mathcal{N}_i^q|-1} \mathbb{E}[\mathcal{B}|X_i > 0, X_{j'} > 0, \Delta_{rs}] P\{X_i > 0, X_{j'} > 0, \Delta_{rs}\}$$

$$= \sum_{r=0}^{|\mathcal{N}_i^p|} \sum_{s=0}^{|\mathcal{N}_i^q|-1} \frac{\binom{|\mathcal{N}_i^p|}{r}\binom{|\mathcal{N}_i^q|-1}{s} \cdot e^t}{(r+s+1)e^t + (|\mathcal{N}_i|-r-s-1)e^{-t}}$$

$$\cdot \underbrace{\int_0^{+\infty} \int_0^{+\infty}}_{r+1} \underbrace{\int_{-\infty}^0 \int_{-\infty}^0}_{|\mathcal{N}_i^p|-r} f(X_i)f(X_1)\dots f(X_{|\mathcal{N}_i^p|}) \, dX_i dX_1 \dots dX_{|\mathcal{N}_i^p|}$$

$$\cdot \underbrace{\int_0^{+\infty} \int_0^{+\infty}}_{s} \underbrace{\int_{-\infty}^0 \int_{-\infty}^0}_{|\mathcal{N}_i^q|-s-1} f(X_{|\mathcal{N}_i^p|+2})\dots f(X_{|\mathcal{N}_i|}) \, dX_{|\mathcal{N}_i^p|+2} \dots dX_{|\mathcal{N}_i|} \cdot \int_0^{+\infty} X_{j'}f(X_{j'}) \, dX_{j'} \tag{47}$$

$$= \sum_{r=0}^{|\mathcal{N}_i^p|-1} \sum_{s=0}^{|\mathcal{N}_i^q|} \frac{\binom{|\mathcal{N}_i^p|-1}{r}\binom{|\mathcal{N}_i^q|}{s} \cdot e^t}{(r+s+1)e^t + (|\mathcal{N}_i|-r-s-1)e^{-t}}$$

$$\cdot \left(1 - \Phi\left(\frac{\mu}{\sigma}\right)\right)^{|\mathcal{N}_i^q|+r-s} \cdot \left(\Phi\left(\frac{\mu}{\sigma}\right)\right)^{|\mathcal{N}_i^p|-r+s} \cdot \int_0^{+\infty} X_{j'}f(X_{j'}) \, dX_{j'}$$

$$\overset{(i)}{=} \sum_{r=0}^{|\mathcal{N}_i^p|-1} \sum_{s=0}^{|\mathcal{N}_i^q|} \frac{\binom{|\mathcal{N}_i^p|-1}{r}\binom{|\mathcal{N}_i^q|}{s} \cdot e^t}{(r+s+1)e^t + (|\mathcal{N}_i|-r-s-1)e^{-t}}$$

$$\cdot \left(1 - \Phi\left(\frac{\mu}{\sigma}\right)\right)^{|\mathcal{N}_i^q|+r-s} \cdot \left(\Phi\left(\frac{\mu}{\sigma}\right)\right)^{|\mathcal{N}_i^p|-r+s} \cdot \left(\frac{\sigma}{\sqrt{2\pi}}e^{-\frac{\mu^2}{2\sigma^2}} - \mu\Phi\left(\frac{\mu}{\sigma}\right)\right),$$

where $(i)$ holds since $X_{j'} \sim N(-\mu, \sigma^2)$ and Lemma 3.

As the other three cases follow the similar approach, we directly state the final result for $\mathbb{E}[\mathcal{B}]$ as

$$
\begin{aligned}
\mathbb{E}[\mathcal{B}] &= \mathbb{E}[\mathcal{B}|X_i > 0, X_j > 0] \cdot P\{X_i > 0, X_j > 0\} + \mathbb{E}[\mathcal{B}|X_i > 0, X_j < 0] \cdot P\{X_i > 0, X_j < 0\} \\
&\quad + \mathbb{E}[\mathcal{B}|X_i < 0, X_j > 0] \cdot P\{X_i < 0, X_j > 0\} + \mathbb{E}[\mathcal{B}|X_i < 0, X_j < 0] \cdot P\{X_i < 0, X_j < 0\} \\
&= \sum_{r=0}^{|\mathcal{N}_i^p|-1} \sum_{s=0}^{|\mathcal{N}_i^q|} \frac{\binom{|\mathcal{N}_i^p|-1}{r}\binom{|\mathcal{N}_i^q|}{s} \cdot e^t}{(r+s+1)e^t + (|\mathcal{N}_i| - r - s - 1)e^{-t}} \\
&\quad \cdot \left(1 - \Phi\left(\frac{\mu}{\sigma}\right)\right)^{|\mathcal{N}_i^q|+r-s} \cdot \left(\Phi\left(\frac{\mu}{\sigma}\right)\right)^{|\mathcal{N}_i^p|-r+s} \cdot \left(\frac{\sigma}{\sqrt{2\pi}} e^{-\frac{\mu^2}{2\sigma^2}} - \mu \Phi\left(\frac{\mu}{\sigma}\right)\right) \\
&\quad + \frac{\binom{|\mathcal{N}_i^p|-1}{r}\binom{|\mathcal{N}_i^q|}{s} \cdot e^{-t}}{(r+s)e^t + (|\mathcal{N}_i| - r - s)e^{-t}} \\
&\quad \cdot \left(1 - \Phi\left(\frac{\mu}{\sigma}\right)\right)^{|\mathcal{N}_i^q|+r-s} \cdot \left(\Phi\left(\frac{\mu}{\sigma}\right)\right)^{|\mathcal{N}_i^p|-r+s} \cdot \left(-\frac{\sigma}{\sqrt{2\pi}} e^{-\frac{\mu^2}{2\sigma^2}} - \mu\left(1 - \Phi\left(\frac{\mu}{\sigma}\right)\right)\right) \\
&\quad + \frac{\binom{|\mathcal{N}_i^p|-1}{r}\binom{|\mathcal{N}_i^q|}{s} \cdot e^{-t}}{(r+s+1)e^t + (|\mathcal{N}_i| - r - s - 1)e^{-t}} \\
&\quad \cdot \left(1 - \Phi\left(\frac{\mu}{\sigma}\right)\right)^{|\mathcal{N}_i^q|+r-s-1} \left(\Phi\left(\frac{\mu}{\sigma}\right)\right)^{|\mathcal{N}_i^p|-r+s+1} \cdot \left(\frac{\sigma}{\sqrt{2\pi}} e^{-\frac{\mu^2}{2\sigma^2}} - \mu \Phi\left(\frac{\mu}{\sigma}\right)\right) \\
&\quad + \frac{\binom{|\mathcal{N}_i^p|-1}{r}\binom{|\mathcal{N}_i^q|}{s} \cdot e^t}{(r+s)e^t + (|\mathcal{N}_i| - r - s)e^{-t}} \\
&\quad \cdot \left(1 - \Phi\left(\frac{\mu}{\sigma}\right)\right)^{|\mathcal{N}_i^q|+r-s-1} \left(\Phi\left(\frac{\mu}{\sigma}\right)\right)^{|\mathcal{N}_i^p|-r+s+1} \left(-\frac{\sigma}{\sqrt{2\pi}} e^{-\frac{\mu^2}{2\sigma^2}} - \mu\left(1 - \Phi\left(\frac{\mu}{\sigma}\right)\right)\right) \\
&\overset{(i)}{\underset{\text{w.h.p.}}{=}} \sum_{r=0}^{|\mathcal{N}_i^p|-1} \sum_{s=0}^{|\mathcal{N}_i^q|} \frac{\binom{|\mathcal{N}_i^p|-1}{r}\binom{|\mathcal{N}_i^q|}{s}(1-z)^{|\mathcal{N}_i^q|+r-s-1}z^{|\mathcal{N}_i^p|-r+s}}{(r+s)e^t + (|\mathcal{N}_i| - r - s)e^{-t}}\Big((1-z)A(z,-t) + zA(z,t)\Big),
\end{aligned}
\tag{48}
$$

where $(i)$ is due to $|\mathcal{N}_i| = \omega(1)$ and Eqn. 44.

After obtaining $\mathbb{E}[\mathcal{A}]$ and $\mathbb{E}[\mathcal{B}]$, by revisiting Eqn. 33, it follows that

$$
\begin{aligned}
\mathbb{E}[X_i'] \overset{\text{w.h.p.}}{=}& \\
|\mathcal{N}_i^p| &\sum_{r=0}^{|\mathcal{N}_i^p|-1} \sum_{s=0}^{|\mathcal{N}_i^q|} \frac{\binom{|\mathcal{N}_i^p|-1}{r}\binom{|\mathcal{N}_i^q|}{s}(1-z)^{|\mathcal{N}_i^q|+r-s}z^{|\mathcal{N}_i^p|-r+s-1}}{(r+s)e^t + (|\mathcal{N}_i| - r - s)e^{-t}}\Big((1-z)A(z,t) + zA(z,-t)\Big) \\
+ |\mathcal{N}_i^q| &\sum_{r=0}^{|\mathcal{N}_i^p|} \sum_{s=0}^{|\mathcal{N}_i^q|-1} \frac{\binom{|\mathcal{N}_i^p|}{r}\binom{|\mathcal{N}_i^q|-1}{s}(1-z)^{|\mathcal{N}_i^q|+r-s-1}z^{|\mathcal{N}_i^p|-r+s}}{(r+s)e^t + (|\mathcal{N}_i| - r - s)e^{-t}}\Big((1-z)A(z,-t) + zA(z,t)\Big) \\
\overset{\text{w.h.p.}}{=}& |\mathcal{N}_i^p| \cdot S\left(z, t, |\mathcal{N}_i^p| - 1, |\mathcal{N}_i^q|\right) \cdot \Big((1-z) \cdot A(z,t) + z \cdot A(z,-t)\Big) \\
+& |\mathcal{N}_i^q| \cdot S\left(z, t, |\mathcal{N}_i^p|, |\mathcal{N}_i^q| - 1\right) \cdot \Big((1-z) \cdot A(z,-t) + z \cdot A(z,t)\Big),
\end{aligned}
\tag{49}
$$

where

$$
S\left(z, t, |\mathcal{N}_i^p|, |\mathcal{N}_i^q|\right) \triangleq \sum_{r=0}^{|\mathcal{N}_i^p|} \sum_{s=0}^{|\mathcal{N}_i^q|} \frac{\binom{|\mathcal{N}_i^p|}{r}\binom{|\mathcal{N}_i^q|}{s}(1 - \Phi\left(\frac{\mu}{\sigma}\right))^{|\mathcal{N}_i^q|-s+r} \cdot \Phi^{|\mathcal{N}_i^p|+s-r}\left(\frac{\mu}{\sigma}\right)}{(r+s)e^t + (|\mathcal{N}_i| - r - s)e^{-t}}.
$$

Notably, given that $\Phi\left(\frac{\mu}{\sigma}\right) \in (0, 1/2)$ and $t > 0$, applying Lemma 5, it follows that

$$
S\left(z, t, |\mathcal{N}_i^p| - 1, |\mathcal{N}_i^q|\right) \overset{\text{w.h.p.}}{=} S\left(z, t, |\mathcal{N}_i^p|, |\mathcal{N}_i^q| - 1\right).
\tag{50}
$$

Hence, it is sufficient to show that

$$\mathbb{E}[X_i'] \overset{\text{w.h.p.}}{=} S\left(z, t, |\mathcal{N}_i^p|, |\mathcal{N}_i^q|\right) \cdot T(z, y, t, |\mathcal{N}_i^p|, |\mathcal{N}_i^q|), \tag{51}$$

where

$$T\left(z, y, t, |\mathcal{N}_i^p|, |\mathcal{N}_i^q|\right) \triangleq |\mathcal{N}_i^p| \cdot \left((1-z)A(z, t) + zA(z, -t)\right) - |\mathcal{N}_i^q| \cdot \left((1-z)A(z, -t) + zA(z, t)\right).$$

Similarly, if node $i$ belongs to community $C_0$, by symmetry, we obtain that

$$\mathbb{E}[X_i'] \overset{\text{w.h.p.}}{=} -S\left(z, t, |\mathcal{N}_i^p|, |\mathcal{N}_i^q|\right) \cdot T(z, y, t, |\mathcal{N}_i^p|, |\mathcal{N}_i^q|). \tag{52}$$

Thus, for any node $i \in C_{\epsilon_i}$, with probability $1 - o(1)$, $\mathbb{E}[X_i']$ equals $(2\epsilon_i - 1)\mu'$, where

$$\mu' = S\left(z, t, |\mathcal{N}_i^p|, |\mathcal{N}_i^q|\right) \cdot T(z, y, t, |\mathcal{N}_i^p|, |\mathcal{N}_i^q|). \tag{53}$$

## F. Proof of Theorem 2 (Variance Part)

We first present a key lemma for proving the variance part of Theorem 2.

**Lemma 6.** *Assume* $0 < x < 1/2$*, for any constant* $t > 0$*, define* $A(n, m) \triangleq \sum_{i=0}^n \sum_{j=0}^m \frac{\binom{n}{i}\binom{m}{j}(1-x)^{m+i-j}x^{n-i+j}}{((i+j)e^t + (n+m-i-j)e^{-t})^2}$*, and* $B(n, m) \triangleq \left(\sum_{i=0}^n \sum_{j=0}^m \frac{\binom{n}{i}\binom{m}{j}(1-x)^{m+i-j}x^{n-i+j}}{(i+j)e^t + (n+m-i-j)e^{-t}}\right)^2$*. Then, for* $n + m \to +\infty$*, we have*

$$A(n, m) = \Theta((n+m)^{-2}), \; B(n, m) = \Theta((n+m)^{-2}), \; A(n, m) - B(n, m) = o((n+m)^{-3}).$$

*Proof.* We provide the detailed proof in Section K. $\qquad\square$

Without loss of generality, we assume that node $i \in C_1$. Note that

$$\text{Var}(X_i') = \mathbb{E}[(X_i')^2] - \mathbb{E}^2[X_i']. \tag{54}$$

Since we have obtained $\mathbb{E}[X_i']$ in the proof of Theorem 2, the key now is how to calculate $\mathbb{E}[(X_i')^2]$. By Eqn. 32, we have

$$
\begin{aligned}
(X_i')^2 &= \left(\sum_{j \in \mathcal{N}_i^p} \frac{X_j \cdot e^{\Psi(X_i, X_j)}}{\sum_{l \in \mathcal{N}_i} e^{\Psi(X_i, X_l)}} + \sum_{j' \in \mathcal{N}_i^q} \frac{X_{j'} \cdot e^{\Psi(X_i, X_{j'})}}{\sum_{l \in \mathcal{N}_i} e^{\Psi(X_i, X_l)}}\right)^2 \\
&= \underbrace{\left(\sum_{j \in \mathcal{N}_i^p} \frac{X_j \cdot e^{\Psi(X_i, X_j)}}{\sum_{l \in \mathcal{N}_i} e^{\Psi(X_i, X_l)}}\right)^2}_{\mathcal{A}} + \underbrace{\left(\sum_{j' \in \mathcal{N}_i^q} \frac{X_{j'} \cdot e^{\Psi(X_i, X_{j'})}}{\sum_{l \in \mathcal{N}_i} e^{\Psi(X_i, X_l)}}\right)^2}_{\mathcal{B}} \\
&\quad + \underbrace{2 \sum_{j \in \mathcal{N}_i^p} \sum_{j' \in \mathcal{N}_i^q} \frac{X_j \cdot X_{j'} \cdot e^{\Psi(X_i, X_j)} \cdot e^{\Psi(X_i, X_{j'})}}{(\sum_{l \in \mathcal{N}_i} e^{\Psi(X_i, X_l)})^2}}_{\mathcal{C}}
\end{aligned}
\tag{55}
$$

Thus, we have established that $\mathbb{E}[(X_i')^2] = \mathbb{E}[\mathcal{A}] + \mathbb{E}[\mathcal{B}] + \mathbb{E}[\mathcal{C}]$. Subsequently, we will calculate each of these three components in turn.

### F.1. Calculation of $\mathbb{E}[\mathcal{A}]$

Firstly, since the node features are generated independently, we have

$$
\begin{aligned}
\mathbb{E}[\mathcal{A}] &= E\Big[\Big(\sum_{j\in\mathcal{N}_i^p}\frac{X_j\cdot e^{\Psi(X_i,X_j)}}{\sum_{l\in\mathcal{N}_i^p}e^{\Psi(X_i,X_l)}+\sum_{l'\in\mathcal{N}_i^q}e^{\Psi(X_i,X_{l'})}}\Big)^2\Big] \\
&= (|\mathcal{N}_i^p|^2-|\mathcal{N}_i^p|)\cdot E\Big[\underbrace{\frac{X_{j_1}\cdot X_{j_2}\cdot e^{\Psi(X_i,X_{j_1})}\cdot e^{\Psi(X_i,X_{j_2})}}{(\sum_{l\in\mathcal{N}_i^p}e^{\Psi(X_i,X_l)}+\sum_{l'\in\mathcal{N}_i^q}e^{\Psi(X_i,X_{l'})})^2}}_{\mathcal{A}_1}\Big] \\
&\quad+|\mathcal{N}_i^p|\cdot E\Big[\underbrace{\frac{X_{j_1}^2\cdot e^{2\Psi(X_i,X_{j_1})}}{(\sum_{l\in\mathcal{N}_i^p}e^{\Psi(X_i,X_l)}+\sum_{l'\in\mathcal{N}_i^q}e^{\Psi(X_i,X_{l'})})^2}}_{\mathcal{A}_2}\Big],
\end{aligned}
\tag{56}
$$

where $j_1,j_2\in\mathcal{N}_i^p$. The key is to compute the expectations of $\mathcal{A}_1$ and $\mathcal{A}_2$.

#### F.1.1. CALCULATION OF $\mathbb{E}[\mathcal{A}_1]$

Given that node $i$ is in $C_1$, and using the graph attention mechanism from Eqn. 3, we break down the discussion into eight cases, each defined by the positive or negative values of $X_i$, $X_{j_1}$, and $X_{j_2}$, as shown in Table 1.

|         | Case 1 | Case 2 | Case 3 | Case 4 | Case 5 | Case 6 | Case 7 | Case 8 |
|---------|--------|--------|--------|--------|--------|--------|--------|--------|
| $X_i$    | $\geq 0$ | $\geq 0$ | $\geq 0$ | $\geq 0$ | $< 0$ | $< 0$ | $< 0$ | $< 0$ |
| $X_{j_1}$ | $\geq 0$ | $\geq 0$ | $< 0$ | $< 0$ | $\geq 0$ | $\geq 0$ | $< 0$ | $< 0$ |
| $X_{j_2}$ | $\geq 0$ | $< 0$ | $\geq 0$ | $< 0$ | $\geq 0$ | $< 0$ | $\geq 0$ | $< 0$ |

Table 1. Different cases of $X_i$, $X_{j_1}$ and $X_{j_2}$.

Hence, we have

$$
\mathbb{E}[\mathcal{A}_1] = \mathbb{E}[\mathcal{A}_1|\textbf{Case 1}]\cdot P\{\textbf{Case 1}\} + \ldots + \mathbb{E}[\mathcal{A}_1|\textbf{Case 8}]\cdot P\{\textbf{Case 8}\}.
\tag{57}
$$

**Case 1:** $X_i\geq 0$, $X_{j_1}\geq 0$, $X_{j_2}\geq 0$, $\Psi(X_i,X_{j_1})=t$, $\Psi(X_i,X_{j_2})=t$.
Using the same notion of event $\Delta_{rs}$ defined in Eqn. 36, we have

$$
\begin{aligned}
&\mathbb{E}[\mathcal{A}_1|\textbf{Case 1}]\cdot P\{\textbf{Case 1}\} = \sum_{r=0}^{|\mathcal{N}_i^p|-2}\sum_{s=0}^{|\mathcal{N}_i^q|}\mathbb{E}[\mathcal{A}_1|\Delta_{rs}]\cdot P\{\Delta_{rs}\} \\
&= \sum_{r=0}^{|\mathcal{N}_i^p|-2}\sum_{s=0}^{|\mathcal{N}_i^q|}\frac{\binom{|\mathcal{N}_i^p|-2}{r}\binom{|\mathcal{N}_i^q|}{s}\cdot e^{2t}}{((r+s+2)e^t+(|\mathcal{N}_i|-r-s-2)e^{-t})^2} \\
&\qquad\cdot\underbrace{\int_0^{+\infty}\int_0^{+\infty}}_{r+1}\underbrace{\int_{-\infty}^{0}\int_{-\infty}^{0}}_{|\mathcal{N}_i^p|-r-2}f(X_i)f(X_1)\ldots f(X_{|\mathcal{N}_i^p|-2})\,dX_idX_1\ldots dX_{|\mathcal{N}_i^p|-2} \\
&\qquad\cdot\underbrace{\int_0^{+\infty}\int_0^{+\infty}}_{s}\underbrace{\int_{-\infty}^{0}\int_{-\infty}^{0}}_{|\mathcal{N}_i^q|-s}f(X_{|\mathcal{N}_i^p|+1})\ldots f(X_{|\mathcal{N}_i|})\,dX_{|\mathcal{N}_i^p|+1}\ldots dX_{|\mathcal{N}_i|} \\
&\qquad\cdot\int_0^{+\infty}X_{j_1}f(X_{j_1})\,dX_{j_1}\cdot\int_0^{+\infty}X_{j_2}f(X_{j_2})\,dX_{j_1} \\
&\overset{(i)}{=} \sum_{r=0}^{|\mathcal{N}_i^p|-2}\sum_{s=0}^{|\mathcal{N}_i^q|}\frac{\binom{|\mathcal{N}_i^p|-2}{r}\binom{|\mathcal{N}_i^q|}{s}\cdot e^{2t}}{((r+s+2)e^t+(|\mathcal{N}_i|-r-s-2)e^{-t})^2} \\
&\qquad\cdot(1-z)^{|\mathcal{N}_i^q|+r-s+1}\cdot z^{|\mathcal{N}_i^p|-r+s-2}\cdot(y+\mu(1-z))^2,
\end{aligned}
\tag{58}
$$

where $(i)$ follows from Lemma 4, Eqn. 37 and Eqn. 13.

**Case 2:** $X_i \geq 0$, $X_{j_1} \geq 0$, $X_{j_2} < 0$, $\Psi(X_i, X_{j_1}) = t$, $\Psi(X_i, X_{j_2}) = -t$.
Following the same approach as in case 1, we have that

$\mathbb{E}[\mathcal{A}_1 | \textbf{Case 2}] \cdot P\{\textbf{Case 2}\}$

$$= \sum_{r=0}^{|\mathcal{N}_i^p|-2} \sum_{s=0}^{|\mathcal{N}_i^q|} \frac{\binom{|\mathcal{N}_i^p|-2}{r}\binom{|\mathcal{N}_i^q|}{s}}{((r+s+1)e^t + (|\mathcal{N}_i| - r - s - 1)e^{-t})^2}$$
$$\cdot (1-z)^{|\mathcal{N}_i^q|+r-s+1} \cdot z^{|\mathcal{N}_i^p|-r+s-2} \cdot (y + \mu(1-z)) \cdot (-y + \mu z).$$
(59)

**Case 3:** $X_i \geq 0$, $X_{j_1} < 0$, $X_{j_2} \geq 0$, $\Psi(X_i, X_{j_1}) = -t$, $\Psi(X_i, X_{j_2}) = t$.
Similarly, we have

$\mathbb{E}[\mathcal{A}_1 | \textbf{Case 3}] \cdot P\{\textbf{Case 3}\}$

$$= \sum_{r=0}^{|\mathcal{N}_i^p|-2} \sum_{s=0}^{|\mathcal{N}_i^q|} \frac{\binom{|\mathcal{N}_i^p|-2}{r}\binom{|\mathcal{N}_i^q|}{s}}{((r+s+1)e^t + (|\mathcal{N}_i| - r - s - 1)e^{-t})^2}$$
$$\cdot (1-z)^{|\mathcal{N}_i^q|+r-s+1} \cdot z^{|\mathcal{N}_i^p|-r+s-2} \cdot (y + \mu(1-z)) \cdot (-y + \mu z).$$
(60)

**Case 4:** $X_i \geq 0$, $X_{j_1} < 0$, $X_{j_2} < 0$, $\Psi(X_i, X_{j_1}) = -t$, $\Psi(X_i, X_{j_2}) = -t$.
Likewise, we have

$\mathbb{E}[\mathcal{A}_1 | \textbf{Case 4}] \cdot P\{\textbf{Case 4}\}$

$$= \sum_{r=0}^{|\mathcal{N}_i^p|-2} \sum_{s=0}^{|\mathcal{N}_i^q|} \frac{\binom{|\mathcal{N}_i^p|-2}{r}\binom{|\mathcal{N}_i^q|}{s} \cdot e^{-2t}}{((r+s)e^t + (|\mathcal{N}_i| - r - s)e^{-t})^2} \cdot (1-z)^{|\mathcal{N}_i^q|+r-s+1} \cdot z^{|\mathcal{N}_i^p|-r+s-2} \cdot (-y + \mu z)^2.$$
(61)

**Case 5:** $X_i < 0$, $X_{j_1} \geq 0$, $X_{j_2} \geq 0$, $\Psi(X_i, X_{j_1}) = -t$, $\Psi(X_i, X_{j_2}) = -t$.
We get that

$\mathbb{E}[\mathcal{A}_1 | \textbf{Case 5}] \cdot P\{\textbf{Case 5}\}$

$$= \sum_{r=0}^{|\mathcal{N}_i^p|-2} \sum_{s=0}^{|\mathcal{N}_i^q|} \frac{\binom{|\mathcal{N}_i^p|-2}{r}\binom{|\mathcal{N}_i^q|}{s} \cdot e^{-2t}}{((r+s+2)e^t + (|\mathcal{N}_i| - r - s - 2)e^{-t})^2}$$
$$\cdot (1-z)^{|\mathcal{N}_i^q|+r-s} \cdot z^{|\mathcal{N}_i^p|-r+s-1} \cdot (y + \mu(1-z))^2.$$
(62)

**Case 6:** $X_i < 0$, $X_{j_1} \geq 0$, $X_{j_2} < 0$, $\Psi(X_i, X_{j_1}) = -t$, $\Psi(X_i, X_{j_2}) = t$.
In the same way, we find that

$\mathbb{E}[\mathcal{A}_1 | \textbf{Case 6}] \cdot P\{\textbf{Case 6}\}$

$$= \sum_{r=0}^{|\mathcal{N}_i^p|-2} \sum_{s=0}^{|\mathcal{N}_i^q|} \frac{\binom{|\mathcal{N}_i^p|-2}{r}\binom{|\mathcal{N}_i^q|}{s}}{((r+s+1)e^t + (|\mathcal{N}_i| - r - s - 1)e^{-t})^2}$$
$$\cdot (1-z)^{|\mathcal{N}_i^q|+r-s} \cdot z^{|\mathcal{N}_i^p|-r+s-1} \cdot (y + \mu(1-z)) \cdot (-y + \mu z).$$
(63)

**Case 7:** $X_i < 0$, $X_{j_1} < 0$, $X_{j_2} \geq 0$, $\Psi(X_i, X_{j_1}) = t$, $\Psi(X_i, X_{j_2}) = -t$.
We obtain that

$$
\mathbb{E}[\mathcal{A}_1 | \textbf{Case 7}] \cdot P\{\textbf{Case 7}\}
$$
$$
= \sum_{r=0}^{|\mathcal{N}_i^p|-2} \sum_{s=0}^{|\mathcal{N}_i^q|} \frac{\binom{|\mathcal{N}_i^p|-2}{r}\binom{|\mathcal{N}_i^q|}{s}}{((r+s+1)e^t + (|\mathcal{N}_i|-r-s-1)e^{-t})^2}
$$
$$
\cdot (1-z)^{|\mathcal{N}_i^q|+r-s} \cdot z^{|\mathcal{N}_i^p|-r+s-1} \cdot (y + \mu(1-z)) \cdot (-y + \mu z).
$$
(64)

**Case 8:** $X_i < 0$, $X_{j_1} < 0$, $X_{j_2} < 0$, $\Psi(X_i, X_{j_1}) = t$, $\Psi(X_i, X_{j_2}) = t$.
Correspondingly, it follows that

$$
\mathbb{E}[\mathcal{A}_1 | \textbf{Case 8}] \cdot P\{\textbf{Case 8}\}
$$
$$
= \sum_{r=0}^{|\mathcal{N}_i^p|-2} \sum_{s=0}^{|\mathcal{N}_i^q|} \frac{\binom{|\mathcal{N}_i^p|-2}{r}\binom{|\mathcal{N}_i^q|}{s} \cdot e^{2t}}{((r+s)e^t + (|\mathcal{N}_i|-r-s)e^{-t})^2} \cdot (1-z)^{|\mathcal{N}_i^q|+r-s} \cdot z^{|\mathcal{N}_i^p|-r+s-1} \cdot (-y + \mu z)^2.
$$
(65)

Next, substituting Eqns. 58-65 into Eqn. 57, we have

$$
\mathbb{E}[\mathcal{A}_1] = \mathbb{E}[\mathcal{A}_1 | \textbf{Case 1}] \cdot P\{\textbf{Case 1}\} + \ldots + \mathbb{E}[\mathcal{A}_1 | \textbf{Case 8}] \cdot P\{\textbf{Case 8}\}
$$
$$
\overset{(i)}{\underset{\text{w.h.p.}}{=}} \sum_{r=0}^{|\mathcal{N}_i^p|-2} \sum_{s=0}^{|\mathcal{N}_i^q|} \frac{\binom{|\mathcal{N}_i^p|-2}{r}\binom{|\mathcal{N}_i^q|}{s} \cdot (1-z)^{|\mathcal{N}_i^q|+r-s} \cdot z^{|\mathcal{N}_i^p|-r+s-2}}{((r+s)e^t + (|\mathcal{N}_i|-r-s)e^{-t})^2}
$$
$$
\cdot \left( (1-z) \cdot \left( e^t(y + \mu(1-z) + e^{-t}(-y + \mu z)) \right)^2 + z \cdot \left( e^{-t}(y + \mu(1-z)) + e^t(-y + \mu z) \right)^2 \right),
$$
(66)

where $(i)$ holds since Lemma 2 ensures that $|\mathcal{N}_i| = \frac{n(p+q)}{2}\left(1 \pm \frac{\sqrt{\log n}}{10}\right) = \omega(1)$.

### F.1.2. CALCULATION OF $\mathbb{E}[\mathcal{A}_2]$

Likewise, we categorize the discussion into four distinct cases as

$$
\mathbb{E}[\mathcal{A}_2]
$$
$$
= \mathbb{E}[\mathcal{A}_2 | X_i \geq 0, X_j \geq 0] \cdot P\{X_i \geq 0, X_j \geq 0\} + \mathbb{E}[\mathcal{A}_2 | X_i \geq 0, X_j < 0] \cdot P\{X_i \geq 0, X_j < 0\}
$$
$$
+ \mathbb{E}[\mathcal{A}_2 | X_i < 0, X_j \geq 0] \cdot P\{X_i < 0, X_j \geq 0\} + \mathbb{E}[\mathcal{A}_2 | X_i < 0, X_j < 0] \cdot P\{X_i < 0, X_j < 0\}.
$$
(67)

With the definition of event $\Delta_{rs}$ in Eqn. 36, it follows that

$$
\mathbb{E}[\mathcal{A}_2 | X_i \geq 0, X_j \geq 0] \cdot P\{X_i \geq 0, X_j \geq 0\} = \sum_{r=0}^{|\mathcal{N}_i^p|-1} \sum_{s=0}^{|\mathcal{N}_i^q|} \mathbb{E}[\mathcal{A}_2 | \Delta_{rs}] \cdot P\{\Delta_{rs}\}
$$
$$
= \sum_{r=0}^{|\mathcal{N}_i^p|-1} \sum_{s=0}^{|\mathcal{N}_i^q|} \frac{\binom{|\mathcal{N}_i^p|-1}{r}\binom{|\mathcal{N}_i^q|}{s}(1-z)^{|\mathcal{N}_i^q|+r-s} z^{|\mathcal{N}_i^p|-r+s-1}}{((r+s+1)e^t + (|\mathcal{N}_i|-r-s-1)e^{-t})^2} (1-z)e^{2t} \int_0^{+\infty} X_{j_1}^2 f(X_{j1}) \, dX_{j_1}
$$
$$
\overset{(i)}{\underset{\text{w.h.p.}}{=}} \sum_{r=0}^{|\mathcal{N}_i^p|-1} \sum_{s=0}^{|\mathcal{N}_i^q|} \frac{\binom{|\mathcal{N}_i^p|-1}{r}\binom{|\mathcal{N}_i^q|}{s} \cdot (1-z)^{|\mathcal{N}_i^q|+r-s} \cdot z^{|\mathcal{N}_i^p|-r+s-1}}{((r+s)e^t + (|\mathcal{N}_i|-r-s)e^{-t})^2}
$$
$$
\cdot (1-z) \cdot e^{2t} \cdot (\mu y + \mu^2(1-z) + \sigma^2(1-z)),
$$
(68)

where $(i)$ follows from Lemma 4.

Similarly, the results for the remaining three cases are as follows,

$$\mathbb{E}[\mathcal{A}_2 | X_i \geq 0, X_j < 0] \cdot P\{X_i \geq 0, X_j < 0\}$$

$$\stackrel{\text{w.h.p.}}{=} \sum_{r=0}^{|\mathcal{N}_i^p|-1} \sum_{s=0}^{|\mathcal{N}_i^q|} \frac{\binom{|\mathcal{N}_i^p|-1}{r}\binom{|\mathcal{N}_i^q|}{s} \cdot (1-z)^{|\mathcal{N}_i|+r-s} \cdot z^{|\mathcal{N}_i^p|-r+s-1}}{((r+s)e^t + (|\mathcal{N}_i| - r - s)e^{-t})^2} \tag{69}$$

$$\cdot (1-z) \cdot e^{-2t} \cdot (-\mu y + \mu^2 z + \sigma^2 z),$$

$$\mathbb{E}[\mathcal{A}_2 | X_i < 0, X_j \geq 0] \cdot P\{X_i < 0, X_j \geq 0\}$$

$$\stackrel{\text{w.h.p.}}{=} \sum_{r=0}^{|\mathcal{N}_i^p|-1} \sum_{s=0}^{|\mathcal{N}_i^q|} \frac{\binom{|\mathcal{N}_i^p|-1}{r}\binom{|\mathcal{N}_i^q|}{s} \cdot (1-z)^{|\mathcal{N}_i^q|+r-s} \cdot z^{|\mathcal{N}_i^p|-r+s-1}}{((r+s)e^t + (|\mathcal{N}_i| - r - s)e^{-t})^2} \tag{70}$$

$$\cdot (1-z) \cdot e^{-2t} \cdot (\mu y + \mu^2 (1-z) + \sigma^2 (1-z)),$$

$$\mathbb{E}[\mathcal{A}_2 | X_i < 0, X_j < 0] \cdot P\{X_i < 0, X_j < 0\}$$

$$\stackrel{\text{w.h.p.}}{=} \sum_{r=0}^{|\mathcal{N}_i^p|-1} \sum_{s=0}^{|\mathcal{N}_i^q|} \frac{\binom{|\mathcal{N}_i^p|-1}{r}\binom{|\mathcal{N}_i^q|}{s} \cdot (1-z)^{|\mathcal{N}_i^q|+r-s} \cdot z^{|\mathcal{N}_i^p|-r+s-1}}{((r+s)e^t + (|\mathcal{N}_i| - r - s)e^{-t})^2} \tag{71}$$

$$\cdot (1-z) \cdot e^{2t} \cdot (-\mu y + \mu^2 z + \sigma^2 z).$$

Subsequently, by integrating Eqn. 68 and 71 into Eqn. 67, we obtain

$$\mathbb{E}[\mathcal{A}_2] \stackrel{\text{w.h.p.}}{=} \sum_{r=0}^{|\mathcal{N}_i^p|-1} \sum_{s=0}^{|\mathcal{N}_i^q|} \frac{\binom{|\mathcal{N}_i^p|-1}{r}\binom{|\mathcal{N}_i^q|}{s} \cdot (1-z)^{|\mathcal{N}_i^q|+r-s} \cdot z^{|\mathcal{N}_i^p|-r+s-1}}{((r+s)e^t + (|\mathcal{N}_i| - r - s)e^{-t})^2}$$

$$\cdot \left( (1-z) \cdot \left( e^{2t}(\mu y + \mu^2(1-z) + \sigma^2(1-z)) \right) + e^{-2t}\left( -\mu y + \mu^2 z + \sigma^2 z \right) \right. \tag{72}$$

$$\left. + z \cdot \left( e^{-2t}(\mu y + \mu^2(1-z) + \sigma^2(1-z)) \right) + e^{2t}\left( -\mu y + \mu^2 z + \sigma^2 z \right) \right).$$

Next, substituting Eqn. 66 and 72 into Eqn. 56 yields that

$$\mathbb{E}[\mathcal{A}] = (|\mathcal{N}_i^p|^2 - |\mathcal{N}_i^p|) \cdot \mathbb{E}[\mathcal{A}_1] + |\mathcal{N}_i^p| \cdot \mathbb{E}[\mathcal{A}_2]$$

$$\stackrel{(i)}{\underset{\text{w.h.p.}}{=}} (|\mathcal{N}_i^p|^2 - |\mathcal{N}_i^p|) \cdot \widehat{S}(z, t, |\mathcal{N}_i^p|, |\mathcal{N}_i^q|)$$

$$\cdot \left( (1-z)\left( e^t(y + \mu(1-z)) + e^{-t}(-y + \mu z) \right)^2 + z\left( e^{-t}(y + \mu(1-z)) + e^t(-y + \mu z) \right)^2 \right)$$

$$+ |\mathcal{N}_i^p| \cdot \widehat{S}(z, t, |\mathcal{N}_i^p|, |\mathcal{N}_i^q|) \tag{73}$$

$$\cdot \left( (1-z) \cdot \left( e^{2t} \cdot (\mu y + \mu^2(1-z) + \sigma^2(1-z)) + e^{-2t} \cdot (-\mu y + \mu^2 z + \sigma^2 z) \right) \right.$$

$$\left. + z \cdot \left( e^{-2t} \cdot (\mu y + \mu^2(1-z) + \sigma^2(1-z)) + e^{2t} \cdot (-\mu y + \mu^2 z + \sigma^2 z) \right) \right),$$

where $\widehat{S}(z, t, |\mathcal{N}_i^p|, |\mathcal{N}_i^q|)$ is defined in Eqn. 17, and $(i)$ is due to Lemmas 5 and 6.

## F.2. Calculation of $\mathbb{E}[\mathcal{B}]$

The calculation of $\mathbb{E}[\mathcal{B}]$ follows the exact same steps as that of $\mathbb{E}[\mathcal{A}]$. Initially, leveraging the independence of the node features, we decompose the entire expectation into the expectations of two distinct types of random variables, as indicated in Eqn. 56. Following this, we calculate the expectations of these parts separately through different cases. For the sake of succinctness, we provide the final expressions directly as follows,

$$
\begin{aligned}
\mathbb{E}[\mathcal{B}] \overset{\text{w.h.p.}}{=}\ & (|\mathcal{N}_i^q|^2 - |\mathcal{N}_i^q|) \cdot \widehat{S}\left(z, t, |\mathcal{N}_i^p|, |\mathcal{N}_i^q|\right) \\
& \cdot \left( (1-z)\left(e^t(y - \mu z) + e^{-t}(-y - \mu(1-z))\right)^2 + z\left(e^{-t}(y - \mu z) + e^t(-y - \mu(1-z))\right)^2 \right) \\
& + |\mathcal{N}_i^q| \cdot \widehat{S}\left(z, t, |\mathcal{N}_i^p|, |\mathcal{N}_i^q|\right) \\
& \cdot \left( (1-z) \cdot \left(e^{2t} \cdot (-\mu y + \mu^2 z + \sigma^2 z) + e^{-2t} \cdot (\mu y + \mu^2(1-z) + \sigma^2(1-z))\right) \right. \\
& \left. \qquad\qquad + z \cdot \left(e^{-2t} \cdot (-\mu y + \mu^2 z + \sigma^2 z) + e^{2t} \cdot (\mu y + \mu^2(1-z) + \sigma^2(1-z))\right) \right).
\end{aligned}
\tag{74}
$$

## F.3. Calculation of $\mathbb{E}[\mathcal{C}]$

First, due to the independence in the generation of node features, we have

$$
\mathbb{E}[\mathcal{B}] = 2|\mathcal{N}_i^p||\mathcal{N}_i^q| \cdot E\left[ \frac{X_{j_1} \cdot X_{j_2} \cdot e^{\Psi(X_i, X_{j_1})} \cdot e^{\Psi(X_i, X_{j_2})}}{(\sum_{l \in \mathcal{N}_i^p} e^{\Psi(X_i, X_l)} + \sum_{l' \in \mathcal{N}_i^q} e^{\Psi(X_i, X_{l'})})^2} \right],
\tag{75}
$$

where $j_i \in \mathcal{N}_i^p$ and $j_2 \in \mathcal{N}_i^q$.

Then, similarly, we divide $X_i$, $X_{j_1}$ and $X_{j_2}$ into eight cases as shown in Table 1. The only difference is that the distribution of $X_{j_2}$ changes to $N(-\mu, \sigma^2)$. After calculation and simplification, we obtain

$$
\begin{aligned}
\mathbb{E}[\mathcal{C}] \overset{\text{w.h.p.}}{=}\ & 2|\mathcal{N}_i^p||\mathcal{N}_i^q| \cdot \widehat{S}\left(z, t, |\mathcal{N}_i^p|, |\mathcal{N}_i^q|\right) \\
& \cdot \left( (1-z) \cdot \left(e^t(y + \mu(1-z)) + e^{-t}(-y + \mu z)\right) \cdot \left((e^t(-y - \mu z) + e^{-t}(y - \mu(1-z)))\right) \right. \\
& \left. \qquad + z \cdot \left(e^{-t}(y + \mu(1-z)) + e^t(-y + \mu z)\right) \cdot \left((e^{-t}(-y - \mu z) + e^t(y - \mu(1-z)))\right) \right)
\end{aligned}
\tag{76}
$$

Thus, using Eqns. 73-76, we can obtain the final result for $\mathbb{E}[(X_i')^2]$ as $\mathbb{E}[(X_i')^2] = \mathbb{E}[\mathcal{A}] + \mathbb{E}[\mathcal{B}] + \mathbb{E}[\mathcal{C}]$.

By incorporating the above results into Eqn. 54, we finally obtain

$$
\begin{aligned}
\mathrm{Var}(X_i') &= \mathbb{E}[(X_i')^2] + \mathbb{E}^2[X_i'] \\
&\overset{(i)}{\underset{\text{w.h.p.}}{=}} (|\mathcal{N}_i^p|^2 - |\mathcal{N}_i^p|) \cdot \sum_{r=0}^{|\mathcal{N}_i^p|} \sum_{s=0}^{|\mathcal{N}_i^q|} \cdot \frac{\binom{|\mathcal{N}_i^p|}{r}\binom{|\mathcal{N}_i^q|}{s} \cdot (1-z)^{|\mathcal{N}_i^q|+r-s} \cdot z^{|\mathcal{N}_i^p|-r+s}}{((r+s)e^t + (|\mathcal{N}_i|-r-s)e^{-t})^2} \\
&\quad \cdot \left( (1-z)\Big(e^t(y+\mu(1-z)) + e^{-t}(-y+\mu z)\Big)^2 + z\Big(e^{-t}(y+\mu(1-z)) + e^t(-y+\mu z)\Big)^2 \right) \\[4pt]
&+ |\mathcal{N}_i^p| \cdot \sum_{r=0}^{|\mathcal{N}_i^p|} \sum_{s=0}^{|\mathcal{N}_i^q|} \cdot \frac{\binom{|\mathcal{N}_i^p|}{r}\binom{|\mathcal{N}_i^q|}{s} \cdot (1-z)^{|\mathcal{N}_i^q|+r-s} \cdot z^{|\mathcal{N}_i^p|-r+s}}{((r+s)e^t + (|\mathcal{N}_i|-r-s)e^{-t})^2} \\
&\quad \cdot \Big( (1-z)\cdot\big(e^{2t}\cdot(\mu y + \mu^2(1-z) + \sigma^2(1-z)) + e^{-2t}\cdot(-\mu y + \mu^2 z + \sigma^2 z)\big) \\
&\qquad\qquad + z\cdot\big(e^{-2t}\cdot(\mu y + \mu^2(1-z) + \sigma^2(1-z)) + e^{2t}\cdot(-\mu y + \mu^2 z + \sigma^2 z)\big) \Big) \\[4pt]
&+ 2|\mathcal{N}_i^p||\mathcal{N}_i^q| \cdot \sum_{r=0}^{|\mathcal{N}_i^p|} \sum_{s=0}^{|\mathcal{N}_i^q|} \cdot \frac{\binom{|\mathcal{N}_i^p|}{r}\binom{|\mathcal{N}_i^q|}{s} \cdot (1-z)^{|\mathcal{N}_i^q|+r-s} \cdot z^{|\mathcal{N}_i^p|-r+s}}{((r+s)e^t + (|\mathcal{N}_i|-r-s)e^{-t})^2} \\
&\quad \cdot \Big( (1-z)\cdot\big(e^t(y+\mu(1-z)) + e^{-t}(-y+\mu z)\big)\cdot\big((e^t(-y-\mu z) + e^{-t}(y-\mu(1-z)))\big) \\
&\qquad\qquad + z\cdot\big(e^{-t}(y+\mu(1-z)) + e^t(-y+\mu z)\big)\cdot\big((e^{-t}(-y-\mu z) + e^t(y-\mu(1-z)))\big) \Big) \\[4pt]
&+ (|\mathcal{N}_i^q|^2 - |\mathcal{N}_i^q|) \cdot \sum_{r=0}^{|\mathcal{N}_i^p|} \sum_{s=0}^{|\mathcal{N}_i^q|} \cdot \frac{\binom{|\mathcal{N}_i^p|}{r}\binom{|\mathcal{N}_i^q|}{s} \cdot (1-z)^{|\mathcal{N}_i^q|+r-s} \cdot z^{|\mathcal{N}_i^p|-r+s}}{((r+s)e^t + (|\mathcal{N}_i|-r-s)e^{-t})^2} \\
&\quad \cdot \left( (1-z)\Big(e^t(y-\mu z) + e^{-t}(-y-\mu(1-z))\Big)^2 + z\Big(e^{-t}(y-\mu z) + e^t(-y-\mu(1-z))\Big)^2 \right) \\[4pt]
&+ |\mathcal{N}_i^q| \cdot \sum_{r=0}^{|\mathcal{N}_i^p|} \sum_{s=0}^{|\mathcal{N}_i^q|} \cdot \frac{\binom{|\mathcal{N}_i^p|}{r}\binom{|\mathcal{N}_i^q|}{s} \cdot (1-z)^{|\mathcal{N}_i^q|+r-s} \cdot z^{|\mathcal{N}_i^p|-r+s}}{((r+s)e^t + (|\mathcal{N}_i|-r-s)e^{-t})^2} \\
&\quad \cdot \Big( (1-z)\cdot\big(e^{2t}\cdot(-\mu y + \mu^2 z + \sigma^2 z) + e^{-2t}\cdot(\mu y + \mu^2(1-z) + \sigma^2(1-z))\big) \\
&\qquad\qquad + z\cdot\big(e^{-2t}\cdot(-\mu y + \mu^2 z + \sigma^2 z) + e^{2t}\cdot(\mu y + \mu^2(1-z) + \sigma^2(1-z))\big) \Big) \\[8pt]
&+ \Bigg( |\mathcal{N}_i^p| \sum_{r=0}^{|\mathcal{N}_i^p|-1} \sum_{s=0}^{|\mathcal{N}_i^q|} \frac{\binom{|\mathcal{N}_i^p|-1}{r}\binom{|\mathcal{N}_i^q|}{s}(1-z)^{|\mathcal{N}_i^q|+r-s} z^{|\mathcal{N}_i^p|-r+s-1}}{(r+s)e^t + (|\mathcal{N}_i|-r-s)e^{-t}} \Big((1-z)A(z,t) + zA(z,-t)\Big) \\
&\quad + |\mathcal{N}_i^q| \sum_{r=0}^{|\mathcal{N}_i^p|} \sum_{s=0}^{|\mathcal{N}_i^q|-1} \frac{\binom{|\mathcal{N}_i^p|}{r}\binom{|\mathcal{N}_i^q|-1}{s}(1-z)^{|\mathcal{N}_i^q|+r-s-1} z^{|\mathcal{N}_i^p|-r+s}}{(r+s)e^t + (|\mathcal{N}_i|-r-s)e^{-t}} \Big((1-z)A(z,-t) + zA(z,t)\Big) \Bigg)^2 \\
&\overset{(ii)}{=} \widehat{S}\big(z,t,|\mathcal{N}_i^p|,|\mathcal{N}_i^q|\big) \cdot \widehat{T}\big(z,y,t,|\mathcal{N}_i^p|,|\mathcal{N}_i^q|\big),
\end{aligned}
\tag{77}
$$

where $(i)$ follows from Eqn. 49, $(ii)$ is derived through calculations and simplifications utilizing Lemmas 5 and 6. The terms $\widehat{S}(z,t,|\mathcal{N}_i^p|,|\mathcal{N}_i^q|)$ and $\widehat{T}\big(z,y,t,|\mathcal{N}_i^p|,|\mathcal{N}_i^q|\big)$ are defined in Eqn. 17.

Similarly, if node $i \in C_0$, due to symmetry, we also have

$$\text{Var}(X_i') \stackrel{\text{w.h.p.}}{=} \widehat{S}\left(z, t, |\mathcal{N}_i^p|, |\mathcal{N}_i^q|\right) \cdot \widehat{T}\left(z, y, t, |\mathcal{N}_i^p|, |\mathcal{N}_i^q|\right). \tag{78}$$

The conclusion on variance in Theorem 2 is hereby proven.

## G. Proof of Corollary 1

This corollary consists of three statements, and we will prove each of these statements individually.

### G.1.

When $t = 0$, for the expectation part, we have for every node $i$

$$
\begin{aligned}
S\left(z, t, |\mathcal{N}_i^p|, |\mathcal{N}_i^q|\right) &= \sum_{r=0}^{|\mathcal{N}_i^p|} \sum_{s=0}^{|\mathcal{N}_i^q|} \frac{\binom{|\mathcal{N}_i^p|}{r}\binom{|\mathcal{N}_i^q|}{s} \cdot (1-z)^{|\mathcal{N}_i^q|+r-s} \cdot z^{|\mathcal{N}_i^p|-r+s}}{(r+s)e^t + (|\mathcal{N}_i| - r - s)e^{-t}} \\
&= \sum_{r=0}^{|\mathcal{N}_i^p|} \sum_{s=0}^{|\mathcal{N}_i^q|} \frac{\binom{|\mathcal{N}_i^p|}{r}\binom{|\mathcal{N}_i^q|}{s} \cdot (1-z)^{|\mathcal{N}_i^q|+r-s} \cdot z^{|\mathcal{N}_i^p|-r+s}}{|\mathcal{N}_i|} \\
&= \frac{(1-z+z)^{|\mathcal{N}_i^p|+|\mathcal{N}_i^q|}}{|\mathcal{N}_i|} = |\mathcal{N}_i|^{-1}
\end{aligned}
\tag{79}
$$

Substituting the above result into Eqn. 53, we get

$$\mu' = S\left(z, t, |\mathcal{N}_i^p|, |\mathcal{N}_i^q|\right) \cdot T\left(z, y, t, |\mathcal{N}_i^p|, |\mathcal{N}_i^q|\right) = \frac{(|\mathcal{N}_i^p| - |\mathcal{N}_i^q|) \cdot \mu}{|\mathcal{N}_i|} \stackrel{(i)}{\underset{\text{w.h.p.}}{=}} \frac{p - q}{p + q}\mu, \tag{80}$$

where $(i)$ follows from the high probability event $\Delta_3$ in Lemma 2.

For the variance part, when $t = 0$, straightforward calculations yield

$$\widehat{S}\left(z, t, |\mathcal{N}_i^p|, |\mathcal{N}_i^q|\right) = \sum_{r=0}^{|\mathcal{N}_i^p|} \sum_{s=0}^{|\mathcal{N}_i^q|} \frac{\binom{|\mathcal{N}_i^p|}{r}\binom{|\mathcal{N}_i^q|}{s}(1-z)^{|\mathcal{N}_i^q|-s+r} \cdot z^{|\mathcal{N}_i^p|+s-r}}{|\mathcal{N}_i|^2} = |\mathcal{N}_i|^{-2}, \tag{81}$$

and

$$\widehat{T}\left(z, y, t, |\mathcal{N}_i^p|, |\mathcal{N}_i^q|\right) = (|\mathcal{N}_i^p| + |\mathcal{N}_i^q|) \cdot \sigma^2 = |\mathcal{N}_i| \cdot \sigma^2. \tag{82}$$

According to the high probability event $\Delta_3$ in Lemma 2, we further obtain

$$(\sigma')^2 = \widehat{S}\left(z, t, |\mathcal{N}_i^p|, |\mathcal{N}_i^q|\right) \cdot \widehat{T}\left(z, y, t, |\mathcal{N}_i^p|, |\mathcal{N}_i^q|\right) = \frac{\sigma^2}{|\mathcal{N}_i|} \stackrel{\text{w.h.p.}}{=} \frac{1}{n(p+q)}\sigma^2. \tag{83}$$

### G.2.

When $\text{SNR} = \omega(\sqrt{\log n})$, for expectation part in the second statement, we first show that the following equation holds for every node $i$,

$$S\left(z, t, |\mathcal{N}_i^p|, |\mathcal{N}_i^q|\right) = \frac{(1-z)^{|\mathcal{N}_i|}}{|\mathcal{N}_i^p|e^t + |\mathcal{N}_i^q|e^{-t}} \cdot (1 + o(1)). \tag{84}$$

Define

$$g(r, s) \triangleq \binom{|\mathcal{N}_i^p|}{r}\binom{|\mathcal{N}_i^q|}{s} \frac{(1-z)^{|\mathcal{N}_i^q|-s+r} \cdot z^{|\mathcal{N}_i^p|+s-r}}{(r+s)e^t + (|\mathcal{N}_i| - r - s)e^{-t}}. \tag{85}$$

Then we have

$$S\left(z, t, |\mathcal{N}_i^p|, |\mathcal{N}_i^q|\right) = \sum_{r}^{|\mathcal{N}_i^p|} \sum_{s}^{|\mathcal{N}_i^q|} g(r, s). \tag{86}$$

Thus, Eqn. 84 indicates that the summation of the sequence $S\left(z, t, |\mathcal{N}_i^p|, |\mathcal{N}_i^q|\right)$ is dominated by one of its terms, specifically the term with $r = |\mathcal{N}_i^p|$ and $s = 0$. To prove Eqn. 84, it is sufficient to show that the following equation holds

$$g(r+1, s) = \omega\Big(g(r, s)\Big) \quad \text{and} \quad g(r, s+1) = o\Big(g(r, s)\Big). \tag{87}$$

Note that this statement assumes that SNR $= \mu/\sigma = \omega(\sqrt{\log n})$, by Lemma 3, we have

$$z \le \frac{1}{2}e^{-\frac{\omega(\log n)}{2}} = o(n^{-1}). \tag{88}$$

Hence,

$$\frac{g(r+1, s)}{g(r, s)} = \frac{(r+s+1)e^t + (|\mathcal{N}_i| - r - s - 1)e^{-t}}{(r+s)e^t + (|\mathcal{N}_i| - r - s)e^{-t}} \frac{\binom{|\mathcal{N}_i^p|}{r+1}}{\binom{|\mathcal{N}_i^p|}{r}} \cdot \frac{1-z}{z}$$
$$\overset{(i)}{\ge} \frac{c}{|\mathcal{N}_i^p|} \cdot \frac{1-z}{z} \overset{(ii)}{\ge} \frac{\omega(n)}{|\mathcal{N}_i^p|} = \omega(1). \tag{89}$$

where $c$ is a bounded constant, $(i)$ follows from the fact that $|\mathcal{N}_i^p|^{-1} \le \binom{|\mathcal{N}_i^p|}{r+1}/\binom{|\mathcal{N}_i^p|}{r} \le |\mathcal{N}_i^p|$ and $(ii)$ is due to Eqn. 88.

Similarly, we can show that $\frac{g(r,s+1)}{g(r,s)} = o(1)$. Then Eqn. 84 is proved. Next, since $\mu/\sigma = \omega(\sqrt{\log n})$, we can derive through simple calculations that

$$T(z, y, t, |\mathcal{N}_i^p|, |\mathcal{N}_i^q|) = |\mathcal{N}_i^p| \cdot e^t \mu(1 + o(1)) - |\mathcal{N}_i^q| \cdot e^{-t}\mu(1 + o(1)). \tag{90}$$

Hence, by combining Eqn. 84 and Eqn. 90, we have

$$\mu' = S\left(z, t, |\mathcal{N}_i^p|, |\mathcal{N}_i^q|\right) \cdot T(z, y, t, |\mathcal{N}_i^p|, |\mathcal{N}_i^q|) = \frac{(1-z)^{|\mathcal{N}_i|}(|\mathcal{N}_i^p|e^t\mu - |\mathcal{N}_i^q|e^{-t}\mu)}{|\mathcal{N}_i^p|e^t + |\mathcal{N}_i^q|e^{-t}}(1 + o(1))$$
$$\overset{(i)}{=} \frac{1 \cdot (|\mathcal{N}_i^p| \cdot e^t\mu - |\mathcal{N}_i^q| \cdot e^{-t}\mu)}{|\mathcal{N}_i^p|e^t + |\mathcal{N}_i^q|e^{-t}} \cdot (1 + o(1)) \overset{(ii)}{\underset{\text{w.h.p.}}{=}} \frac{pe^t - qe^{-t}}{pe^t + qe^{-t}}\mu, \tag{91}$$

where $(i)$ is due to the fact that $z = o(n^{-1})$ and $|\mathcal{N}_i| < n$, $(ii)$ follows from the high probability event $\Delta_3$ in Lemma 2.

For the variance part, we first define

$$\widehat{g}(r, s) \triangleq \binom{|\mathcal{N}_i^p|}{r}\binom{|\mathcal{N}_i^q|}{s} \frac{(1-z)^{|\mathcal{N}_i^q|-s+r} \cdot z^{|\mathcal{N}_i^p|+s-r}}{((r+s)e^t + (|\mathcal{N}_i| - r - s)e^{-t})^2}. \tag{92}$$

Then

$$\widehat{S}\left(z, t, |\mathcal{N}_i^p|, |\mathcal{N}_i^q|\right) = \sum_r^{|\mathcal{N}_i^p|} \sum_s^{|\mathcal{N}_i^q|} \widehat{g}(r, s). \tag{93}$$

Following the same steps as in Eqns. 87-89, we can deduce that

$$\widehat{g}(r+1, s) = \omega\Big(\widehat{g}(r, s)\Big) \quad \text{and} \quad \widehat{g}(r, s+1) = o\Big(\widehat{g}(r, s)\Big). \tag{94}$$

This implies that the summation of the sequence $\widehat{S}\left(z, t, |\mathcal{N}_i^p|, |\mathcal{N}_i^q|\right)$ is dominated by one of its terms, specifically the term with $r = |\mathcal{N}_i^p|$ and $s = 0$. Then we have

$$\widehat{S}\left(z, t, |\mathcal{N}_i^p|, |\mathcal{N}_i^q|\right) = \frac{(1-z)^{|\mathcal{N}_i|}}{(|\mathcal{N}_i^p|e^t + |\mathcal{N}_i^q|e^{-t})^2} \cdot (1 + o(1)) \overset{(i)}{=} \frac{1}{(|\mathcal{N}_i^p|e^t + |\mathcal{N}_i^q|e^{-t})^2} \cdot (1 + o(1)), \tag{95}$$

where $(i)$ is due to Eqn. 88.

Next, since $\mu/\sigma = \omega(\sqrt{\log n})$, we can derive through simple calculations that

$$\widehat{T}(z, y, t, |\mathcal{N}_i^p|, |\mathcal{N}_i^q|) = (|\mathcal{N}_i^p|e^{2t} + |\mathcal{N}_i^q|e^{-2t})\sigma^2 \cdot (1 + o(1)). \tag{96}$$

Hence, $(\sigma')^2$ is given by

$$
\begin{aligned}
(\sigma')^2 &= \widehat{S}\left(z, t, |\mathcal{N}_i^p|, |\mathcal{N}_i^q|\right) \cdot \widehat{T}(z, y, t, |\mathcal{N}_i^p|, |\mathcal{N}_i^q|) \\
&= \frac{|\mathcal{N}_i^p|e^{2t} + |\mathcal{N}_i^q|e^{-2t}}{(|\mathcal{N}_i^p|e^t + |\mathcal{N}_i^q|e^{-t})^2} \sigma^2 \cdot (1 + o(1)) \overset{(i)}{\underset{\text{w.h.p.}}{=}} \frac{pe^{2t} + qe^{-2t}}{(pe^t + qe^{-t})^2} \sigma^2,
\end{aligned}
\tag{97}
$$

where $(i)$ follows from Lemma 2.

### G.3.

When SNR $= o(1)$ and $t = O(1)$, for expectation part in the third statement, note that SNR $= \mu/\sigma = o(1)$, then with high probability $z = 1 - z = \frac{1}{2}$.

First, we establish the bound for $S\left(z, t, |\mathcal{N}_i^p|, |\mathcal{N}_i^q|\right)$ as

$$
\frac{1}{|\mathcal{N}_i|e^t} = \sum_{r=0}^{|\mathcal{N}_i^p|} \sum_{s=0}^{|\mathcal{N}_i^q|} \frac{\binom{|\mathcal{N}_i^p|}{r}\binom{|\mathcal{N}_i^q|}{s}}{2^{|\mathcal{N}_i|} \cdot |\mathcal{N}_i| \cdot e^t} \leq S\left(z, t, |\mathcal{N}_i^p|, |\mathcal{N}_i^q|\right) \leq \sum_{r=0}^{|\mathcal{N}_i^p|} \sum_{s=0}^{|\mathcal{N}_i^q|} \frac{\binom{|\mathcal{N}_i^p|}{r}\binom{|\mathcal{N}_i^q|}{s}}{2^{|\mathcal{N}_i|} \cdot |\mathcal{N}_i| \cdot e^{-t}} = \frac{1}{|\mathcal{N}_i|e^{-t}}.
\tag{98}
$$

Since $t = O(1)$, the above bound also implies $S\left(z, t, |\mathcal{N}_i^p|, |\mathcal{N}_i^q|\right) = \Theta(|\mathcal{N}_i|^{-1})$. Next, through simple calculations, we obtain

$$
T(z, y, t, |\mathcal{N}_i^p|, |\mathcal{N}_i^q|) = \frac{e^t + e^{-t}}{2} \cdot (|\mathcal{N}_i^p| - |\mathcal{N}_i^q|) \cdot \mu = \Theta\left((|\mathcal{N}_i^p| - |\mathcal{N}_i^q|) \cdot \mu\right)
\tag{99}
$$

Hence, by Lemma 2 and Eqn. 98-99, it follows that

$$
\mu' = S\left(z, t, |\mathcal{N}_i^p|, |\mathcal{N}_i^q|\right) \cdot T(z, y, t, |\mathcal{N}_i^p|, |\mathcal{N}_i^q|) = \Theta\left(\frac{|\mathcal{N}_i^p| - |\mathcal{N}_i^q|}{|\mathcal{N}_i|} \cdot \mu\right) = \Theta\left(\frac{p - q}{p + q}\mu\right)
\tag{100}
$$

As for the variance part, note that SNR $= \mu/\sigma = o(1)$, then with high probability $z = 1 - z = \frac{1}{2}$.

Following the same step as Eqn. 98, we establish the bound for $\widehat{S}\left(z, t, |\mathcal{N}_i^p|, |\mathcal{N}_i^q|\right)$ as

$$
\frac{1}{|\mathcal{N}_i|^2 \cdot e^{2t}} \leq \widehat{S}\left(z, t, |\mathcal{N}_i^p|, |\mathcal{N}_i^q|\right) \leq \frac{1}{|\mathcal{N}_i|^2 \cdot e^{-2t}}.
\tag{101}
$$

Since $t = O(1)$, the above bound also implies $\widehat{S}\left(z, t, |\mathcal{N}_i^p|, |\mathcal{N}_i^q|\right) = \Theta(|\mathcal{N}_i|^{-2})$. Next, through simple calculations, we get that

$$
\widehat{T}(z, y, t, |\mathcal{N}_i^p|, |\mathcal{N}_i^q|) = \left((|\mathcal{N}_i^p|^2 + |\mathcal{N}_i^q|^2) \cdot \frac{(e^t - e^{-t})^2}{2\pi} + (|\mathcal{N}_i^p| + |\mathcal{N}_i^q|) \cdot \frac{e^{2t} + e^{-2t}}{2}\right)\sigma^2 \cdot (1 + o(1)).
\tag{102}
$$

Hence,

$$
(\sigma')^2 = \widehat{S}\left(z, t, |\mathcal{N}_i^p|, |\mathcal{N}_i^q|\right) \cdot \widehat{T}(z, y, t, |\mathcal{N}_i^p|, |\mathcal{N}_i^q|) \overset{(i)}{=} \Theta\left(\left(c_1 \cdot (e^t - e^{-t})^2 + c_2 \cdot \frac{1}{n(p + q)}\right)\sigma^2\right),
\tag{103}
$$

where $c_1$ and $c_2$ are positive constants and $(i)$ is due to the high probability events in Lemma 2.

## H. Proof of Lemma 1

Firstly, we have

$$
\gamma(X) = \frac{1}{\sqrt{n}}\|X - \frac{\mathbf{1} \cdot \mathbf{1}^T}{n}X\|_F = \sqrt{\frac{\sum_{i=1}^n (X_i - \bar{X})^2}{n}},
\tag{104}
$$

where $\bar{X}$ is the mean value of all node features.

Based on Lemma 2, approximately half of the nodes' features are drawn independently from $N(\mu, \sigma)$, while the other half are drawn from $N(-\mu, \sigma)$. Consequently, $\bar{X} \sim N(0, \frac{\sigma^2}{n})$. As $n$ tends to infinity, we can approximate that $X_i - \bar{X} \sim N(2(\epsilon_i - 1)\mu, \sigma^2)$ for each node $i$. Thus, we obtain that, with high probability,

$$
\begin{aligned}
\mathbb{E}[(X_i - \bar{X})^2] &= \mathrm{Var}(X_i - \bar{X}) + \mathbb{E}^2[X_i - \bar{X}] = \mu^2 + \sigma^2, \\
\mathrm{Var}((X_i - \bar{X})^2) &= \mathbb{E}[(X_i - \bar{X})^4] - \mathbb{E}^2[(X_i - \bar{X})^2] \\
&\stackrel{(i)}{=} 3\sigma^4 + 6\mu^2\sigma^2 + \mu^4 - (\mu^2 + \sigma^2)^2 = 2\sigma^4 + 4\sigma^2\mu^2,
\end{aligned}
\tag{105}
$$

where $(i)$ follows from the calculation of the moment of a Gaussian distribution (see page 148 of (Edition et al., 2002)).

Note that, it suffices to prove $\sum_{i=1}^n (X_i - \bar{X})^2$ equals to $n(\mu^2 + \sigma^2)$ with high probability. Next, we apply Chebyshev's inequality to bound $\sum_{i=1}^n (X_i - \bar{X})^2$ as follows

$$
P\left\{|\sum_{i=1}^n (X_i - \bar{X})^2 - n(\mu^2 + \sigma^2)| \geq n\tau\right\} \leq \frac{(2\sigma^4 + 4\sigma^2\mu^2)^2}{n\tau^2}
\tag{106}
$$

Setting $\tau = (\mu^2 + \sigma^2)/\sqrt{\log n}$, then we have

$$
\begin{aligned}
&P\left\{n(\mu^2 + \sigma^2) \cdot (1 - \frac{1}{\sqrt{\log n}}) \leq \sum_{i=1}^n (X_i - \bar{X})^2 \leq n(\mu^2 + \sigma^2) \cdot (1 + \frac{1}{\sqrt{\log n}})\right\} \\
&\geq 1 - \frac{\log n \cdot (2\sigma^4 + 4\sigma^2\mu^2)}{n \cdot (\mu^2 + \sigma^2)^2}
\end{aligned}
\tag{107}
$$

which implies $\sum_{i=1}^n (X_i - \bar{X})^2 \stackrel{\text{w.h.p.}}{=} n(\mu^2 + \sigma^2)$.

## I. Proof of Theorem 3

According to Theorem 2, for a GAT layer, when the input node features follow a Gaussian distribution, we can precisely compute the expectation and variance of the output node features. Therefore, when $t = 0$, i.e., the graph attention layer degenerates into a simple graph convolution layer, the attention coefficients become independent of the node features, and the output node features of each layer still follow a Gaussian distribution. Subsequently, according to Corollary 1, for an $L$-layer GCN, we have

$$
\mu^{(l)} \stackrel{\text{w.h.p.}}{=} \left(\frac{p - q}{p + q}\right)^l \mu,
\tag{108}
$$

where $l \in [L]$ denotes the $l$-th layer and $\mu^{(l)}$ indicates the expectation after the $l$-th layer.

When SNR$= \omega(\sqrt{\log n})$, according to Eqn. 20, the graph attention mechanism is capable to distinguish all intra-class and inter-class edges with high probability. Consequently, the attention coefficients can be approximated as independent of the node features: setting the attention coefficient to $e^t$ for all intra-class edges and to $e^{-t}$ for all inter-class edges. Thus, the output of each layer in a multi-layer GAT also follows a Gaussian distribution. Similarly, according to Corollary 1, for an $L$-layer GAT where the attention coefficient $t$ is the same for each layer, we have

$$
\mu^{(l)} \stackrel{\text{w.h.p.}}{=} \left(\frac{pe^t - qe^{-t}}{pe^t + qe^{-t}}\right)^l \mu.
\tag{109}
$$

According to Lemma 1, we know that $\gamma(X) = \sqrt{\mu^2 + \sigma^2}$. Note that we consider the case where SNR$= \omega(\sqrt{\log n})$. According to Corollary 1, along with Eqn. 4, the SNR decreases after every GCN or GAT layer. Therefore, it follows that, for every $l \in [L]$, $\gamma(X^{(l)}) = \mu^{(l)} \cdot (1 + o(1))$.

Then, by Eqn. 108, for an $L$-layer GCN, we have that for all $l \in [L]$:

$$
\begin{aligned}
\gamma(X^{(l)}) &= \mu^{(l)} \cdot (1 + o(1)) \\
&= \left(\frac{p - q}{p + q}\right)^l \mu \cdot (1 + o(1)) = \left(1 - \frac{2q}{p + q}\right)^l \mu(1 + o(1)) \leq 2e^{\log(1 - 2q/p+q) \cdot l}\mu,
\end{aligned}
\tag{110}
$$

which indicates that the over-smoothing problem will arise.

For an $L$-layer GAT where $L = O(n)$ and a sufficiently large attention coefficient, i.e., $t = \omega(\sqrt{\log n})$, Eqn. 109 yields that

$$
\begin{aligned}
\gamma(X^{(l)}) &= \left( \frac{pe^t - qe^{-t}}{pe^t + qe^{-t}} \right)^l \mu \cdot (1 + o(1)) \\
&= \left( 1 - \frac{2q}{pe^{2t} + q} \right)^l \mu \cdot (1 + o(1)) = \Theta\left( (1 - \omega(n)^{-1})^{O(n)} \mu \right) = \Theta(\mu),
\end{aligned}
\tag{111}
$$

which indicates that the over-smoothing problem is resolved.

## J. Proof of Theorem 4

According to Theorem 1, we know that a single-layer GAT can achieve perfect node classification when SNR$= \omega(\sqrt{\log n})$. Furthermore, from Eqns. 4 and 5, we understand that over a wide range, we can ensure an increase in SNR after one layer of GAT by adjusting the value of $t$. Therefore, considering a simple case where $t = 0$, and the graph attention layer degenerates into a graph convolution layer, we have the following lemma based on the work by (Wu et al., 2022b).

**Lemma 7.** *For a featured graph generated from CSBM$(p, q, \mu, \sigma)$, suppose $p = \frac{a \log^2 n}{n}$, $q = \frac{b \log^2 n}{n}$ and $a > b > 0$ are positive constants. Given an $L$-th layer linear GCN with each layer being defined in Eqn. 6 without the non-linear activation function, let $\mu'$ and $\sigma^{(l)}$ be the expectation and variance of the output node feature after the $l$-th layer. For $L = O\left( \frac{\log n}{\log(b \log^2 n)} \right)$, the following holds with high probability:*

$$
\text{(i). } \mu^{(l)} = \left( \frac{a - b}{a + b} \right)^l \mu, \qquad \text{(ii). } (\sigma^2)^{(l)} = \frac{c_1}{(c_2 \cdot \log^2 n)^l} \sigma^2,
\tag{112}
$$

*where $c_1, c_2$ are two positive constants.*

*Proof.* See Appendix K for the detailed proof. $\square$

Based on Lemma 7 and Theorem 1, we consider a multi-layer GAT network where the first $L$ layers use $t = 0$, and the $(L + 1)$-th layer sets $t$ to a sufficiently large value. To achieve perfect node classification, it is sufficient to ensure that the expectation and variance of the node features after $L$ layers satisfy $\mu^{(L)}/\sigma^{(L)} = \omega(\sqrt{\log n})$. Note that, by setting $L = \frac{\log n}{\log(b \log^2 n)}$ and using Eqn. 112, it follows that

$$
\frac{\mu^{(L)}}{\sigma^{(L)}} = \frac{\left( \frac{a-b}{a+b} \right)^L \cdot (\sqrt{c_2} \log n)^L}{\sqrt{c_1}} \cdot \frac{\mu}{\sigma} = \frac{(c' \log n)^{\frac{\log n}{\log(b \log^2 n)}}}{\sqrt{c_1}} \cdot \frac{\mu}{\sigma} \geq (\log n)^{\frac{\log n}{3 \log \log n}} \cdot \frac{\mu}{\sigma} = n^{\frac{1}{3}} \cdot \frac{\mu}{\sigma},
\tag{113}
$$

where $c' = \sqrt{c_2}(a - b)/(a + b)$ is a constant.

Hence, to satisfy the condition $\mu^{(L)}/\sigma^{(L)} = \omega(\sqrt{\log n})$, it is sufficient to satisfy condition $n^{\frac{1}{3}} \cdot \mu/\sigma = \omega(\sqrt{\log n})$, i.e., SNR $= \omega(\sqrt{\log n}/\sqrt[3]{n})$. This completes the proof.

## K. Additional Proofs of Lemmas

In this part, we present the proofs for several lemmas that are utilized in the preceding proofs. For clarity, we restate each lemma before presenting its proof.

**Lemma 3** Assume a random variable $y \sim N(0, 1)$, then for any constant $s > 0$, the following tail bound holds,

$$
P\{y \geq s\} = \Phi(s) \leq \min \left\{ \frac{1}{2} e^{-\frac{s^2}{2}}, \frac{1}{s\sqrt{2\pi}} e^{-\frac{s^2}{2}} \right\}.
\tag{114}
$$

*Proof.* We first prove the former part of the tail bound,

$$
\begin{aligned}
P\{y \geq s\} &= \int_s^{+\infty} \frac{1}{\sqrt{2\pi}} e^{-\frac{y^2}{2}} \, dy \\
&= \int_0^{+\infty} \frac{1}{\sqrt{2\pi}} e^{-\frac{(y+s)^2}{2}} \, dy.
\end{aligned}
\tag{115}
$$

For any $y \geq 0$, we have

$$
\begin{aligned}
e^{-\frac{(y+s)^2}{2}} &= e^{-\frac{y^2+2ys+s^2}{2}} \\
&\leq e^{-\frac{y^2}{2}} \cdot e^{-\frac{s^2}{2}}.
\end{aligned}
\tag{116}
$$

Hence,

$$
\begin{aligned}
P\{y > s\} &\leq \int_0^{+\infty} \frac{1}{\sqrt{2\pi}} e^{-\frac{y^2}{2}} \cdot e^{-\frac{s^2}{2}} \, dy \\
&= e^{-\frac{t^2}{2}} \cdot \int_0^{+\infty} \frac{1}{\sqrt{2\pi}} e^{-\frac{y^2}{2}} \, dy \\
&= \frac{1}{2} e^{-\frac{s^2}{2}}.
\end{aligned}
\tag{117}
$$

Then, we give the proof of the second part. Note that

$$
\begin{aligned}
P\{y > s\} &= \int_s^{+\infty} \frac{1}{\sqrt{2\pi}} e^{-\frac{y^2}{2}} \, dy \\
&\leq \int_s^{+\infty} \frac{y}{s} \frac{1}{\sqrt{2\pi}} e^{-\frac{y^2}{2}} \, dy \\
&= \frac{1}{t\sqrt{2\pi}} e^{-\frac{t^2}{2}}.
\end{aligned}
\tag{118}
$$

By integrating Eqn. 117 with Eqn. 118, the proof is completed.

$\square$

**Lemma 4** Assume a random variable $x \sim N(\mu, \sigma^2)$ with $f(x)$ being the probability density function of $x$, then

$$
\begin{cases}
\int_0^{+\infty} x f(x) \, dx = \frac{\sigma}{\sqrt{2\pi}} e^{-\frac{\mu^2}{2\sigma^2}} + \mu\left(1 - \Phi\left(\frac{\mu}{\sigma}\right)\right), \\
\int_{-\infty}^0 x f(x) \, dx = -\frac{\sigma}{\sqrt{2\pi}} e^{-\frac{\mu^2}{2\sigma^2}} + \mu\Phi\left(\frac{\mu}{\sigma}\right),
\end{cases}
\tag{119}
$$

and

$$
\begin{cases}
\int_0^{+\infty} x^2 f(x) \, dx = \mu\frac{\sigma}{\sqrt{2\pi}} e^{-\frac{\mu^2}{2\sigma^2}} + \mu^2\left(1 - \Phi\left(\frac{\mu}{\sigma}\right)\right) + \sigma^2\left(1 - \Phi\left(\frac{\mu}{\sigma}\right)\right), \\
\int_{-\infty}^0 x^2 f(x) \, dx = -\mu\frac{\sigma}{\sqrt{2\pi}} e^{-\frac{\mu^2}{2\sigma^2}} + \mu^2\Phi\left(\frac{\mu}{\sigma}\right) + \sigma^2\Phi\left(\frac{\mu}{\sigma}\right).
\end{cases}
\tag{120}
$$

Accordingly, if $x \sim N(-\mu, \sigma^2)$, then

$$
\begin{cases}
\int_0^{+\infty} x f(x) \, dx = \frac{\sigma}{\sqrt{2\pi}} e^{-\frac{\mu^2}{2\sigma^2}} - \mu\Phi\left(\frac{\mu}{\sigma}\right), \\
\int_{-\infty}^0 x f(x) \, dx = -\frac{\sigma}{\sqrt{2\pi}} e^{-\frac{\mu^2}{2\sigma^2}} - \mu\left(1 - \Phi\left(\frac{\mu}{\sigma}\right)\right),
\end{cases}
\tag{121}
$$

and

$$
\begin{cases}
\int_0^{+\infty} x^2 f(x) \, dx = -\mu\frac{\sigma}{\sqrt{2\pi}} e^{-\frac{\mu^2}{2\sigma^2}} + \mu^2\Phi\left(\frac{\mu}{\sigma}\right) + \sigma^2\Phi\left(\frac{\mu}{\sigma}\right), \\
\int_{-\infty}^0 x^2 f(x) \, dx = \mu\frac{\sigma}{\sqrt{2\pi}} e^{-\frac{\mu^2}{2\sigma^2}} + \mu\left(1 - \Phi\left(\frac{\mu}{\sigma}\right)\right) + \sigma^2\left(1 - \Phi\left(\frac{\mu}{\sigma}\right)\right).
\end{cases}
\tag{122}
$$

*Proof.* Here, we only present the proof when $x \sim N(\mu, \sigma^2)$, the proof for the other case can be obtained similarly. Note that

$$
\begin{aligned}
\int_0^{+\infty} x f(x)\, dx &= \int_0^{+\infty} x \cdot \frac{1}{\sigma\sqrt{2\pi}} \cdot e^{-\frac{(x-\mu)^2}{2\sigma^2}}\, dx \\
&= \int_0^{+\infty} (x-\mu)\frac{1}{\sigma\sqrt{2\pi}} \cdot e^{-\frac{(x-\mu)^2}{2\sigma^2}}\, dx + \mu \int_0^{+\infty} \frac{1}{\sigma\sqrt{2\pi}} \cdot e^{-\frac{(x-\mu)^2}{2\sigma^2}}\, dx \\
&= -\frac{\sigma}{\sqrt{2\pi}} e^{-\frac{(x-\mu)^2}{2\sigma^2}}\Big|_0^{+\infty} + \mu\Big(1 - \Phi\Big(\frac{\mu}{\sigma}\Big)\Big) = \frac{\sigma}{\sqrt{2\pi}} e^{-\frac{\mu^2}{2\sigma^2}} + \mu\Big(1 - \Phi\Big(\frac{\mu}{\sigma}\Big)\Big).
\end{aligned}
\tag{123}
$$

Likewise, we have

$$
\int_{-\infty}^0 x f(x)\, dx = -\frac{\sigma}{\sqrt{2\pi}} e^{-\frac{\mu^2}{2\sigma^2}} + \mu\Phi\Big(\frac{\mu}{\sigma}\Big).
\tag{124}
$$

Next, Eqn. 120 is obtained by

$$
\begin{aligned}
\int_0^{+\infty} x^2 f(x)\, dx &= \int_0^{+\infty} (x^2 - 2x\mu + \mu^2) f(x)\, dx + \int_0^{+\infty} 2x\mu \cdot f(x)\, dx - \mu^2 \int_0^{+\infty} f(x)\, dx \\
&= \int_0^{+\infty} (x-\mu)^2 f(x)\, dx + 2\mu\Big(\frac{\sigma}{\sqrt{2\pi}} e^{-\frac{\mu^2}{2\sigma^2}} + \mu\Big(1 - \Phi\Big(\frac{\mu}{\sigma}\Big)\Big)\Big) - \mu^2 \cdot \Big(1 - \Phi\Big(\frac{\mu}{\sigma}\Big)\Big),
\end{aligned}
\tag{125}
$$

and

$$
\begin{aligned}
\int_0^{+\infty} (x-\mu)^2 f(x)\, dx &= \int_0^{+\infty} (x-\mu)^2 \frac{1}{\sigma\sqrt{2\pi}} e^{-\frac{(x-\mu)^2}{2\sigma^2}}\, dx \\
&= \int_0^{+\infty} -\frac{\sigma}{\sqrt{2\pi}}(x-\mu)\Big(e^{-\frac{(x-\mu)^2}{2\sigma^2}}\Big)'\, dx \\
&= -\frac{\sigma}{\sqrt{2\pi}} e^{-\frac{(x-\mu)^2}{2\sigma^2}}\Big|_0^{+\infty} + \int_0^{+\infty} \frac{\sigma}{\sqrt{2\pi}} e^{-\frac{(x-\mu)^2}{2\sigma^2}}\, dx \\
&= -\mu\frac{\sigma}{\sqrt{2\pi}} e^{-\frac{\mu^2}{2\sigma^2}} + \sigma^2\Big(1 - \Phi\Big(\frac{\mu}{\sigma}\Big)\Big).
\end{aligned}
\tag{126}
$$

Hence,

$$
\begin{aligned}
&\int_0^{+\infty} x^2 f(x)\, dx \\
&= -\mu\frac{\sigma}{\sqrt{2\pi}} e^{-\frac{\mu^2}{2\sigma^2}} + \sigma^2\Big(1 - \Phi\Big(\frac{\mu}{\sigma}\Big)\Big) + 2\mu\Big(\frac{\sigma}{\sqrt{2\pi}} e^{-\frac{\mu^2}{2\sigma^2}} + \mu\Big(1 - \Phi\Big(\frac{\mu}{\sigma}\Big)\Big)\Big) - \mu^2 \cdot \Big(1 - \Phi\Big(\frac{\mu}{\sigma}\Big)\Big) \\
&= \mu\frac{\sigma}{\sqrt{2\pi}} e^{-\frac{\mu^2}{2\sigma^2}} + \mu^2\Big(1 - \Phi\Big(\frac{\mu}{\sigma}\Big)\Big) + \sigma^2\Big(1 - \Phi\Big(\frac{\mu}{\sigma}\Big)\Big).
\end{aligned}
\tag{127}
$$

Similarly, it can be calculated that

$$
\int_{-\infty}^0 x^2 f(x)\, dx = -\mu\frac{\sigma}{\sqrt{2\pi}} e^{-\frac{\mu^2}{2\sigma^2}} + \mu^2\Phi\Big(\frac{\mu}{\sigma}\Big) + \sigma^2\Phi\Big(\frac{\mu}{\sigma}\Big).
\tag{128}
$$

$\square$

**Lemma 5**  Assume $0 < x < 1/2$, for any constants $t > 0$ and $k > 0$, let

$$
\Gamma(n, m) \triangleq \sum_{i=0}^n \sum_{j=0}^m \frac{\binom{n}{i}\binom{m}{j}(1-x)^{m+i-j} x^{n-i+j}}{((i+j)e^t + (n+m-i-j)e^{-t})^k}.
$$

Then the following equation holds

$$
\lim_{n,m\to+\infty} \Gamma(n + c_1, m + c_2) = \Gamma(n, m),
\tag{129}
$$

where $c_1$ and $c_2$ are positive integer constants.

*Proof.* Our approach to the proof starts with establishing the boundedness of the sequence $\Gamma(n, m)$. Subsequently, we show that the sequence is monotonically decreasing in both $n$ and $m$. Then applying the Monotone Convergence Theorem (Yeh, 2014) is sufficient to complete the proof.

Firstly, since $t > 0$, it is important to note the following facts that

$$\sum_{i=0}^{n} \sum_{j=0}^{m} \binom{n}{i} \binom{m}{j} (1-x)^{m+i-j} x^{n-i+j} = (1-x+x)^{n+m} = 1, \tag{130}$$

and

$$\sum_{i=0}^{n} \sum_{j=0}^{m} \frac{\binom{n}{i}\binom{m}{j}(1-x)^{m+i-j} x^{n-i+j}}{(n+m)^k \cdot e^{kt}} \leq \Gamma(n,m) \leq \sum_{i=0}^{n} \sum_{j=0}^{m} \frac{\binom{n}{i}\binom{m}{j}(1-x)^{m+i-j} x^{n-i+j}}{(n+m)^k \cdot e^{-kt}}. \tag{131}$$

Thus, $\Gamma(n, m)$ is bounded by

$$\frac{1}{(n+m)^k e^{kt}} \leq \Gamma(n,m) \leq \frac{1}{(n+m)^k e^{-kt}}. \tag{132}$$

Then, for a positive integer constant $c_1$, we have

$$\sum_{i=0}^{n+c_1} \sum_{j=0}^{m} \binom{n+c_1}{i} \binom{m}{j} (1-x)^{m+i-j} x^{n+c_1-i+j} = \sum_{i=0}^{n} \sum_{j=0}^{m} \binom{n}{i} \binom{m}{j} (1-x)^{m+i-j} x^{n-i+j} = 1. \tag{133}$$

And, for any $i, j$,

$$\frac{1}{(i+j)e^t + (n+c_1+m-i-j)e^{-t}} \leq \frac{1}{(i+j)e^t + (n+m-i-j)e^{-t}} \tag{134}$$

Hence, $\Gamma(n + c_1, m) \leq \Gamma(n, m)$ holds. Likewise, assuming another positive integer constant $c_2$, it can be deduced that

$$\Gamma(n + c_1, m + c_2) \leq \Gamma(n, m). \tag{135}$$

Consequently, for the sequence $\Gamma(n, m)$, Eqn. 132 and 135 guarantee both the monotonicity and the boundedness of the sequence. By the Monotone Convergence Theorem, it follows that the sequence converges, which also ensures that $\lim_{n,m \to +\infty} \Gamma(n + c_1, m + c_2)/\Gamma(n, m) = 1$.

$\square$

**Lemma 6** Assume $0 < x < 1/2$, for any constant $t > 0$, we define $A(n,m) \triangleq \sum_{i=0}^{n} \sum_{j=0}^{m} \frac{\binom{n}{i}\binom{m}{j}(1-x)^{m+i-j} x^{n-i+j}}{((i+j)e^t+(n+m-i-j)e^{-t})^2}$, and $B(n,m) \triangleq \left( \sum_{i=0}^{n} \sum_{j=0}^{m} \frac{\binom{n}{i}\binom{m}{j}(1-x)^{m+i-j} x^{n-i+j}}{(i+j)e^t+(n+m-i-j)e^{-t}} \right)^2$. Then, for $n + m \to +\infty$, we have

$$A(n,m) = \Theta((n+m)^{-2}), \ B(n,m) = \Theta((n+m)^{-2}), \ A(n,m) - B(n,m) = o((n+m)^{-3}).$$

*Proof.* Define $a_{ij} \triangleq e^{-t}[(n+m) + (e^{2t}-1)(i+j)]$, $b_{ij} \triangleq \binom{n}{i}\binom{m}{j}(1-x)^{m+i-j} x^{n-i+j}$ and $[n] \times [m] = \{(i,j)|0 \leq i \leq n, 0 \leq j \leq m, i, j \in \mathbb{Z}\}$, $[n] \times [m] \times [n] \times [m] = \{(i_1, j_1, i_2, j_2)|0 \leq i_l \leq n, 0 \leq j_l \leq m, i_l, j_l \in \mathbb{Z}, \ l \in \{1, 2\}\}$, then we can rewrite:

$$A(n,m) = \sum_{(i,j) \in [n] \times [m]} \frac{b_{ij}}{a_{ij}^2}, \ B(n,m) = \left( \sum_{(i,j) \in [n] \times [m]} \frac{b_{ij}}{a_{ij}} \right)^2$$

Firstly, note that $\sum\limits_{(i,j)\in[n]\times[m]} b_{ij} = 1$, to see this:

$$
\begin{aligned}
\sum_{(i,j)\in[n]\times[m]} b_{ij} &= \sum_{(i,j)\in[n]\times[m]} \binom{n}{i}\binom{m}{j}(1-x)^{m+i-j}x^{n-i+j} \\
&= \sum_{(i,j)\in[n]\times[m]} \left(\binom{n}{i}(1-x)^i x^{n-i}\right)\left(\binom{m}{j}x^j(1-x)^{m-j}\right) \\
&= \left(\sum_{i=0}^{n}\binom{n}{i}(1-x)^i x^{n-i}\right)\left(\sum_{j=0}^{m}\binom{m}{j}x^j(1-x)^{m-j}\right) \\
&= (1-x+x)^n(x+1-x)^m \\
&= 1
\end{aligned}
$$

By definition, it is clear that $e^{-t}(n+m) \le a_{ij} \le e^t(n+m)$, then:

$$
|A(n,m)| = \sum_{(i,j)\in[n]\times[m]} \frac{b_{ij}}{a_{ij}^2} \le \frac{e^{2t}}{(n+m)^2}\sum_{(i,j)\in[n]\times[m]} b_{ij} = \frac{e^{2t}}{(n+m)^2}
$$

$$
|A(n,m)| = \sum_{(i,j)\in[n]\times[m]} \frac{b_{ij}}{a_{ij}^2} \ge \frac{e^{-2t}}{(n+m)^2}\sum_{(i,j)\in[n]\times[m]} b_{ij} = \frac{e^{-2t}}{(n+m)^2}
$$

Hence, $A(n,m) = \Theta((n+m)^{-2})$, Now we show that $|A(n,m)-B(n,m)|$ can be upper bounded by $e^{6t}x(1-x)(n+m)^{-3}$. The key observation is that $b_{ij} = P(X=i, Y=j)$, where $X$ and $Y$ follow from two Binomial distributions, i.e., $X \sim \text{Bino}(n, 1-x), Y \sim \text{Bino}(m, x)$, while $X$ and $Y$ are independent:

$$
\begin{aligned}
&|A(n,m) - B(n,m)| \\
&= \left| \sum_{(i,j)\in[n]\times[m]} \frac{b_{ij}}{a_{ij}^2} - \left(\sum_{(i,j)\in[n]\times[m]} \frac{b_{ij}}{a_{ij}}\right)^2 \right| \\
&= \left| \left(\sum_{(i,j)\in[n]\times[m]} \frac{b_{ij}}{a_{ij}^2}\right)\left(\sum_{(i,j)\in[n]\times[m]} b_{ij}\right) - \left(\sum_{(i,j)\in[n]\times[m]} \frac{b_{ij}}{a_{ij}}\right)^2 \right| \\
&\overset{(i)}{=} \frac{1}{2}\left| \sum_{\substack{(i_1,j_1)\in[n]\times[m] \\ (i_2,j_2)\in[n]\times[m]}} \left( \frac{\sqrt{b_{i_1 j_1}}}{a_{i_1 j_1}}\sqrt{b_{i_2 j_2}} - \frac{\sqrt{b_{i_2 j_2}}}{a_{i_2 j_2}}\sqrt{b_{i_1 j_1}} \right)^2 \right| \\
&= \frac{1}{2}\left| \sum_{\substack{(i_1,j_1)\in[n]\times[m] \\ (i_2,j_2)\in[n]\times[m]}} b_{i_1 j_1} b_{i_2 j_2}\left(\frac{a_{i_1 j_1} - a_{i_2 j_2}}{a_{i_1 j_1} a_{i_2 j_2}}\right)^2 \right| \\
&= \frac{1}{2}\left| \sum_{\substack{(i_1,j_1)\in[n]\times[m] \\ (i_2,j_2)\in[n]\times[m]}} b_{i_1 j_1} b_{i_2 j_2}\left(\frac{(e^t - e^{-t})[(i_1 + j_1) - (i_2 + j_2)]}{a_{i_1 j_1} a_{i_2 j_2}}\right)^2 \right|
\end{aligned}
$$

(136)

$$\overset{(ii)}{\leq} \frac{e^{6t}}{2(n+m)^4} \Big| \sum_{\substack{(i_1,j_1)\in[n]\times[m] \\ (i_2,j_2)\in[n]\times[m]}} b_{i_1j_1} b_{i_2j_2} [(i_1+j_1)-(i_2+j_2)]^2 \Big|$$

$$\overset{(iii)}{=} \frac{e^{6t}}{2(n+m)^4} \Big| \sum_{\substack{(i_1,j_1)\in[n]\times[m] \\ (i_2,j_2)\in[n]\times[m]}} P(X_1=i_1,Y_1=j_1)P(X_2=i_2,Y_2=j_2)[(i_1+j_1)-(i_2+j_2)]^2 \Big|$$

$$= \frac{e^{6t}}{2(n+m)^4} \mathbb{E}[(X_1+Y_1-X_2-Y_2)^2]$$

$$\overset{(iv)}{=} \frac{e^{6t}}{(n+m)^4}\Big(\mathrm{Var}(X_1)+\mathrm{Var}(Y_1)\Big) = \frac{e^{6t}x(1-x)}{(n+m)^3}$$

Here is some notes for the above proof: $(i)$ Apply Lagrange's identity; $(ii)$ Plug in $a_i j$; $(iii)$ using previous observe for $b_{ij}$, where $X_l \sim \mathrm{Bino}(n, 1-x), Y_l \sim \mathrm{Bino}(m, x)$, $l \in \{1, 2\}$ and they are independent; $(iv)$ Linearity of Expectation.

Finally, given $A(n,m) = \Theta((n+m)^{-2})$ and $A(n,m) - B(n,m) = o((n+m)^{-3})$, it is easy to see $B(n,m) = \Theta((n+m)^{-2})$, so we finish the proof. □

**Lemma 7** For a featured graph generated from CSBM$(p,q,\mu,\sigma)$, suppose $p = \frac{a\log^2 n}{n}$, $q = \frac{b\log^2 n}{n}$ and $a > b > 0$ are positive constants. Given an $L$-th layer linear GCN with each layer being defined in Eqn. 6 without the non-linear activation function, let $\mu'$ and $\sigma^{(l)}$ be the expectation and variance of the output node feature after the $l$-th layer. For $L = O\Big(\frac{\log n}{\log(b\log^2 n)}\Big)$, the following holds with high probability:

$$1.\ \mu^{(l)} = \Big(\frac{a-b}{a+b}\Big)^l \mu, \quad 2.\ (\sigma^2)^{(l)} = \frac{c_1}{(c_2 \cdot \log^2 n)^l}\sigma^2, \tag{137}$$

where $c_1, c_2$ are two positive constants.

*Proof.* The proof of the first part in Eqn. 137 can be directly derived by substituting the values of $p$ and $q$ into Eqn. 108. For the second part concerning the change in variance, we refer to Theorem 2 from (Wu et al., 2022b). By substituting the values of $p = \frac{a\log^2 n}{n}$ and $q = \frac{b\log^2 n}{n}$, we obtain

$$\frac{c_3}{((a+b)\cdot \log^2 n)^l} \cdot \sigma^2 \leq (\sigma^2)^{(l)} \leq \frac{c_4}{(a\cdot \log^2 n)^l} \cdot \sigma^2, \tag{138}$$

where $c_3, c_4$ are two positive constants. Thus, apparently, there exists two constants $c_1 \in (a, a+b)$ and $c_2 > 0$ such that $(\sigma^2)^{(l)} = \frac{c_1}{(c_2 \cdot \log^2 n)^l}\sigma^2$.

The above equation demonstrates that using multiple layers of graph convolution can reduce the variance of node features. However, Theorem 2 in (Wu et al., 2022b) also indicates that this improvement is only effective in the initial layers. Specifically, the proof of Theorem 2 in (Wu et al., 2022b) reveals that the enhancement fundamentally arises from incorporating higher-order neighbor information. In the context of random graphs, we can estimate the graph's diameter, which allows us to determine the maximum number of hops between any two nodes. This estimation consequently indicates the upper limit on the number of graph convolution layers (i.e., the value of $L$) that can effectively reduce variance.

For a graph $G$ generated by the above CSBM, let $\mathrm{diam}(G)$ denote its diameter. According to Theorem 7.2 in (Frieze & Karoński, 2015), we have

$$\mathrm{diam}(G) \overset{\text{w.h.p.}}{\geq} \frac{\log n}{\log(b\log^2 n)}, \tag{139}$$

which means the maximum number of GCN layers that can reduce the variance of node features is $L = O\Big(\frac{\log n}{\log(b\log^2 n)}\Big)$.

□

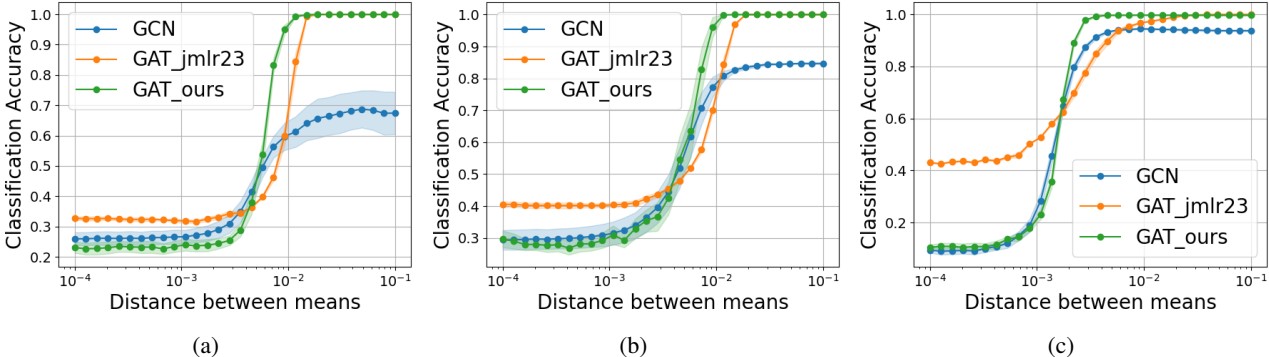

*Figure 3.* Additional experimental results on real-world datasets. Figures 3a, 3b and 3c illustrate the results for the `Citeseer`, `Cora`, and `Pubmed` datasets, respectively.

*Table 2.* Dataset characteristics.

| Dataset | Number of Nodes | Number of Edges | Number of Classes | Feature Dimension |
|---|---|---|---|---|
| `Citeseer` | 3,327 | 4,732 | 6 | 3,703 |
| `Cora` | 2,708 | 5,429 | 7 | 1,433 |
| `Pubmed` | 19,717 | 44,338 | 3 | 500 |
| `ogbn-arxiv` | 169,343 | 1,166,243 | 40 | 128 |

*Table 3.* Comparison of running times for GCN, GAT-jmlr and GAT*.

| Method | GCN | GAT-jmlr | GAT* |
|---|---|---|---|
| Runtime (/s) | 8.63 | 10.03 | 8.93 |

## L. Additional Experiments

### L.1. Comparative Analysis of Our Graph Attention Mechanism and Fountoulakis et al. (2023)

We conducted additional experiments on three real-world datasets (`Citeseer`, `Cora`, and `Pubmed`) to compare the capabilities of our proposed graph attention mechanism with the mechanism from (Fountoulakis et al., 2023). The characteristics of the datasets is provided in Table 2. The experimental setup mirrors that used in the experiments from (Fountoulakis et al., 2023). Specifically, the three datasets contain multiple classes, and in each experiment, we perform one-vs-all classification for a single class, converting it into a binary classification problem, as our attention mechanism is designed for binary classification. To control the mean of node features across different classes, we compute the mean of the features for each class using their labels and then adjust the features of nodes in that class by subtracting the mean and adding either $\mu$ or $-\mu$.

For the three datasets, we classify the 0 class in a one-vs-all manner and record the classification accuracy for that class. The training and testing set splits follow the default settings of PyTorch Geometric. We designed three models: a graph convolutional network, a GAT network utilizing the attention mechanism from (Fountoulakis et al., 2023) (denoted as GAT-jmlr), and a GAT employing the attention mechanism defined in Eqn. 12 (denoted as GAT*). Each of these models incorporates a single attention layer. In GAT-jmlr, the parameters $\beta$ and $R$ are set to $0.2$ and $1$, respectively, while the parameter $t$ in GAT* is set to $1$. Figure 3 illustrates how the classification accuracy of the three models varies with changes in the distance between the means of the node features for the two classes. From Figure 3, we see that when the distance between the means of the node features for the two classes is large, indicating low feature noise, GAT* performs the best. In contrast, when the distance is small, suggesting high feature noise, GAT-jmlr delivers the best results. Overall, GAT* significantly enhances GCN performance, especially under conditions of low feature noise.

Additionally, Table 3 presents the runtime of the three methods. For the three datasets, we set the number of epochs to 100 and ran each dataset once, recording the total time taken for all runs. Table 3 shows that the graph attention mechanism we designed is slightly more computationally efficient than the one presented in (Fountoulakis et al., 2023), which confirms our

analysis in Appendix B.

## L.2. Additional experiments on `obgn-arxiv` dataset

To further substantiate our theoretical findings, we conducted additional experiments on a larger dataset—ogbn-arxiv—and employed a more advanced graph attention mechanism, GATv2. Specifically, we evaluated the performance of four models under two types of noise: feature noise ($f_n$) and structure noise ($s_n$). The four models are: GCN, GATv2, GATv2*, and GATv2*(temp). Here, GATv2* denotes a hybrid architecture that uses graph convolutional layers in the first two layers and a GATv2 layer in the final layer, similar to the GAT* model described in 4.2. GATv2*(temp) extends this architecture by introducing a tunable temperature parameter $T = t^{-1}$ into the softmax computation of the attention coefficients, allowing for precise control over attention intensity. In our implementation, the values of $T$ are set to $[2, 2, 0.5]$ across the three GATv2 layers. Feature noise ($f_n$) is introduced by reducing the expected distance between node features of different classes, while structure noise ($s_n$) is added by randomly perturbing graph edges. Both $f_n$ and $s_n$ take values in the range $[0, 1]$, with larger values indicating higher noise levels. The accuracy heatmaps are shown in Figure 4.

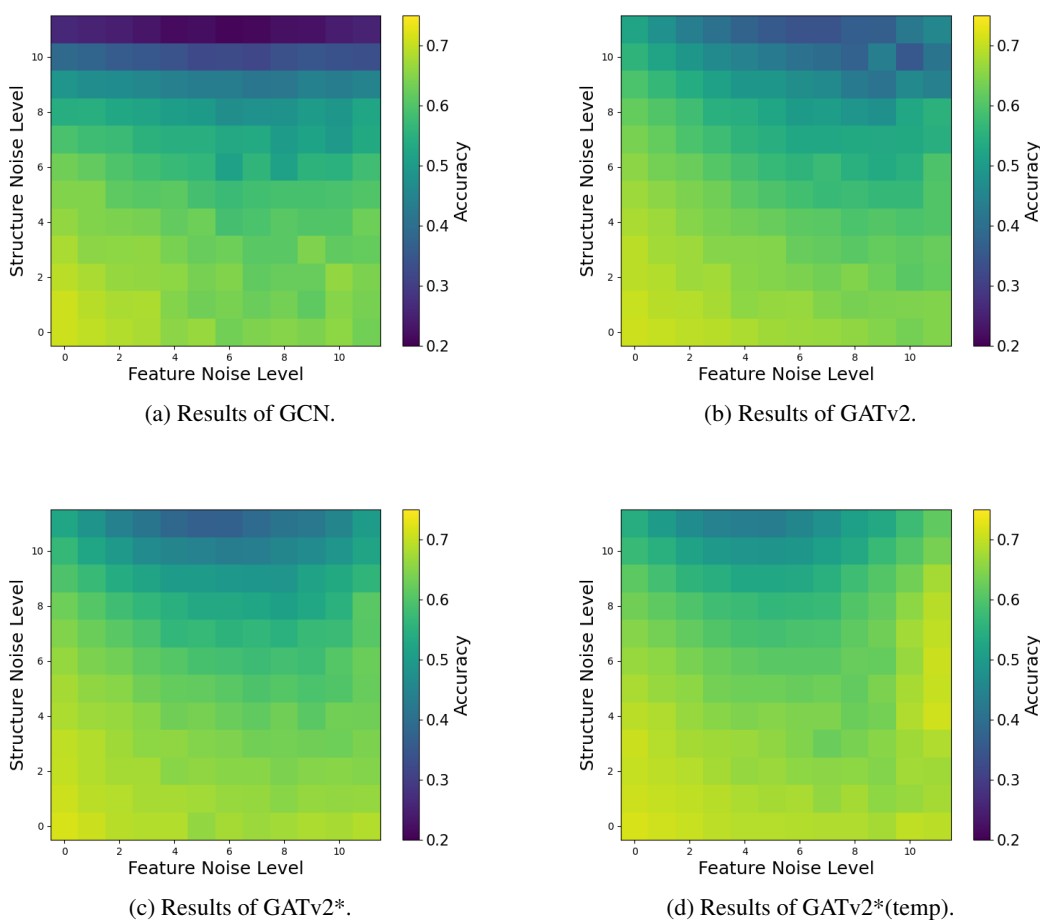

(a) Results of GCN.

(b) Results of GATv2.

(c) Results of GATv2*.

(d) Results of GATv2*(temp).

*Figure 4.* Accuracy heatmaps of models on `ogbn-arxiv` under varying structure and feature noise levels.

From Figure 4, we observe that GATv2*(temp) achieves the best overall performance. Moreover, methods incorporating attention mechanisms exhibit significant improvements when structure noise is strong and feature noise is weak. In contrast, when feature noise dominates and structure noise is minimal, the performance gain is marginal. This observation is broadly consistent with our theoretical results. However, there is a subtle discrepancy: according to theory, attention-based methods are expected to degrade performance under strong feature noise and weak structure noise. Yet, in the figure, these methods

do not show notably worse performance; instead, their results are comparable to those of GCN.

To investigate this discrepancy, we visualized the attention coefficients of the GATv2 model under conditions of high feature noise and low structure noise, focusing on the nine nodes with the highest number of neighbors, as shown in Figure 5. We found that in this setting, the attention mechanism assigns nearly uniform weights to all neighbors through its learnable parameters, effectively degenerating into a graph convolutional layer. This indicates that GATv2 possesses stronger learning capability compared to GAT. Importantly, this behavior does not contradict our theoretical analysis, which is based on attention layers without learnable parameters.

Additionally, we visualized the attention coefficients under the opposite setting—strong structure noise and weak feature noise—and found that GATv2 assigns highly differentiated weights to neighbors. This selective weighting highlights important neighbors while down-weighting noisy ones, thereby enhancing the model's final classification performance.

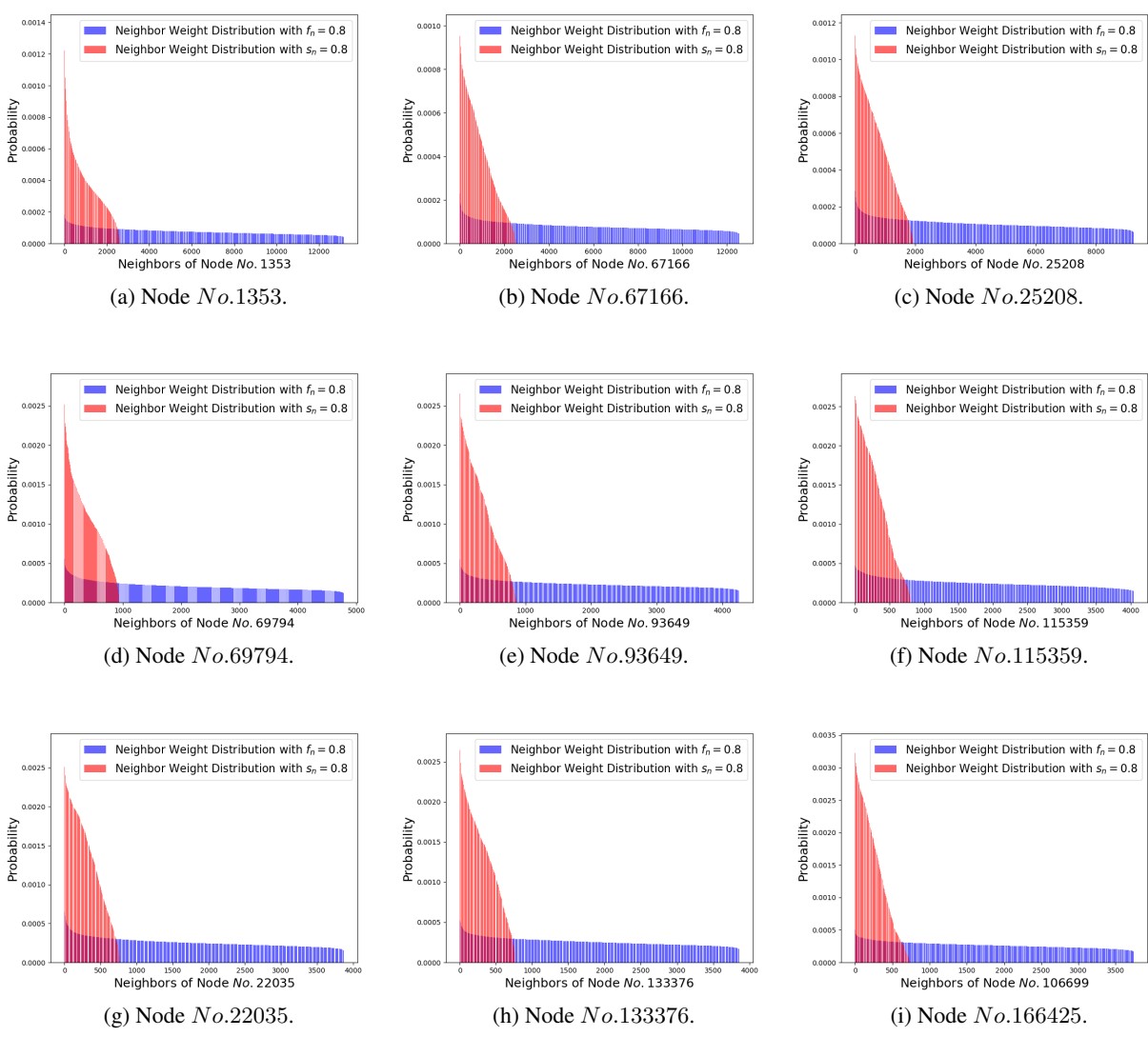

(a) Node $No.1353$.  (b) Node $No.67166$.  (c) Node $No.25208$.

(d) Node $No.69794$.  (e) Node $No.93649$.  (f) Node $No.115359$.

(g) Node $No.22035$.  (h) Node $No.133376$.  (i) Node $No.166425$.

*Figure 5.* Comparison of attention coefficients under high feature noise ($f_n = 0.8$) and high structure noise ($s_n = 0.8$) on the `ogbn-arxiv` dataset. The visualization is based on the last layer of the GATv2 model, focusing on the top 9 nodes with the highest degrees.

