# OpenReview forum: "Graph Attention is Not Always Beneficial: A Theoretical Analysis of Graph Attention Mechanisms via Contextual Stochastic Block Models"
_ICML.cc/2025/Conference — ICML 2025 poster_

### Official Review · Reviewer_BjhC · 2025-02-25

**Overall Recommendation:** 3

**Summary:**

The paper rigorously investigates when graph attention mechanisms help—and when they do not—in the context of node classification for graphs generated by Contextual Stochastic Block Models (CSBM). It introduces a simplified non-linear attention mechanism and demonstrates theoretically that attention improves classification when structure noise outweighs feature noise, but may degrade performance when the reverse is true. The analysis further claims that, in high signal-to-noise regimes, graph attention can effectively counteract the over-smoothing problem that plagues traditional graph convolutional networks. Building on these insights, the authors propose a novel multi-layer Graph Attention Network architecture that substantially relaxes the conditions for perfect node classification compared to single-layer variants. Experiments on synthetic and real-world datasets seem to corroborate these theoretical findings.

**Claims And Evidence:**

While the paper provides rigorous theoretical analyses and supportive experiments within the CSBM framework, some claims may be viewed as less convincingly supported in broader contexts. In particular, the assertion that multi-layer GATs can achieve perfect node classification under significantly relaxed signal-to-noise conditions is heavily dependent on idealized assumptions inherent in the CSBM, which might not extend to more heterogeneous real-world graphs.

**Essential References Not Discussed:**

While I appreciate the contributions of the paper, a discussion of previous attempts at rigorously understanding attention in graph neural networks seems to be missing or at the very least incomplete. In particular, [1] establish that attention in GNNs cannot mitigate oversmoothing. Could the authors clarify why this does not contradict their results?

[1] Wu, X., Ajorlou, A., Wu, Z. and Jadbabaie, A., 2023. Demystifying oversmoothing in attention-based graph neural networks. Advances in Neural Information Processing Systems, 36, pp.35084-35106.

**Experimental Designs Or Analyses:**

I reviewed the experimental designs and analyses, particularly those in Section 4. The experiments are generally well-structured to validate the theoretical claims, with controlled synthetic settings that mirror the assumptions of the CSBM and provide clear benchmarks for evaluating over-smoothing and SNR improvements. However, as I mentioned above, the limited number and scale of the real-world datasets used is a notable issue.

**Methods And Evaluation Criteria:**

One notable limitation is the reliance on only a few standard datasets—Citeseer, Cora, and Pubmed—which, while widely recognized, are relatively small and may not represent the complexity or scale of contemporary real-world graphs. This constrained evaluation could limit the generalizability of the findings, as the performance and robustness of the proposed multi-layer GAT architecture in larger, more diverse networks remain untested. Evaluating on a broader and more challenging collection of datasets would provide stronger evidence of the method’s practical utility across a variety of realistic scenarios.

**Other Comments Or Suggestions:**

I would encourage the authors to extend their experiments to larger datasets. For example, I would appreciate it if the authors could provide some results on the heterophilous node classification datasets [2].

[2] Platonov, O., Kuznedelev, D., Diskin, M., Babenko, A. and Prokhorenkova, L., 2023. A critical look at the evaluation of GNNs under heterophily: Are we really making progress?. arXiv preprint arXiv:2302.11640.

**Other Strengths And Weaknesses:**

-

**Questions For Authors:**

Please see "Essential References Not Discussed".

**Relation To Broader Scientific Literature:**

The paper’s contributions extend a well-established line of research on graph neural networks by deepening our understanding of when and how graph attention mechanisms provide benefits. It builds on previous findings regarding the limitations of standard aggregation methods, such as over-smoothing in deep architectures, and clarifies the interplay between different types of noise in graph data. By deriving precise conditions under which attention mechanisms enhance performance compared to simpler operations, the work refines theoretical models of node classification. Moreover, it introduces a multi-layer architecture that relaxes previously stringent signal-to-noise requirements.

**Theoretical Claims:**

I examined the proofs for the core theoretical claims, particularly those underpinning Theorems 1, 2, and 3. Overall, the proofs appear largely rigorous; however, some steps—especially those involving asymptotic bounds and the handling of high-probability events—could benefit from additional clarification.

---

> ### Author Rebuttal · Authors · 2025-03-31
>
> Thank you for your valuable feedback and recognition of our paper. In response to your questions and suggestions, we provide the following clarifications:
>
> ### 1. **Additional experiments on more comprehensive datasets:**
>    Based on your suggestion, we conducted supplementary experiments on a larger dataset (*obgn-arxiv*) and five heterophily datasets [2] to validate the correctness and practical value of our theoretical results. Please refer to the following link for the results and details: https://drive.google.com/file/d/1ALWkkazk1LPjaWSSkL28RCM7ywOsoBEW/view?usp=drive_link. Below is a brief overview of the experimental setup and corresponding results.
>
> (1). According to our theoretical findings, when feature noise dominates, the graph attention mechanism becomes ineffective, and we should reduce the attention intensity parameter $t$. Conversely, when structure noise dominates, the attention mechanism becomes effective, and we should increase the attention intensity parameter $t$. Accordingly, we design GATv2*(temp), a graph attention model with adjustable attention intensity, based on this idea. Actually, the parameter $t$ can be seen as the reciprocal of temperature adjustment coefficient $T $ applied to the softmax layer in the attention coefficient computation (i.e., $ T = t^{-1} $). Therefore, we adjust the attention mechanism's intensity by tuning the softmax temperature coefficient $T$.  In the original paper, this experiment was implemented only on simulated datasets, but we extend it to real-world datasets in the supplementary experiments. We conduct experiments on six real-world datasets, and for the heterophily datasets with stronger structure noise, we use a design to enhance the attention intensity by setting $ T = t^{-1} = [0.2, 0.5, 1] $ for the three layers. For the more homophilic *ogbn-arxiv* dataset, we used an initial smaller and then progressively larger attention intensity (corresponding to the discussion in lines 369-376 of the paper), i.e., $ T = t^{-1} = [2, 2, 1] $. The accuracy results of the different models are shown in Figure 1 and Table 1 in the above link, with GATv2*(temp) achieving the best overall performance.
>
>  (2). Additionally, we conduct a comparison of different models under two types of noise on the *ogbn-arxiv* dataset and plot accuracy heatmaps for each model. Upon observation, we find that the GAT-based method show an improvement over GCN primarily in areas with stronger structure noise, which is reflected in the upper portion of the heatmap. This validates our theoretical findings.
>
> (3). In the case of strong feature noise, the GAT-based method did not show a significant performance degradation. Upon visualizing the parameters, we find that the GAT method, through learning, assigned nearly equal weights to all neighbors, eventually degenerating into a GCN method. However, when structure noise is strong, GAT perform a noticeable selection of valuable neighbors, which lead to superior performance over GCN. These results can be clearly seen in Figure 3 of the linked material.
>
> ### 2. **Clarification on the contradiction with [1]:**
>    This is a very valuable question, and we appreciate your inquiry. The apparent contradiction with the conclusions in [1] lies in the different definitions in the measure of over-smoothing. The over-smoothing definition used in [1] is derived from [3], where the relationship between node features and the number of layers $ L $ is analyzed without considering the relationship between layer depth and the number of nodes $n$. As a result, even if over-smoothing occurs after $ L = \omega(n) $ layers, the over-smoothing measure defined in [1] would still detect it as an over-smoothing phenomenon, which does not align with real-world scenarios. In practice, over-smoothing typically happens at much smaller depths than the total number of nodes.
>
>    Therefore, we introduce an improved measure of over-smoothing in our paper, restricting $ L = O(n) $, as detailed at the beginning of Section 3.3 and in Definition 2. Additionally, our simulation results, shown in Figure 1(c), demonstrate this phenomenon: in regimes with sufficiently high SNR, as the attention intensity $ t $ increases, the over-smoothing measure (i.e., node-similarity measure) decays from an exponential to a nearly linear change.
>
> [1] Wu, X., Ajorlou, A., Wu, Z., and Jadbabaie, A. Demystifying oversmoothing in attention-based graph neural networks.
>
> [2] Platonov, O., Kuznedelev, D., Diskin, M., Babenko, A., and Prokhorenkova, L. A critical look at the evaluation of GNNs under heterophily: Are we really making progress?.
>
> [3] Rusch, T. K., Bronstein, M. M., and Mishra, S. A survey on oversmoothing in graph neural networks.

---

### Official Review · Reviewer_QLjj · 2025-03-12

**Overall Recommendation:** 2

**Summary:**

The paper studies effectiveness of graph attention networks in the contextual stochastic block model (CSBM) setting. It builds on prior work by Fountalakis et al (JMLR, 2023) and graph attention retrospective in the same setting. In comparison to that work, the main difference appears to be a linear multi-layer graph attention mechanism with non-linearity at the end, along with an assumed simpler attention scoring function (Eq. 3).
The main focus is on theoretical results in this simplistic setting, and trade-offs between SNR (Eq. 2) and inverse SNR relative to over-smoothing. There are several results on over-smoothing, but most compelling ones are in Theorems 3 and 4 which indicate at setting where linear multi-layer GAT might be helpful and graph convolutions suffer from over-smoothing.

### POST-REBUTTAL
I will not be arguing against this paper and if the other reviewers feel strongly about it, I'm fine that it is accepted.

**Claims And Evidence:**

Several theoretical results but not really sure how realistic the underlying assumptions are and the overall setting, especially in the context of real-world applications and graph attention architectures. Experimental results provided on synthetic data and three real-world datasets. In the latter, there does not appear to be differentiation in performance between GAT* (multi-layer linear graph attention) and GCNs.

**Essential References Not Discussed:**

I’d say that the main reference was listed but the discussion could be improved relative to Fountalakis et al (JMLR, 2023), both in terms of proof techniques, limitations of the problem setting, and results.

**Experimental Designs Or Analyses:**

Limited set of experiments and this aspect of the paper can be improved, especially when illustrating the point on GCNs and GAT* on real-world datasets.

**Methods And Evaluation Criteria:**

This is a theoretical paper and might be fine with a limited set of experiments. However, these don’t seem to illustrate difference between GCNs and GAT*s on real-world datasets. Proof techniques, in my understanding, are mimicking prior work by Fountalakis et al (JMLR, 2023) and in that regard the contribution is rather incremental.

**Other Comments Or Suggestions:**

NA

**Other Strengths And Weaknesses:**

See above

**Questions For Authors:**

NA

**Relation To Broader Scientific Literature:**

Related work appears to be adequately covered.

**Theoretical Claims:**

I have not checked the proofs carefully, but have skimmed through few of them. What I did checked seemed fine, but there was one aspect that was confusing. The entire analysis seems to be done via the simplistic attention mechanism in Eq. (3). In that regard, the mechanism in Fountalakis et al (JMLR, 2023) while still simplistic seemed more characteristic of what one can expect in GATs.

---

> ### Author Rebuttal · Authors · 2025-03-31
>
> Thank you for your valuable feedback. Our response is as follows:
>
> ### 1. **Additional Experiments**
>
> You mentioned that the experiments on real-world datasets were insufficient. We greatly appreciate your feedback and have conducted additional experiments on more comprehensive datasets, with the results available at https://drive.google.com/file/d/1ALWkkazk1LPjaWSSkL28RCM7ywOsoBEW/view?usp=drive_link. We select an additional 6 datasets, including a larger-scale dataset, *ogbn-arxiv*, and 5 heterophily graph datasets (e.g., *roman-empire*). In addition to the basic GAT model, we also employ GATv2 attention mechanism.
>
> These new experiments on the extended datasets validate both the correctness and practical value of our theoretical results. Specifically, based on our theoretical findings (Theorem 2), similar to GAT*, we design GATv2*(temp), which adjusts the attention intensity (i.e., the parameter $ t $ in the paper). We confirm that by simply selecting the value of $ t $ based on dataset and noise characteristics, we can achieve better results than the baseline GAT model across multiple datasets (see Figure 1 and Table 1 in the above link). In the presence of feature noise and structure noise, we plot accuracy heatmaps for different algorithms to make a more detailed comparison. GATv2*(temp) performed the best, particularly under high noise conditions (see Figure 2 in the above link). Notably, we find that the parameter $ t $, which controls the attention intensity, is actually the inverse of the temperature coefficient $ T $ that controls the sharpness of the softmax output attention coefficients distribution (i.e., $ t = T^{-1} $), so we adjust the attention intensity by tuning the value of $ T $ in the experiments.
>
> Furthermore, we observe that when only feature noise was added, the GAT-based method did not show a clear advantage or disadvantage compared to GCN. By visualizing the attention weights (see Figure 3 in the above link), we find that GAT assigned nearly equal weights to all neighbors, eventually becoming a GCN. However, with strong structure noise, GAT successfully selected valuable neighbors and outperformed GCN.
>
> ### 2. **Comparison of Proof Techniques with Fountalakis et al. (JMLR, 2023)**
>
> Our work is inspired by the JMLR paper, but the proof techniques we use are significantly different, and we have derived broader and more novel results. Specifically:
>
> (1) **We consider multi-layer GAT, whereas Fountalakis et al. (JMLR, 2023) considers a single-layer GAT.** We must emphasize that this extension is **non-trivial** and involves several theoretical challenges.
>
> The analysis of multi-layer GATs requires us to accurately characterize the distribution of node features after passing through the GAT layers, rather than simply focusing on the sign of the output, as in single-layer GAT analysis (as done in JMLR23). The attention mechanism used in JMLR23 is too complex to be analyzed in a multi-layer setting, which is why we choose to design a simpler attention mechanism for theoretical analysis. Although this mechanism is simpler, we theoretically prove that its performance in the CSBM perfect node classification task is comparable to that of the attention mechanism in JMLR23, as shown in Theorem 1. The proof techniques used here are the only part that shares similarities with those in JMLR23.
>
> However, the core of our proof is presented in Theorem 2, which precisely characterizes the distribution of node features after one GAT layer. This is a significant challenge because, even with the simplest attention mechanism, the process is highly nonlinear, meaning that after passing through the GAT layer, node features no longer follow a Gaussian distribution. Moreover, this distribution is also influenced by the number and distribution of neighboring nodes, introducing further randomness, all of which make it difficult to accurately characterize this distribution. In our paper, proof of Theorem 2 spans pages 17 to 32 of the appendix, and this part forms the core of our theoretical analysis. Finally, extending from single-layer to multi-layer analysis is a well-known challenge in deep learning theory, with related works published in top venues such as FOCS, COLT, and JMLR.
>
> (2) **We also address the over-smoothing issue (see Theorem 3).** This analysis was not covered in Fountoulakis et al. (2023) and involves many completely different proof techniques. We define a new measure for over-smoothing and, using the results from Theorem 2 on the distribution changes of node features in multi-layer GAT, proved that graph attention mechanisms can positively mitigate the over-smoothing problem. We also find that the effectiveness of graph attention mechanisms in addressing over-smoothing depends on the SNR of the graph data. The higher the initial SNR, the more pronounced the role of the graph attention mechanism.
>
> Thank you for your thoughtful feedback, and we look forward to further discussions.

---

> > ### Comment · Reviewer_QLjj · 2025-04-02
> >
> > Thank you for the additional experiments. Unfortunately, I’m not keen on following a link to google-drive and would appreciate if the experiments were summarized in the response. My understanding is that the additional experiment shows that GATv2* version does behave as the theoretical results indicate but that the same might not be true for GAT*.
> >
> > I have listed in my summary that the paper considers multi-layer GAT but not in the classical sense. Namely, the focus is on multi-layer *linear GAT* with a single non-linearity at the end. While slightly different from Fountalakis et al. where there was a single linear layer, the difference is not substantial relative to how GATs are used in practice. I also did not rate theoretical contributions as trivial but the results, findings, and overall technique as incremental.
> >
> > I have read all the reviews and the rebuttal, and will make the final decision during the discussion with other reviewers.

---

> > > ### Author Response · Authors · 2025-04-03
> > >
> > > Thank you for your response. For your convenience, as well as for those reviewers who may not have had time to view the link, we summarize the key aspects of our additional experiments as follows:
> > >
> > > ## **1. Experimental Settings and Main Results**
> > > We conducted supplementary experiments on six additional datasets and explored two attention mechanisms: GAT and GATv2. For the *ogbn-arxiv* dataset, we gradually increased both feature noise and structure noise, and recorded the performance of each model. For comparison, in addition to the GAT* setup from the original paper (which applies a GCN layer followed by a graph attention layer), we designed GAT*(temp) and GATv2*(temp) to precisely control the intensity of the attention mechanism, parameterized by $ t $. Notably, GAT* can be seen as a **special case** of GAT*(temp) where the first layer’s attention strength is fixed at $t=0$. Another motivation behind introducing GAT*(temp) and GATv2*(temp) was to handle heterophily graph datasets, where increasing rather than decreasing the attention intensity is necessary.
> > >
> > > For the larger homophily dataset *ogbn-arxiv*, we use GAT(v2)* and GAT(v2)* (temp) as comparison methods, with the primary focus on reducing the attention intensity $ t $. For the five heterophily datasets, we use GAT(v2)*(temp) as the comparison method, with a larger attention intensity $ t$. Below, we provide a table that reports not only the classification accuracy of each model across different datasets but also the corresponding values of the attention intensity parameter $ t $.
> > >
> > > |  | ogbn-arxiv | ogbn-arxiv (with high $F_{noise}$) |  ogbn-arxiv (with high $S_{noise}$) | Attention Intensity $t$ | roman-empire | amazon-ratings | minesweeper | questions | tolokers | Attention Intensity $t$ |
> > > |-------------|----------------|-----------|------|----------|------------|-----------|--------|----------|----------|----------|
> > > | GCN | 71.06%    | 69.23%  | 47.89% | $[0, 0, 0]$  | 32.94%    | 46.96%    | 58.21%    | **57.44%**    | 65.04%    | $[0, 0, 0]$                      |
> > > | GAT| 70.86% | 68.63%    | 50.31% | $[1, 1, 1]$ | 44.72%   | 48.62% | 65.95%        | 52.68% | 65.46% | $[1, 1, 1]$ |
> > > | GAT*| 70.89%    | 69.60%| 50.60% | $ [0, 1, 1]   $ | /| /| /| /| /                 |/              |
> > > | GAT*(temp)| 70.98% | 69.57% | 50.86% | $ [\frac{1}{2},\frac{1}{2}, 1] $ | 46.69%   | 49.08% | 69.33% | 52.81% | 66.71% | $ [2, 2, 1] $ |
> > > | GATv2| 71.41% | 69.17% | 60.04% | $ [1, 1, 1] $  |74.45%   | 48.89% | **70.92%** | 53.85% | 66.46% | $ [1, 1, 1] $ |
> > > | GATv2*| 71.19% | 69.82% | 60.29% |  $ [0, 1, 1] $  | / | /    | /| /| /               | / |
> > > | GATv2*(temp) | **71.61%** | **69.97%** | **61.36%** | $ [\frac{1}{2},\frac{1}{2}, 1] $ | **76.91%**   | **49.38%** | **70.89%** | 53.12% | **67.44%** | $ [2, 2, 1] $             |
> > >
> > > ## **2. Conclusion**
> > > As you pointed out, our experiments validate our theoretical results. Specifically, GAT*(temp) and GATv2*(temp) consistently outperform baseline models. The results for GAT* and GATv2* also align with our theoretical conclusions, though models with tunable $t$ (i.e., the (temp) variants) demonstrate superior performance.
> > >
> > > On the *obgn-arxiv* dataset, graph attention-based methods show a significant improvement when structure noise is strong. However, when feature noise is dominant, they have little to no effect (with the original version of GAT performing even worse than GCN). Moreover, in heterophily graph datasets, although it falls outside the scope of our paper ($p > q$), this can be seen as a case with very strong structure noise, where the noise from neighboring nodes features even outweighs the information. A simple conjecture is that a stronger graph attention intensity, i.e., a larger parameter $ t $, is needed, which is also confirmed by our experiments.
> > >
> > >
> > > ## Additional Clarification:
> > > We would like to clarify that, although our theoretical analysis is based on a simplified version of GAT, it provides new insights by precisely characterizing the conditions under which graph attention mechanisms are effective. Additionally, our theoretical results offer clear guidance on how to adjust attention intensity under different noise conditions, which serve as a crucial reference for selecting and tuning attention mechanisms in practical applications. During our supplementary experiments, we observed that most existing open-source graph attention mechanisms lack the ability to control over attention intensity. However, our theoretical and empirical findings reveal that incorporating such a control mechanism significantly enhances performance—an improvement that is applicable to nearly all graph attention models. This could be an important broader impact for practical value of our work.
> > >
> > > Thank you once again for your response. We hope our explanation of the additional experiments has addressed your concerns and look forward to further discussions with you.

---

### Official Review · Reviewer_JGKV · 2025-03-17

**Overall Recommendation:** 3

**Summary:**

The paper theoretically analyzes the effectiveness of graph attention for node classification tasks in graphs with a CSBM structure and varying levels of feature and structure noise and conclude that high feature noise renders graph attion ineffective whereas graph attention is beneficial in the case of low feature noise and high structure noise.

## update after rebuttal:
I thank the authors for answering my questions and appreciate their efforts in providing further results during the rebuttal. While I am not too fond of tuning another parameter (t) for each layer in the model, the concept of attention intensity is interesting and perhaps it could be possible to make it learnable in the future. Also, while it intuitively makes sense that the attention coefficients would vary more when a node has a more varied neighborhood (possibly due to high structure noise), a rigorous theoretical analysis to back it is good. Secondly, attention coefficients do not necessarily always learn varied and sparser patterns due to trainability issues. In such cases, having an aid such as an attention intensity parameter to improve the attention mechanism could be promising. Therefore, I have raised my score.

**Claims And Evidence:**

The authors supports their theoretical claims with proofs and empirical evidence from experiments on synthetic data.

**Essential References Not Discussed:**

None that I can recall.

**Experimental Designs Or Analyses:**

While the experiments to verify the theorical claims of the relationship between SNR ratio and effectiveness of attention mechanism are well designed, I do think that experiments that show the benefit of the proposed/analyzed attention mechanism on real-world datasets is missing. Is it competitive with the standard attention mechanism used in GAT [5] or its various variants? See review section 'method and evaluation criteria' for  a similar discussion.

[5] Brody et al. How Attentive are Graph Attention Networks?

**Methods And Evaluation Criteria:**

- While the theoretical claims are supported by empirical evidence on synthetic datasets, their practical benefits for real-world scenarios is not very clear. For example, the authors mention that their findings provide insights for practical applications  such as selecting graph attention based on graph data characteristics and designing noise-robust networks. However, the SNR ratio, based on which the decision to use graph attention or not is to be made, is not usually a known property of the graph beforehand.

- Secondly, it is unclear how $t$ is determined, for instance, is it a learnable parameter or a hyperparameter to be tuned with grid search In the experiments, it is an independent variable that is manually varied.

- Furthermore, it is mentioned that the new proposed attention mechanism is designed with homophilic graphs in mind. While this is a  limtiation in my opinion given the current focus on devising attention  based methods that adapt to both homophily and heterophily [1-4], I would still like to request the authors for further clarification on this, because intuitively it seems that having both negative and positive and negative values for $t$ based on dot product of  of two node feature vectors that signify their similarity/dissimilarity and/or alignment/unalignment could also possibly deal with both homophily and heterophily.

[1] Eliasof et al. Improving Graph Neural Networks with Learnable Propagation Operator

[2] Bo et al. Beyond Low-frequency Information in Graph Convolutional Networks

[3] Mustafa et al. GATE: How to Keep Out Intrusive Neighbors

[4]  Finkelshtein et al. Cooperative Graph Neural Networks

**Other Comments Or Suggestions:**

• It may be worthwhile to visualize the distribution of actual attention coefficients ($c_ij$) for the conducted experiments to better understand/verify whats going on under the hood (for example, in a similar way to [3]).

[3] Mustafa et al. GATE: How to Keep Out Intrusive Neighbors

**Other Strengths And Weaknesses:**

**Strengths**

The paper contributes to theoretically understanding an important aspect of graph learning, i.e. graph attention.


**Weaknesses**

The analyzed graph attention is different from the mechanism in the original GAT or its variants that are usually employed and the effectiveness of then proposed attention mechanism is unclear in the real-world setting. A more detailed discussion of this is in the evaluation criteria and experiment design sections of the review.

**Questions For Authors:**

1. For experiments on real-world datasets, are the original node features discarded, since perfect node classification is achieved?

2. How can the value of $t$ be determined for real-world datasets?

3. Are inter-class edges considered as structural noise? If so, is this a fair assumption as it is not necessarily true and also unlikely in the real-world setting with perfect node classification not always being aligned with the community structure?

4. Further questions inline in the review sections of evaluation criteria and experimental design.

Edit: While I am not too fond of tuning another parameter (t) for each layer in the model, the concept of attention intensity is interesting and perhaps it could be possible to make it learnable in the future. Also, while it intuitively makes sense that the attention coefficients would vary more when a node has a more varied neighborhood (possibly due to high structure noise), a rigorous theoretical analysis to back it is good. Secondly, attention coefficients do not necessarily always learn varied and sparser patterns due to trainability issues. In such cases, having an aid such as an attention intensity parameter to improve the attention mechanism could be promising, as shown by the authors in the additional experiments and analysis during the rebuttal. Therefore, I have raised my score.

**Relation To Broader Scientific Literature:**

The role of attention in graph learning is of great interest to the GNN community today. While most literature on improving/understanding graph attention is focused on the original GAT architecture or its various variants, this paper analyzes a simpler attention mechanism but provides novel insights into its relationship with signal-noise ratio of node features. However, its applicability in real-world scenarios is not well-evaluated.

**Theoretical Claims:**

The theoretical claims seem correct but the mathematical details were not checked in detail.

---

> ### Author Rebuttal · Authors · 2025-03-31
>
> Thank you for your thoughtful comments. In response to multiple reviewers’ suggestions, we have added experiments on a larger dataset (*ogbn-arxiv*) and five heterophily datasets (e.g., *roman-empire*). Please find the details at the following link https://drive.google.com/file/d/1ALWkkazk1LPjaWSSkL28RCM7ywOsoBEW/view?usp=drive_link. Below, we address your concerns point by point.
>
> ### 1. **Practical Benefits in Real-World Scenarios**
> Our conclusions consider both SNR and structure noise, providing insights into when and how to use graph attention mechanisms. Specifically, by measuring these two factors, we can determine whether to apply graph attention and how to adjust its intensity (see Remark 3 in lines 278-292 and the discussion in lines 369-376).
>
> Moreover, the SNR of real-world datasets is not entirely unknown. It can be estimated using the expectation and variance of node features for each class. Even if not all node features are observable, we can approximate SNR using only training data. More importantly, the absolute SNR value matters less than its comparison with structure noise, which helps guide the choice and design of GAT. For example, in heterophily graphs with high structure noise, increasing the intensity of graph attention is beneficial. Conversely, if the features of nodes from different classes are not well distinguishable or have high variance, it may be preferable to avoid using GAT or reduce its intensity.
>
> ### 2. **Choice of Parameter $t$**
> First, we would like to emphasize that the attention mechanism proposed in this paper is a simplified version, designed to distill the essence of various graph attention mechanisms for theoretical analysis. The parameter $ t $ controls the intensity of attention and is set based on structure and feature noise before running experiments.
>
> Conceptually, if we view our attention mechanism as assigning +1 weight to intra-class edges and -1 to inter-class edges before applying softmax, then $ t $ functions similarly to the temperature coefficient $ T $ in softmax, controlling the sharpness of the output weight distribution, where $ t = \frac{1}{T} $.
>
> ### 3. **Analysis of Heterophily**
> This is a great question. For the binary symmetric CSBM model, a heterophily analysis can indeed be performed. When $ q > p $, we can modify Equation (3) to assign a weight of $ -t $ to edges where $ X_i X_j > 0 $ and $ t $ to edges where $ X_i X_j < 0 $, without significantly affecting later results.
>
> However, we focus on the homogeneity assumption ($ p > q $) because real-world tasks often involve multi-class classification, where simply adjusting $ t $ is insufficient and requires more complex analysis. More fundamentally, in heterophily graphs, a key question is whether GNNs’ message-passing paradigm remains effective. Understanding how to incorporate global structure information into node features should take precedence over studying attention mechanisms in this setting.
>
> ### 4. **Experiments and Analysis**
> As noted earlier, our proposed attention mechanism is a simple version designed for theoretical analysis and does not include learnable parameters. However, as discussed in point 2, our findings suggest a simple way to improve existing attention mechanisms by introducing a temperature coefficient $ T $ (i.e., $ t^{-1} $) in the softmax layer to adjust attention intensity.
>
> Our theory indicates that when structure noise dominates, a larger $ t $ (smaller $ T $) is preferable, whereas when feature noise dominates, a smaller $ t $ (larger $ T $) works better. We add this in GATv2, denoted as GATv2* (temp), and show that it consistently outperforms the unmodified GATv2 across multiple datasets, including heterophily graphs (see Figure 1,2 and Table 1 in the link above).
>
> ### 5. **Visualization of Attention Coefficients**
> This is a great suggestion. We add visualizations of attention coefficients in our supplementary experiments (see Figure 3 in the link above). We find that, for the *ogbn-arxiv* dataset, when feature noise increases, the performance of the GAT-based model is similar to that of GCN, with neither improvement nor degradation. The visualizations results show that this happens because GAT, by learning to assign equal weights to all neighbors, essentially degenerates into GCN. However, when structure noise increases, GAT shows significant performance gains (see Figure 1 in the link above for the comparison). The attention weight visualization in Figure 3 in the link above confirms that GAT effectively filters important neighbors in such cases.
>
> ### Additional Clarifications:
> (1) No, the preprocessing of node features is the same as in all GNNs.
> (2) See point 2 and 4 for details.
> (3) Not all inter-class edges are considered noise; noise is measured by the ratio of inter-class to intra-class edges, $ \frac{p+q}{p-q} $.
>
> We appreciate your feedback and hope these responses clarify your concerns.

---

> > ### Comment · Reviewer_JGKV · 2025-04-09
> >
> > I thank the authors for answering my questions and appreciate their efforts in providing further results. While I am not too fond of tuning another parameter (t) for each layer in the model, the concept of attention intensity is interesting and perhaps it could be possible to make it learnable in the future. Also, while it intuitively makes sense that the attention coefficients would vary more when a node has a more varied neighborhood (possibly due to high structure noise), a rigorous theoretical analysis to back it is good. Secondly, attention coefficients do not necessarily always learn varied and sparser patterns due to trainability issues. In such cases, having an aid such as an attention intensity parameter to improve the attention mechanism could be promising. Therefore, I have raised my score.

---

> > > ### Author Response · Authors · 2025-04-09
> > >
> > > Thank you very much for recognizing our response and for raising your score. Making the attention intensity a learnable parameter is an interesting idea, and we will explore this direction in the future work.

---

### Decision · Program_Chairs · 2025-05-01

**Decision:**

Accept (poster)

**Comment:**

This paper extends the work of Fountoulakis et al. (JMLR, 2023). The JMLR paper analyzes the single-layer GAT model. The present paper extends this work to the multilayer setting. Analyzing the multilayer setting using CSBM is extremely difficult due to recursive non-linearities. The authors bypass some of these issues by placing an activation layer only at the final layer. However, the non-linearities in the attention weights are retained. It is this fact that makes the current work non-trivial.

I believe this paper makes a valuable theoretical contribution and is worthy of acceptance at ICML. However, I would like the authors to include in the paper all the limitations that have been pointed out by the reviewers.